## RESEARCH ARTICLE

# Broad H3K4me3 domains support oocyte genome silencing and maturation but are dispensable for repression in early embryos

Trine Skuland[1,2,3], Madeleine Fosslie[3,4], Sherif Khodeer[4,*], Endalkachew Ashenafi Alemu[4,‡],
Jens Vilstrup Johansen[5,§], Marie Indahl[1,3], Blanca Corral Castroviejo[1,3], Yanjiao Li[3,4], Mads Lerdrup[3,6],
John Arne Dahl[3,4], Peter Zoltan Fedorcsak[1,2,3,¶] and Gareth D. Greggains[1,3,¶]

## ABSTRACT

Epigenetic reprogramming and embryonic genome activation (EGA) are crucial events during early development. Establishment of distinctive broad histone H3 lysine 4 trimethylation (H3K4me3) domains in the oocyte is necessary for genome silencing, and their removal in the 2-cell embryo is crucial for EGA and development in mice. However, the stage-specific requirement for broad H3K4me3 domains is unclear. Here, we show that inducing broad H3K4me3 removal in mouse oocytes can relieve genome silencing, impair oocyte maturation timing, and may cause transcriptional reactivation in resulting parthenogenetic 1-cell embryos. We further demonstrate that broad H3K4me3 demethylation precedes EGA but, surprisingly, premature depletion in zygotes or early 2-cell embryos does not alter the transcriptional program. Our findings suggest that broad H3K4me3 domains are required for oocyte genome silencing, timely maturation and post-fertilisation silencing, but onward pre-EGA transcriptional repression is not dependent on the original mark. This work contributes to the understanding of events and mechanisms involved in genome silencing and activation in early development, providing insight into potential modes of failure that may contribute to infertility.

**KEY WORDS: Embryo, Activation, H3K4me3, Oocyte, Silencing, KDM5B**

[1]Department of Reproductive Medicine, Women's Clinic, Oslo University Hospital, 0855 Oslo, Norway. [2]Division of Gynaecology and Obstetrics, Institute of Clinical Medicine, Faculty of Medicine, University of Oslo, 0450 Oslo, Norway. [3]CRESCO - Centre for Embryology and Healthy Development, Institute of Clinical Medicine, Faculty of Medicine, University of Oslo, 0373 Oslo, Norway. [4]Department of Microbiology, Division of Laboratory Medicine, Oslo University Hospital, 0373 Oslo, Norway. [5]Biotech Research & Innovation Centre (BRIC), University of Copenhagen, 2200 Copenhagen, Denmark. [6]DNRF Centre for Chromosome Stability, Department of Cellular and Molecular Medicine, University of Copenhagen, 2200 Copenhagen, Denmark.
*Present address: The Stem Cell Institute, Department of Development and Regeneration, University of Leuven, 3000 Leuven, Belgium. ‡Present address: Department of Medical Genetics, Division of Laboratory Medicine, Oslo University Hospital, 0450 Oslo, Norway. §Present address: Center for Gene Expression (CGEN), University of Copenhagen, 2200 Copenhagen, Denmark.

¶Authors for correspondence (g.d.greggains@ous-research.no; p.z.fedorcsak@medisin.uio.no)

T.S., 0000-0001-9618-5123; M.F., 0000-0002-2902-4394; G.D.G., 0000-0003-2424-4694

## INTRODUCTION

Histone modifications play a key role in regulating gene expression and cell identity during mammalian development (Eckersley-Maslin et al., 2018). The presence of histone H3 lysine 4 trimethylation (H3K4me3) at transcriptional start sites is associated with active genes in a range of species and cell types (Bernstein et al., 2005; Heintzman et al., 2007; Santos-Rosa et al., 2002). Remarkably, mouse metaphase II (MII) oocytes exhibit a unique broad H3K4me3 pattern, which covers 22% of the oocyte genome (Dahl et al., 2016; Zhang et al., 2016). In contrast to the frequent association of H3K4me3 with active genes, these broad H3K4me3 domains have been proposed to promote global genome silencing in oocytes, and contribute to acquisition of meiotic and developmental competence (Andreu-Vieyra et al., 2010; Zhang et al., 2016). Similar broad domains have been identified in bovine, porcine and rat oocytes, but notably, they have not been observed in immature human germinal vesicle (GV) oocytes (Lu et al., 2021; Xia et al., 2019). Nonetheless, human post-fertilisation dynamics show shared features of H3K4me3 reprogramming, and mouse studies remain useful for understanding transcriptional silencing and genome activation. Global genome silencing is observed during late oocyte growth in mouse, and coincides with, but is not dependent on, the transition of the GV from a loosely packed non-surrounded nucleolus (NSN) to a more condensed surrounded nucleolus (SN) chromatin configuration (Bouniol-Baly et al., 1999; De La Fuente et al., 2004; Debey et al., 1993). Knockout of KMT2B (also known as MLL2), the methyltransferase depositing broad H3K4me3 domains (Hanna et al., 2018), at the primordial follicle stage results in incomplete silencing of the SN oocyte genome and ovulation failure (Andreu-Vieyra et al., 2010). Furthermore, overexpression of the H3K4me3 demethylase KDM5B (also known as PLU-1 or JARID1B) can lead to transcriptional reactivation in otherwise silenced SN oocytes (Zhang et al., 2016).

Following oocyte maturation to MII stage and subsequent fertilisation, activation of the newly constituted embryonic genome is a crucial event in early development. Failure to properly activate the genome has been proposed as a major cause of embryo arrest, which is frequently seen during *in vitro* culture of human embryos (Mora et al., 2023; Yang et al., 2022). In both human and mouse, a minor genome activation is initiated at the zygote stage (Aoki et al., 1997; Ram and Schultz, 1993; Xue et al., 2013). The timing of major embryonic genome activation (EGA) has been established in several species (Jukam et al., 2017). Mouse embryos exhibit EGA at the 2-cell (2C) stage (Aoki et al., 1997; Flach et al., 1982), whereas human EGA occurs between the 4- and 8-cell stages (Braude et al., 1988). However, the molecular mechanisms controlling transcriptional initiation at EGA are still subject to ongoing research in both species (Anger et al., 2021; Aoki, 2022; Schulz and Harrison, 2019).

**DEVELOPMENT**

Removal of broad H3K4me3 domains in the early preimplantation mouse embryo has been shown to be essential to development. Depletion of KDM5A (also known as RBP2 or JARID1A) and KDM5B in 2C mouse embryos impairs active demethylation of broad H3K4me3 domains, resulting in downregulation of numerous EGA genes and subsequent developmental arrest (Dahl et al., 2016; Liu et al., 2016). However, some findings suggest that removal of broad H3K4me3 domains is a consequence rather than a cause of increased transcription (Abe et al., 2018; Zhang et al., 2016). Alternative hierarchical models of events leading to EGA have been proposed, based on chromatin remodelling, transcription-driven chromatin accessibility, minor genome activation products and an interconnected cascade (Eckersley-Maslin et al., 2018).

Although increased levels of H3K4me3 have been found to be necessary for establishing genome silencing in the mouse oocyte, and removal of broad H3K4me3 domains in 2C embryos is required for normal EGA and development, the role of broad H3K4me3 domains in maintaining transcriptional repression in the mature oocyte and during the early hours after fertilisation is unknown. Additionally, it remains unclear whether continued maintenance of broad H3K4me3 domains is required for oocyte maturation and normal embryo development. Moreover, the exact sequence of events, timing and dynamics of H3K4me3 reprogramming and EGA in the early preimplantation embryo have not been fully characterised. Further investigation is therefore required to establish what role broad H3K4me3 domains play in transcriptional regulation in the oocyte and early embryo, and whether their removal is sufficient for genome reactivation.

In this study, we show that induced removal of broad H3K4me3 domains in SN oocytes can lead to genome reactivation and impaired maturation timing. Furthermore, reduction of H3K4me3 in MII stage oocytes may compromise gene repression in the resulting 1-cell embryos, following parthenogenetic activation. However, while we find that onset of H3K4me3 demethylation precedes EGA, induced removal of broad H3K4me3 domains in zygotes and early 2C embryos does not lead to precocious EGA and is compatible with normal embryo development. Our results indicate that genome silencing promoted by broad H3K4me3 domains is important for timely oocyte maturation and possibly for transmitting transcriptional repression of the maternal genome to the early zygote. Nevertheless, the repressive state is then maintained independently of the original epigenetic mark until EGA.

## RESULTS
### Broad H3K4me3 is involved in oocyte genome silencing, maturation and transfer of transcriptional repression to the zygote

Previous studies have shown that high levels of H3K4me3 are established during oocyte growth and associated with gene silencing (Andreu-Vieyra et al., 2010; Dahl et al., 2016; Zhang et al., 2016). Broad H3K4me3 domains persist in the zygote, with higher levels found in the maternal pronucleus (PN), as well as in the early 2C embryo (Andreu-Vieyra et al., 2010; Dahl et al., 2016; Zhang et al., 2016). Induced removal of H3K4me3 by Kdm5b mRNA injection was found to impair genome silencing in around half of the injected SN oocytes (Zhang et al., 2016), but the scale of transcriptional reactivation and its functional significance remains unclear. To further explore the role of broad H3K4me3 domains in maintaining transcriptional repression in the oocyte and zygote, as well as their functional importance for oocyte maturation, we overexpressed KDM5B in SN- and MII-stage oocytes. We cultured SN oocytes in 3-isobutyl-1-methylxanthine (IBMX), a phosphodiesterase inhibitor

that induces reversible meiotic prophase arrest, to ensure sufficient time for KDM5B expression. For the Kdm5b mRNA used for overexpression, we employed a yellow fluorescent protein (YFP) tag to ensure successful protein expression prior to downstream analysis and/or in vitro maturation (IVM) (Fig. 1A). The KDM5B overexpression approach was validated by live-cell imaging of YFP fluorescence and RNA sequencing (RNA-seq), confirming robust expression at both protein and RNA level (Fig. S1).

To assess transcriptional reactivation and H3K4me3 levels, we used the alkyne-modified nucleoside 5-ethynyl uridine (EU) for labelling nascent RNA, fixed the samples and performed Click-iT RNA imaging assay (here abbreviated as CiRia) coupled with anti-H3K4me3 immunofluorescence staining. Following injection of Kdm5b-yfp mRNA and 12-14 h of arrest (Fig. 1A), we found that 44% of SN oocytes showed elevated levels of RNA transcription (Fig. 1B). However, quantification of nascent RNA levels after KDM5B overexpression showed that levels were significantly lower than in control NSN or intermediate NSN-SN oocytes (Fig. S2A,B), representing an incomplete genome reactivation.

To examine the effect of broad H3K4me3 removal on oocyte maturation, we used a shorter meiotic arrest (Fig. 1A) to avoid potential masking of a KDM5B overexpression phenotype by low overall levels of maturation observed in controls after extended (12-14 h) meiotic arrest (Fig. S2C,D). Kdm5b-yfp mRNA-injection and 6 h arrest was sufficient to reduce H3K4me3 to NSN levels, although transcriptional reactivation was not yet visible by the imaging assay prior to initiation of IVM (Fig. S2E). In the Kdm5b-yfp mRNA-injected group, there was a trend towards fewer oocytes undergoing germinal vesicle breakdown (GVBD) and polar body (PB) extrusion, compared to uninjected and control injected oocytes (Fig. 1C). We therefore assessed timing of IVM to see if these differences could be explained by altered maturation dynamics. We found that Kdm5b-yfp mRNA-injected oocytes underwent GVBD significantly later than water-injected controls, and PB extrusion was delayed, although not significantly so (Fig. 1D).

We next performed IVM and chemical activation on Kdm5b-yfp mRNA-injected SN-stage oocytes to determine if a reduction in H3K4me3 in oocytes would impair genome silencing in early the 1-cell (1C) embryo (Fig. 1A). H3K4me3 levels in Kdm5b-yfp-injected oocytes remained low at MII stage following IVM (Fig. S2F) and in resulting parthenogenetic 1C embryos after activation (Fig. S2G, left), but nascent RNAs in the 1C embryos were at similar levels to controls (Fig. S2G, middle). Interestingly, PN chromatin in the KDM5B overexpression group showed higher H3K4me3 levels than PB chromatin (Fig. S2G, right), in contrast to similar levels seen between MII oocyte chromatin and first PBs (Fig. S2F), suggesting that some H3K4 remethylation may occur in the early 1C embryo. This could reflect degradation of Kdm5b-yfp mRNA as part of maternal RNA decay during meiotic maturation (Paynton et al., 1988), with the resulting elevated H3K4me3 levels potentially preventing genome reactivation in zygotes derived from SN-stage injected oocytes. We therefore injected in vivo matured MII oocytes with Kdm5b-yfp mRNA, immediately followed by chemical activation (Fig. 1E, left). The resulting 1C embryos were then EU labelled, fixed and stained (Fig. S2H). In these parthenogenetic 1C embryos derived from MII oocytes injected with Kdm5b-yfp mRNA, we found that H3K4me3 was maintained at a similarly low level to uninjected 2C embryos at the time of mid-EGA (Fig. S2I). However, in contrast to 1C embryos from SN-stage injected oocytes, 44% of the zygotes originating from MII oocytes injected with Kdm5b-yfp mRNA displayed elevated levels of nascent RNA (Fig. 1E, right).

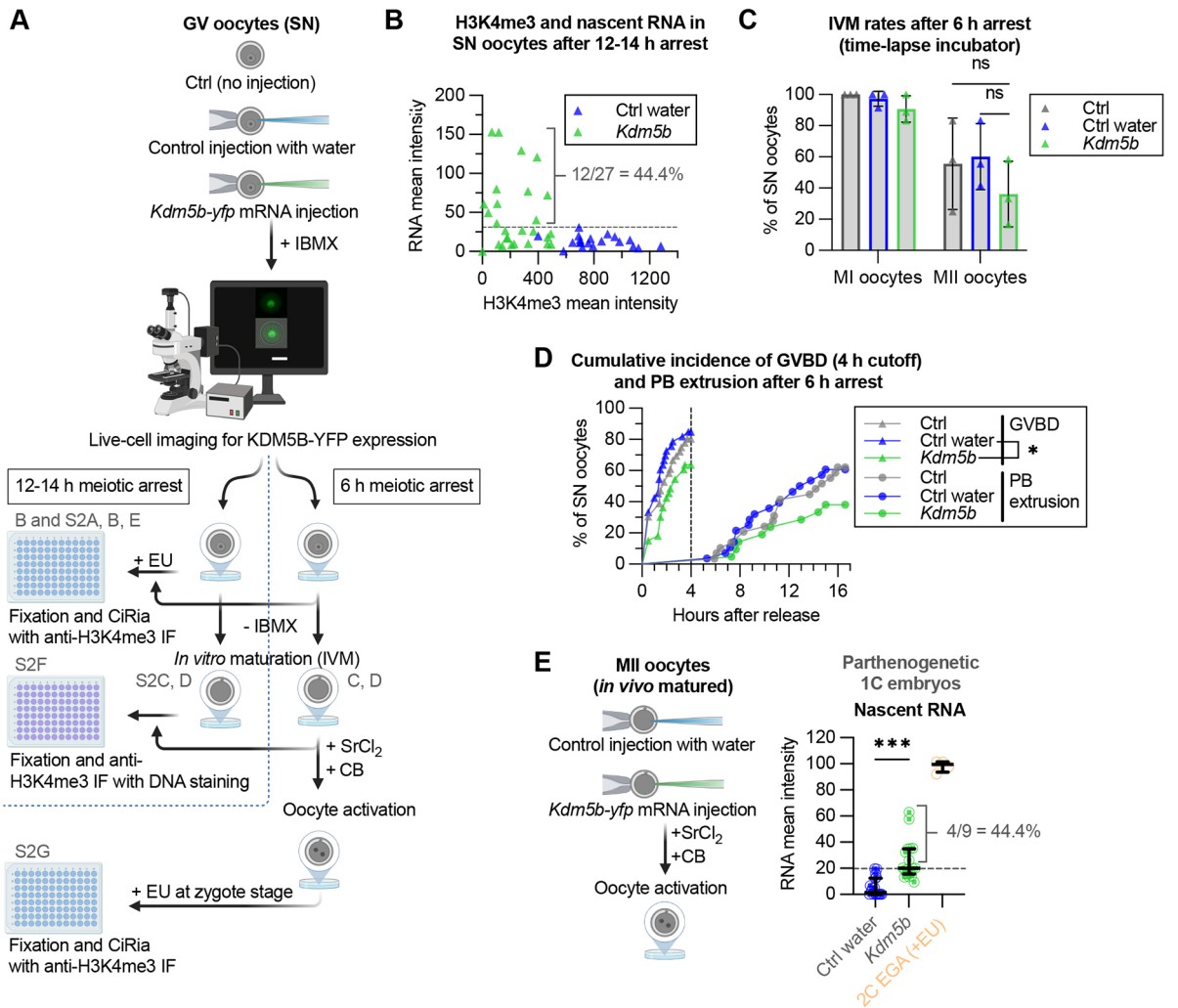

**Fig. 1. Broad H3K4me3 is involved in oocyte genome silencing, maturation and transfer of transcriptional repression to the zygote.**
(A) Experimental workflow: surrounded nucleolus (SN)-stage GV oocytes were injected with *Kdm5b-yfp* mRNA; uninjected and water-injected oocytes served as controls. The C-terminal YFP tag enabled live-cell confirmation of KDM5B expression. Scale bar: 50 µm. Oocytes were arrested in meiosis with IBMX for 12-14 h or 6 h post-injection. Following arrest, subsets of oocytes were: (1) EU labelled for 2 h, fixed and processed for CiRia with anti-H3K4me3 immunofluorescence; (2) washed free of IBMX and matured by *in vitro* maturation (IVM) for ~15-16 h, with MII oocytes stained for H3K4me3 and DNA; (3) chemically activated with strontium chloride ($SrCl_2$) and cytochalasin B (CB) – 6 h arrest group only – and either (3a) EU labelled for 4 h, fixed 11 h after activation and processed for CiRia with anti-H3K4me3 IF, or (3b) cultured to day 5 for developmental assessment. Created in BioRender. Indahl, M. (2025) https://BioRender.com/infz8oe. (B) H3K4me3 signal intensity (*x*-axis) versus nascent RNA signal intensity (*y*-axis) in nuclei of SN-stage oocytes after a 12-14 h meiotic arrest and 2 h EU labelling. Total number of oocytes (summed across two biological replicates): Ctrl water, *n*=20; *Kdm5b*, *n*=27 (subset of S2A dataset). No correlation was found (Spearman correlation). See also Fig. S2A,B. (C) Maturation after 15.4-16.5 h IVM following 6 h meiotic arrest. Each triangle represents one biological replicate. Data are mean±s.d., showing mean percentage reaching MI and MII stages. Total number of SN oocytes (summed across three replicates): Ctrl, *n*=36; Ctrl water, *n*=33; *Kdm5b*, *n*=33. Pairwise comparisons by Fisher's exact test on pooled counts; significance indicated by *P*-value notation. (D) Cumulative incidence plot of germinal vesicle breakdown (GVBD) and polar body (PB) extrusion over time following a 6 h meiotic arrest. Only oocytes from C that underwent GVBD within 4 h of arrest release (dotted line) are included. Total number of oocytes (summed across three biological replicates): Ctrl MI/MII, *n*=29/18; Ctrl water MI/MII, *n*=28/17; *Kdm5b* MI/MII, *n*=21/8. Median time to GVBD: Ctrl=1.8 h, Ctrl water=1.4 h and *Kdm5b*=2.7 h. Median time to PB extrusion: Ctrl=14.7 h, Ctrl water=13.4 h and *Kdm5b* undefined. Pairwise comparison by log-rank (Mantel–Cox) tests with Holm–Šídák correction for multiple comparisons (alpha=0.05); adjusted significance indicated by *P*-value notation. (E) Left: shortened experimental workflow for injection and chemical activation of *in vivo* matured MII oocytes. Activated oocytes were EU labelled for 2 h, fixed for 9-10 h after activation and stained (not included in workflow schematic). Created in BioRender. Indahl, M. (2025) https://biorender.com/zq6cibi. The full workflow is in Fig. S2H. Right: nascent RNA signal intensity in parthenogenetic 1C embryos, with 2C embryos at mid-embryonic genome activation (EGA) (orange) as a reference. Total number of samples (summed across two biological replicates): Ctrl water, *n*=9; *Kdm5b*, *n*=9; 2C EGA (+EU), *n*=2. Data are median and IQR. Pre-selected pair was compared with a Mann–Whitney test; significance indicated by *P*-value notation. See also Fig. S2I. ns, *P*>0.05; **P*≤0.05; ***P*≤0.01; ****P*≤0.001.

In summary, our results confirm a repressive role for broad H3K4me3 domains in SN-stage oocytes and suggest a potential role for these domains in transferring repression to 1C embryos. Further, broad H3K4me3 domains may be involved in acquisition of full meiotic competence, as induced H3K4me3 removal perturbs oocyte maturation timing.

## EGA occurs after onset of H3K4me3 demethylation but prior to full reprogramming

It has previously been shown that by the late 2C stage of mouse development, broad H3K4me3 domains that were established during oogenesis are lost, with only a narrow promoter-specific pattern remaining around the time of EGA (Dahl et al., 2016; Zhang

et al., 2016). However, the exact sequence of events and dynamics of H3K4me3 reprogramming in relation to transcriptional activity during the first and second cell cycles of embryonic development has not been determined.

To elucidate the relationship between H3K4me3 and genome activation in the zygote and 2C embryo, we performed high temporal resolution analysis of global transcription and H3K4me3 by EU labelling and CiRia coupled with immunofluorescence staining. Pronuclei exhibited asymmetric H3K4me3 staining, with higher levels in maternal PNs (Fig. 2A, top left, and Fig. 2B), as previously reported (Andreu-Vieyra et al., 2010; Lepikhov and Walter, 2004). Transcription levels during the 2 h EU labelling were low at the reported time of minor genome activation (Aoki et al., 1997), reflecting transcription at ∼20% of major EGA levels (Fig. 2A, bottom). In 2C embryos, H3K4me3 was lost at a steady linear rate throughout the second cell cycle (Fig. 2A, top right; Fig. 2B). By contrast, nascent RNA levels were low during the early 2C stage, before exhibiting a steep increase 9 h into the second embryonic cell cycle, corresponding to G2 phase (Fig. 2A, bottom right; Fig. 2B). By late 2C stage, approximately halfway through the prolonged G2 phase, transcription levels began decreasing, prior to entry into M phase (Fig. 2A, bottom right).

H3K4 demethylases KDM5A and KDM5B have been shown to reduce and narrow H3K4me3 domains in the early embryo, and are required for embryo development (Dahl et al., 2016; Liu et al., 2016). The early initiation of H3K4me3 demethylation in the 2C embryo, observed here (Fig. 2A, top right), suggested a dependence on *Kdm5a* and *Kdm5b* transcription in the late zygote or early 2C embryo. To determine the exact timing of transcription and whether the H3K4 demethylase mRNAs originated from the minor genome activation, we quantified transcript levels in late zygotes and early 2C embryos by RNA-seq. We found that low levels of *Kdm5a* and *Kdm5b* mRNA were present in late zygotes (17 h after ovulation), followed by a substantial increase in *Kdm5b*, but not *Kdm5a*, transcripts at the early 2C stage (5.5-8 h after division) prior to EGA (Fig. 2C). This suggests that *Kdm5b* transcription in the early 2C embryo, after minor genome activation, initiates H3K4me3 removal before EGA onset. Consistent with this, previous studies have shown minimal *Kdm5b* expression in MII oocytes, as well as RNA polymerase II (RNAPII) binding at the *Kdm5b* locus and protein detection in 2C embryos, supporting transcriptional activation, rather than maternal mRNA stabilisation, as the source of KDM5B after fertilisation (Dahl et al., 2016; Dang et al., 2022; Liu et al., 2020; Shao et al., 2014).

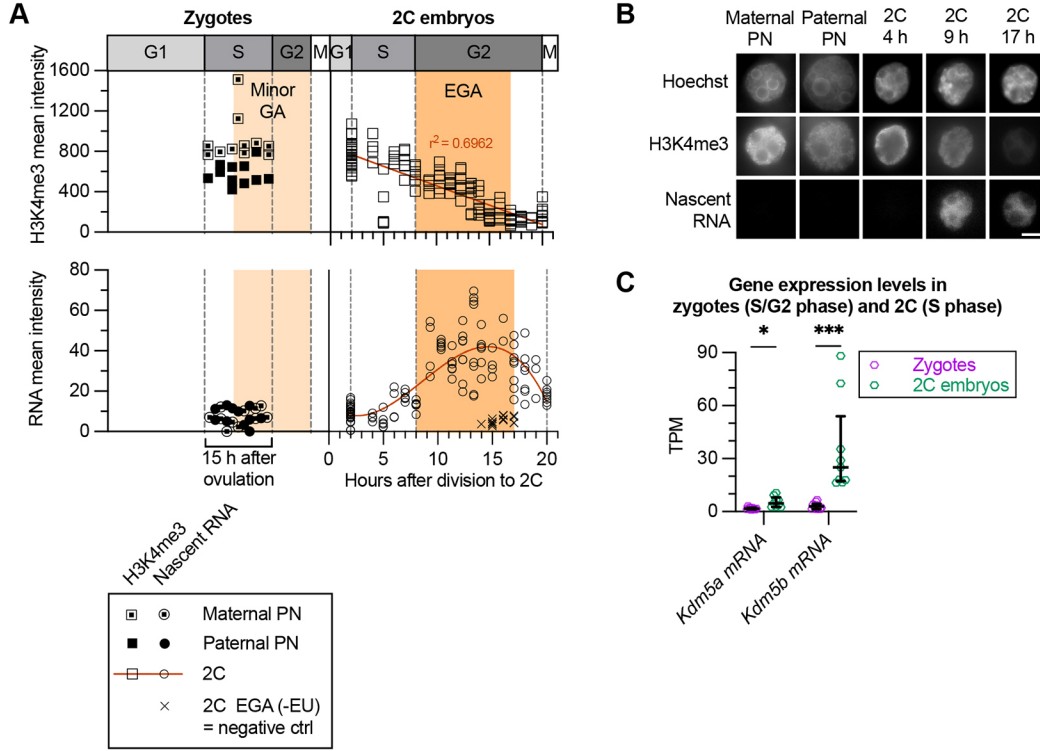

**Fig. 2. Embryonic genome activation occurs after onset of H3K4me3 demethylation but before full reprogramming.** (A) H3K4me3 (top) and nascent RNA (bottom) signal intensity of *in vivo*-fertilised mouse zygotes and 2C embryos (all uninjected) that were EU labelled for 2 h at multiple timepoints, followed by CiRia with anti-H3K4me3 immunofluorescence staining. Each point represents an individual pronucleus (PN) or 2C nucleus. Total number of samples (summed across two biological replicates): zygotes, *n*=11; 2C embryos H3K4me3, *n*=68.5; RNA, *n*=60.5; 2C embryonic genome activation (EGA) (−EU), *n*=8. To aid visualisation and given the uncertainty in timing of fertilisation, zygote points were scattered. For 2C embryos, the H3K4me3 line shows simple linear regression; RNA curve fitted with centred third order polynomial model. Timing of cell cycle phases (G1, S, G2 and M), minor genome activation (GA) and EGA were based on previous studies (Aoki et al., 1997; Artus and Cohen-Tannoudji, 2008; Bouniol et al., 1995). (B) Representative fluorescence microscopy images from A, showing maternal PN, paternal PN and nuclei of 2C embryos at 4 h (S phase), 9 h (early G2) and 17 h (late G2) after division. Panels show DNA (Hoechst 33342), H3K4me3 and nascent RNA, with brightness and contrast adjusted consistently across all images. Scale bar: 10 μm. (C) Gene expression levels (transcripts per million, TPM) of *Kdm5a* and *Kdm5b* from RNA-seq (see Fig. 4A) of control late zygotes (*n*=10) and early 2C embryos (*n*=9). Samples were collected for RNA-seq ∼5-6 h post-injection (17 h after ovulation, late S/G2 phase, for zygotes; 5.5-8 h after division, late S phase, for 2C embryos). Data are median and IQR. Pre-selected pairs were compared using a Kruskal–Wallis test with Dunn's multiple comparisons test; adjusted significance is indicated by *P*-value notation: ns, *P*>0.05; **P*≤0.05; ***P*≤0.01; ****P*≤0.001.

Taken together, initiation of H3K4me3 demethylation precedes EGA in the early 2C embryo and is associated with upregulation of *Kdm5b*. H3K4me3 demethylation continues in a linear fashion, whereas a step-change in transcription occurs before full removal of broad H3K4me3 domains.

### Early induced removal of broad H3K4me3 domains in zygotes and early 2C embryos does not trigger transcriptional reactivation

Based on studies showing that active removal of broad H3K4me3 domains is necessary for normal EGA, and findings that overexpression of KDM5B can bring about transcriptional reactivation in oocytes (Dahl et al., 2016; Zhang et al., 2016), we investigated whether demethylation of broad H3K4me3 is sufficient to trigger premature EGA in mouse zygotes. We injected zygotes, at a time corresponding to early S phase, with *Kdm5b-yfp* mRNA or a catalytic domain deleted version, Del *Kdm5b-yfp* mRNA. Zygotes were screened for expression of the fusion protein, cultured and fixed at the estimated time of minor genome activation (18 h after ovulation, late S/early G2 phase) – a stage known to be conducive with increased transcription (Aoki et al., 1997; Bouniol et al., 1995). EU labelling was initiated 2 h prior to fixation, and zygotes were then analysed by CiRia coupled with immunofluorescence staining (Fig. 3A, left side). A substantial reduction in H3K4me3 levels, below those found in 2C embryos at EGA, was observed in *Kdm5b-yfp* mRNA-injected zygotes at the time of late S/G2 phase, compared to Del *Kdm5b-yfp* injected and uninjected controls (Fig. 3B, left; Fig. S3A). Furthermore, reduction in H3K4me3 was apparent in both the maternal and paternal pronuclei, with both reaching similar levels, despite an elevated starting point in the maternal PN. However, even with reduced H3K4me3, the level of newly synthesised transcripts was similar to controls in both pronuclei of the *Kdm5b-yfp* mRNA-injected zygotes (Fig. 3B, right).

A stepwise model of gene activation in early embryos has been proposed, where EGA is dependent on proteins originating from transcripts produced during minor genome activation (Hamatani et al., 2004). Moreover, a recent study demonstrated that translation of several ribosomal and transcription-related proteins is reactivated in the early 2C embryo prior to EGA initiation (Xiong et al., 2022). We considered the possibility that there was insufficient time following experimental H3K4me3 removal for accumulation and assembly of the transcriptional machinery required for large-scale genome activation. We therefore cultured injected zygotes on to the early 2C stage, corresponding to G1 or S phase and prior to the normal timing of EGA. In common with the zygote stage, a strong reduction in H3K4me3 levels was observed (Fig. 3C, left), but this did not give rise to elevated global transcription (Fig. 3C, right; Fig. S3A).

Several studies have shown that a subset of proteins encoded by minor genome activation transcripts are not synthesised until after the first division (Flach et al., 1982; Latham et al., 1991; Nothias et al., 1996). With this in mind, we allowed minor genome activation to proceed unperturbed, and set out to test whether *Kdm5b-yfp* mRNA injection ~1 h after division to 2C could provoke a precocious EGA. In order to reduce the effect of inter-sample variation, we injected only one of the blastomeres with either *Kdm5b-yfp* mRNA or Del *Kdm5b-yfp* mRNA, leaving the other blastomere as an uninjected internal control (Fig. 3A, right side). H3K4me3 and nascent RNA staining showed similar results to other stages, namely considerable reduction in H3K4me3 levels following KDM5B overexpression (Fig. 3D, left), but no discernible difference in transcriptional activity between control and injected blastomeres (Fig. 3D, right; Fig. S3B).

To investigate a potential *Kdm5b-yfp* dosage effect, we plotted H3K4me3 against RNA intensity for all experimental groups (Fig. S3C-E). However, we did not observe any negative correlations between the H3K4me3 level and global RNA transcription in any of the stages or groups that might be indicative of a dosage effect. Nevertheless, high H3K4me3 levels were incompatible with high levels of transcription in all groups (Fig. S3C-E). Interestingly, early 2C embryos demonstrated a greater heterogeneity in transcriptional activity than zygotes (Fig. S3C-E), with blastomeres from the same embryo more closely correlated than within the same experimental group (Fig. 3D, right).

Overall, H3K4me3 levels were consistently reduced across oocyte, zygote and embryo stages studied by an average of 51-71% after overexpression of KDM5B (Fig. 3E, left). This is similar to the average 51% H3K4me3 reduction seen in unperturbed 2C embryos, from early 2C to mid-EGA (Fig. 2A, top right). However, timing of induced H3K4me3 reduction had a stage-dependent effect on transcription (Fig. 3E, right). H3K4me3 depletion in SN and MII stage oocytes (Fig. S2A,I) resulted in 3.6 and 4.7 times increases in nascent RNA compared to controls, respectively (Fig. 3E, right). Surprisingly, H3K4me3 depletion in zygotes and early 2C embryos did not give rise to increased transcription (Fig. 3E, right), suggesting that maintenance of broad H3K4me3 domains in the late zygote and early 2C embryo is not required for continued genome silencing, and, as a result, removal of these domains does not lead to precocious genome activation.

### Broad H3K4me3 removal does not substantially alter expression of genes or transposons in zygotes and early 2C embryos

Whilst we found that bulk transcription levels were unchanged following reduction of H3K4me3 in zygotes and 2C embryos, we could not rule out an overall neutral up- and downregulation of maternal and embryonic genes or transposons. We therefore employed RNA-seq on single zygotes and early 2C embryos that had been injected with either *Kdm5b-yfp* mRNA or water (Fig. 4A). Zygotes and 2C embryos were collected for RNA-seq 5-6 h post-injection to allow sufficient time for H3K4me3 demethylation and potential genome activation, but prior to the normal time of EGA. As previously, this time corresponded to late S or G2 phase for zygotes (17 h after ovulation) and S phase for 2C embryos (5.5-8 h after division).

RNA-seq results showed few genes with a statistically significant differential expression (DE) between *Kdm5b-yfp* injected and water-injected controls at zygote and 2C embryo stage, with only two upregulated transcripts in *Kdm5b-yfp* injected zygotes (Fig. 4B) and no DE transcripts between the early 2C groups (Fig. 4C). In order to validate the sensitivity of our single embryo RNA-seq experiment, we compared control zygotes to control early 2C embryos. We found that control early 2C embryos had 57 upregulated and 237 downregulated genes compared to control zygotes (Fig. 4D), consistent with maternal transcript clearance prior to EGA. The top three most significant hits for upregulated genes (*Zscan4a*, *Psme4* and *Tmem132c*) have previously been shown to be expressed in early embryos (Falco et al., 2007; Huang et al., 2017), and the top three most significant hits for downregulated genes (*Tacc3*, *Txnip* and *Pdcd5*) are known oocyte-expressed transcripts that are degraded after fertilisation (Dobson et al., 2004; Evsikov et al., 2004; Hao et al., 2002; Lee et al., 2013). The paucity of upregulated genes in the early 2C was consistent with our bulk transcription data for this stage, which showed low levels of nascent RNA (Fig. 2A, bottom right).

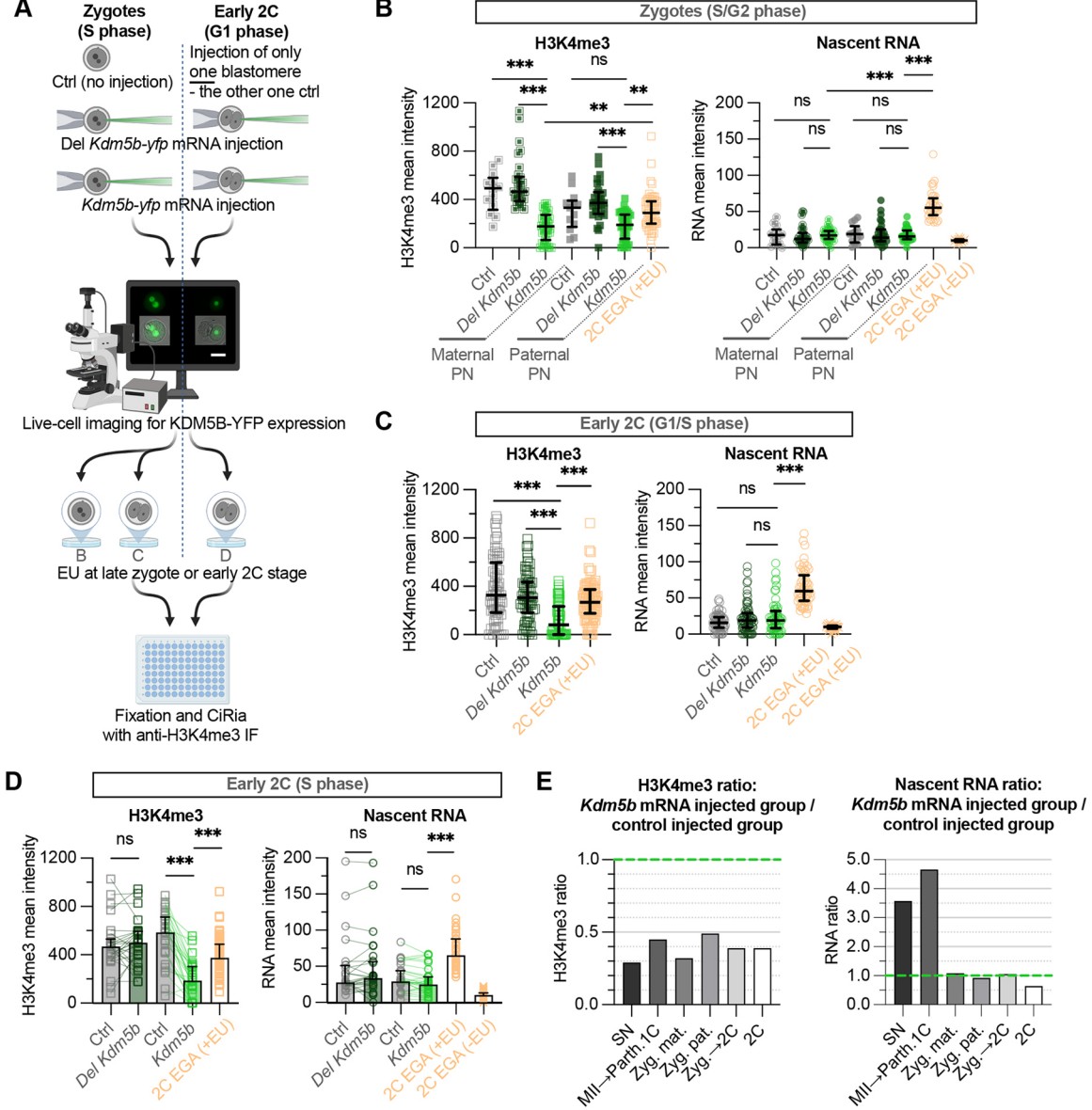

**Fig. 3. Early induced removal of broad H3K4me3 domains in zygotes and early 2C embryos does not trigger transcriptional reactivation.**
(A) Experimental workflow: S-phase zygotes or one blastomere of G1-phase 2C embryos were injected with *Kdm5b-yfp* mRNA; uninjected zygotes/2C and zygotes/2C blastomeres injected with mRNA encoding a catalytically inactive KDM5B (Del *Kdm5b-yfp*) served as controls. The C-terminal YFP tag enabled live-cell confirmation of KDM5B expression. Scale bar: 50 µm. Zygotes and 2C embryos (with corresponding figure panel letters) were EU labelled for 2 h and processed for CiRia with anti-H3K4me3 immunofluorescence. Uninjected 2C embryos cultured to mid-embryonic genome activation (EGA) were included in each experiment as a reference (not included in workflow schematic). Created in BioRender. Indahl, M. (2025) https://biorender.com/e9e50of. (B) H3K4me3 and nascent RNA signal intensity in maternal and paternal PNs of zygotes, with 2C embryos at mid-EGA (orange) as a reference. Total number of samples (summed across two biological replicates): Ctrl, *n*=13; Del *Kdm5b*, *n*=30; *Kdm5b*, *n*=34; H3K4me3 2C EGA (+EU), *n*=29.5; RNA 2C EGA (+EU), *n*=19; 2C EGA (−EU), *n*=10.5. Zygotes fixed ~4-6 h post-injection. Data are median and IQR. Pre-selected pairs were compared using a Kruskal–Wallis test with Dunn's multiple comparisons test; adjusted significance indicated by *P*-value notation. See also Fig. S3A,C. (C) H3K4me3 and nascent RNA signal intensity in nuclei of early 2C embryos injected at zygote stage, with 2C embryos at mid-EGA (orange) as a reference. Total number of samples (summed across three biological replicates): Ctrl, *n*=34; Del, *Kdm5b*, *n*=34.5; *Kdm5b*, *n*=30; H3K4me3 2C EGA (+EU), *n*=44.5; RNA 2C EGA (+EU), *n*=28; 2C EGA (−EU), *n*=16.5. Zygotes were cultured to early 2C and fixed ~1-8 h after division (6-10.5 h post-injection). Data are median and IQR. Pre-selected pairs were compared using a Kruskal–Wallis test with Dunn's multiple comparisons test; adjusted significance indicated by *P*-value notation. See also Fig. S3A,D. (D) H3K4me3 and nascent RNA signal intensity in nuclei of early 2C embryos in which one blastomere was injected at the start of 2C stage, with 2C embryos at mid-EGA (orange) as reference. Total number of samples (summed across two biological replicates): Ctrl (uninjected sister blastomeres to Del *Kdm5b*) and Del *Kdm5b*, *n*=27; Ctrl (uninjected sister blastomeres to *Kdm5b*) and *Kdm5b*, *n*=28; H3K4me3 2C EGA (+EU), *n*=36; RNA 2C EGA (+EU), *n*=21; 2C EGA (−EU), *n*=15. Embryos were fixed ~4-6 h after division (3-4.5 h post-injection). Data are median and IQR. Uninjected and injected sister blastomeres were compared using Wilcoxon matched-pairs signed-rank tests (blastomere pairs shown by green lines). Comparisons to the 2C EGA (+EU) group were made using a Mann–Whitney test with Holm–Šídák correction (alpha=0.05). Adjusted significance indicated by *P*-value notation. See also Fig. S3B,E. (E) Ratio of H3K4me3 and RNA signal intensity at oocyte and early embryo stages, calculated as the mean signal in *Kdm5b* mRNA-injected samples divided by the corresponding control-injected group (Ctrl water or Del *Kdm5b*). Data include: SN (Fig. S2A), MII→Parth. 1C (H3K4me3, Fig. S2I), MII→ Parth. 1C (RNA, Fig. 1E), Zyg. mat./pat. (Fig. 3B), Zyg.→2C (Fig. 3C) and 2C (Fig. 3D). Stages on the *x*-axis indicate the culture period; if two stages are listed, the first indicates the injection stage. *P*-value notation: ns, *P*>0.05; **P*≤0.05; ***P*≤0.01; ****P*≤0.001.

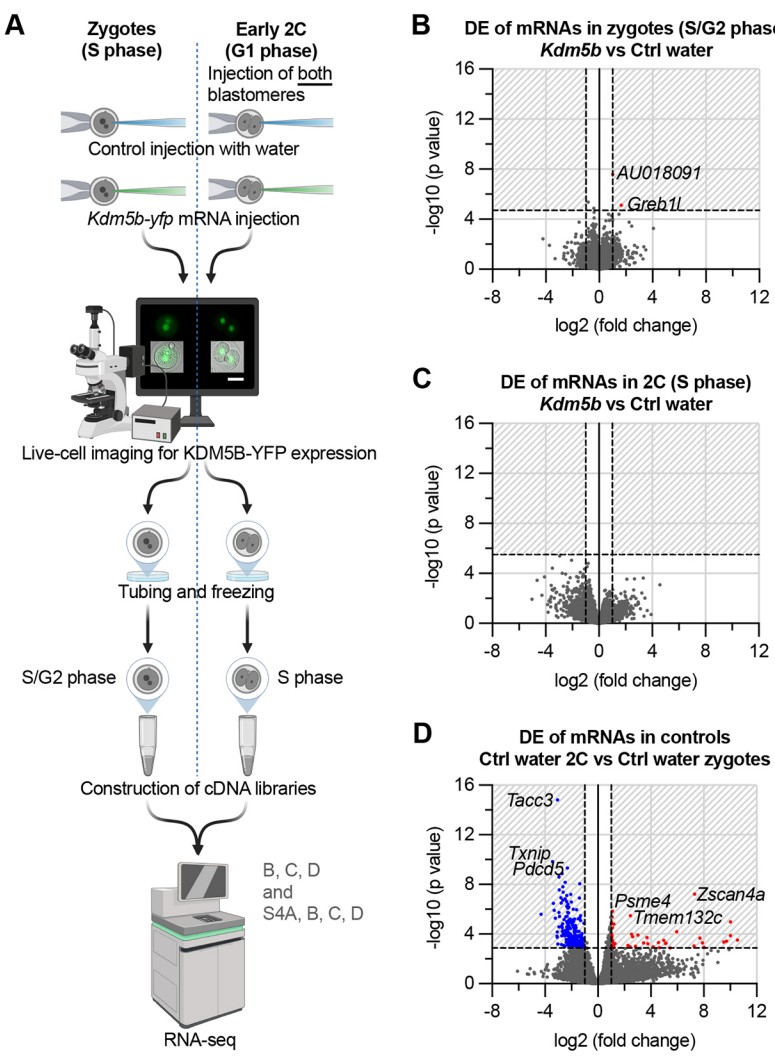

**Fig. 4. Broad H3K4me3 removal does not substantially alter expression of genes in zygotes and early 2C embryos.** (A) Experimental workflow: S-phase zygotes or both blastomeres of G1-phase 2C embryos were injected with *Kdm5b-yfp* mRNA; water-injected zygotes or 2C embryos served as controls. The C-terminal YFP tag enabled live-cell confirmation of KDM5B expression. Scale bar: 50 µm. Samples were collected for RNA-seq ~5-6 h post-injection (zygotes: 17 h after ovulation, late S/G2 phase; 2C: 5.5-8 h after division, late S phase). See also Fig. S4. Created in BioRender. Indahl, M. (2025) https://biorender.com/o3gape4. (B) RNA-seq volcano plot showing differential expression (DE) of mRNA transcripts in *Kdm5b*-injected zygotes (*n*=10) versus Ctrl water zygotes (*n*=10). The two significantly upregulated transcripts are labelled by their gene symbol. *Kdm5b*-assigned reads were removed prior to analysis. (C) RNA-seq volcano plot showing DE of mRNA transcripts in *Kdm5b*-injected 2C (*n*=6) versus Ctrl water 2C (*n*=9). No significantly up- or downregulated genes were identified. *Kdm5b*-assigned reads were removed prior to analysis. (D) RNA-seq volcano plot showing DE of mRNA transcripts in Ctrl water 2C (*n*=9) versus *Kdm5b*-injected zygotes (*n*=10). The three most significantly up- and downregulated transcripts are labelled by gene symbol. For all volcano plots: DE was assessed using the DESeq2 R package. Genes with adjusted *P*<0.05 and log2(fold change)>1 or <−1 are shown as red (upregulated) or blue (downregulated) points. All coloured points fall within the shaded region denoting the significance threshold.

Principal component analysis (PCA) of RNA-seq data from *Kdm5b-yfp* and control-injected zygotes and early 2C embryos confirms that most gene expression variance is accounted for by developmental stage rather than KDM5B overexpression (Fig. S4A).

Next, we examined transposable elements (TEs), as their expression at the zygote/2C stage has been shown to be important for onward development (Jachowicz et al., 2017; Kigami et al., 2003; Peaston et al., 2004), and because H3K4me3 is frequently detected at TEs in embryos (Zhang et al., 2016). The analysis showed a similar picture to mRNAs, with no TEs being differentially expressed following KDM5B overexpression in zygotes or early 2C embryos (Fig. S4B, C), although several TEs exhibited up- or downregulation during the transition from the zygote to 2C stage in controls (Fig. S4D).

Overall, wide-scale removal of broad H3K4me3 at mid-zygote and early 2C embryo stage does not alter expression of genes or TEs at the late zygote or early 2C stage, respectively. In contrast to what we observed in oocytes, this suggests that genome silencing is not dependent on broad H3K4me3 domains in the late zygote or early preimplantation embryo.

## Premature removal of broad H3K4me3 does not disrupt EGA or impair preimplantation embryo development

Finally, having found that induced removal of broad H3K4me3 does not lead to reactivation of the genome in the zygote or early 2C embryo, we asked whether this premature removal might impair

the normal progress of EGA during the G2 phase of the 2C stage, as well as subsequent preimplantation development. We repeated injections of early S phase zygotes with *Kdm5b-yfp* or Del *Kdm5b-yfp* mRNA and assessed either H3K4me3 and nascent transcript levels at mid-EGA of the 2C stage or onward developmental progress (for schematic, see Fig. S5A). H3K4me3 levels were similar in *Kdm5b-yfp* mRNA-injected, Del *Kdm5b-yfp* mRNA-injected and uninjected controls at the timing of 2C EGA (Fig. S5B). Nascent RNA was also detected at similar levels in all groups (Fig. 5A), indicating that early removal of H3K4me3 does not affect overall transcriptional activity during EGA.

Developmental progression was equivalent between *Kdm5b-yfp* mRNA-injected zygotes and uninjected controls (Fig. 5B), and the number of cells in resulting blastocysts was also comparable (Fig. 5C and Fig. S5C). In addition, embryos subject to KDM5B overexpression had a similar proportion of developmentally advanced blastocysts (Fig. 5D). However, embryos in the Del *Kdm5b* control group had a greater tendency for arresting before reaching the blastocyst stage (Fig. 5B). Nevertheless, those Del *Kdm5b* embryos that did reach blastocyst stage, had a comparable number of cells to the other groups, although they had a trend towards more epiblast cells (Fig. 5C and Fig. S5C). Whilst not statistically significant, there was also a higher percentage of blastocysts that had not yet expanded in the Del *Kdm5b* group (Fig. 5D). We suspect that Del KDM5B proteins may persist after binding to broad H3K4me3 domains due to

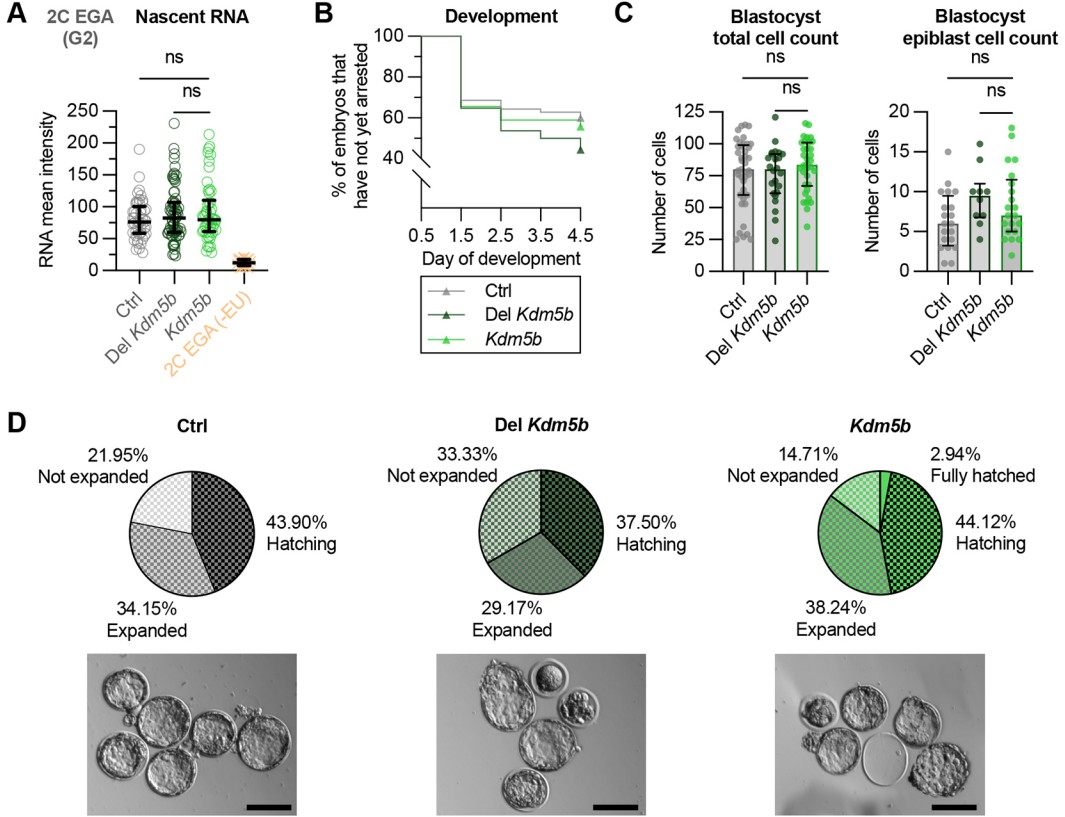

**Fig. 5. Premature removal of broad H3K4me3 does not disrupt embryonic genome activation or impair preimplantation embryo development.** (A) Nascent RNA signal intensity in nuclei of 2C embryos at mid-embryonic genome activation (EGA) following injection at zygote stage, with 2C embryos not treated with EU (orange) also included. Total number of samples (summed across three biological replicates): Ctrl, *n*=24.5; Del *Kdm5b*, *n*=35; *Kdm5b*, *n*=25; 2C EGA (−EU), *n*=12. Embryos were fixed on average 15 h after division (range: 10-20 h, corresponding to ~20-23 h post-injection). Data are median and IQR. Pre-selected pairs were compared using a Kruskal–Wallis test with Dunn's multiple comparisons test; adjusted significance indicated by *P*-value notation. See also Fig. S5A,B. (B) Kaplan–Meier plot showing the percentage of uninjected (Ctrl) and injected (Del *Kdm5b-yfp* or *Kdm5b-yfp* mRNA) zygotes that had not yet arrested on each day of development. Total number of zygotes (summed across three biological replicates): Ctrl, *n*=70; Del *Kdm5b*, *n*=54; *Kdm5b*, *n*=61. No statistically significant differences were detected by pairwise log-rank (Mantel–Cox) tests with Holm–Šídák correction (alpha=0.05). (C) Total and epiblast cell count of day 4.5 blastocysts from the experiment in B. Total number of blastocysts (summed across three biological replicates): Ctrl, *n*=41; Del *Kdm5b*, *n*=24; *Kdm5b*, *n*=34. Epiblast cell counts were performed in two replicates: Ctrl, *n*=20; Del *Kdm5b*, *n*=10; *Kdm5b*, *n*=21. Data are median and IQR. Pre-selected pairs were compared using a Kruskal–Wallis test with Dunn's multiple comparisons test; adjusted significance is indicated by *P*-value notation. See also Fig. S5C. (D) Pie charts showing developmental progression of day 4.5 blastocysts from the experiment in B. Representative bright-field images are shown for each group. Scale bars: 100 µm. *P*-value notation: ns, *P*>0.05; **P*≤0.05; ***P*≤0.01; ****P*≤0.001.

the absence of catalytic activity. This may interfere with timely H3K4me3 reprogramming by endogenous KDM5B and affect developmental progression, as previously shown (Dahl et al., 2016; Liu et al., 2016). Taken together, these results show that early removal of broad H3K4me3 domains does not affect transcription levels during EGA or perturb developmental progression during the preimplantation period.

## DISCUSSION
Several approaches have been applied to investigate the role of broad H3K4me3 in oogenesis and preimplantation embryo development. Conditional knockout of H3K4 methyltransferase, KMT2B, in oocytes during early folliculogenesis has been shown to prevent establishment of broad H3K4me3 domains, impede transcriptional silencing, reduce oocyte survival, impair maturation, severely reduce ovulation and result in embryo arrest (Andreu-Vieyra et al., 2010; Hanna et al., 2018). Knockdown of KDM5B in 2C embryos prevents H3K4me3 demethylation, dysregulates EGA and severely impairs embryo development (Dahl et al., 2016; Liu et al., 2016). However, the role of this unique broad H3K4me3 profile during oocyte maturation and early preimplantation embryo development is not well understood. By quantifying H3K4me3 and nascent RNA in oocytes, zygotes and embryos, in combination with KDM5B overexpression, we were able to examine the ongoing role of broad H3K4me3 in transcriptional repression and early development. We found that broad H3K4me3 maintains silencing in oocytes and is necessary for timely oocyte maturation, and may also play a role in transferring transcriptional repression to the early 1C embryo. Further, we show that H3K4me3 demethylation is initiated before EGA in the early 2C embryo, and in the presence of increased *Kdm5b* expression. However, we were surprised to find that genome silencing in the late zygote and early preimplantation embryo is not dependent on broad H3K4me3, and premature removal does not impair EGA or early preimplantation development. Similar findings were independently reported in a recent study, which showed that experimental depletion of H3K4me3, by either KDM5B overexpression or KMT2B knockdown, did not markedly alter transcriptional activity or the expression of EGA genes in 2C embryos (Zhang et al., 2025). Taken together, our findings suggest an enigmatic role for broad H3K4me3 domains, where their contribution to genome silencing changes during the course of oogenesis and early preimplantation embryo development.

The surprising lack of transcriptional upregulation in zygotes and pre-EGA embryos following removal of broad H3K4me3 domains may have several explanations. Previous studies have shown that removal of H3K4me3 broad domains in embryos is dependent on transcription (Abe et al., 2018; Zhang et al., 2016), suggesting that initiation of EGA may precede broad H3K4me3 removal, e.g. through H3K36me3 recruitment of KDM5B, as shown in mouse ESCs (Xie et al., 2011). Indeed, there is a wider debate over the degree to which epigenetic marks direct transcription, and whether marks are merely consequential or supportive (Gardner et al., 2011; Henikoff and Shilatifard, 2011; Howe et al., 2017; Millán-Zambrano et al., 2022; Wang et al., 2022a). However, our data support previous findings showing a repressive role for broad H3K4me3 and a requirement for at least a partial removal of these domains by KDM5B prior to EGA (Dahl et al., 2016; Liu et al., 2016; Zhang et al., 2016). Firstly, we observed initiation of H3K4me3 demethylation before EGA in the early 2C embryo. Secondly, our RNA-seq data shows that *Kdm5b* transcripts already show significantly increased expression in the early 2C embryo. Thirdly, overexpression of KDM5B gives rise to a rapid and widescale depletion of H3K4me3 within a few hours of mRNA injection, and in the absence of, or prior to, increased transcription. Finally, SN oocytes undergoing H3K4me3 depletion display a lag of several hours before increased transcription is detected, evident from our observation of reactivation after 12-14 h of arrest but not after 6 h. Together, this suggests that removal of broad H3K4me3 domains is not dependent on transcription per se but rather on the expression of *Kdm5b* in the early 2C embryo, and therefore supports an instructive role for this epigenetic mark.

An alternative explanation for the lack of transcriptional reactivation in late zygotes and early 2C embryos may be insufficient chromatin accessibility due to transient redundant or compensatory repressive marks. Potential compensatory repressive epigenetic marks following experimental broad H3K4me3 removal include DNA methylation (Ooi et al., 2007), which anti-correlates with broad H3K4me3 in oocytes (Dahl et al., 2016; Zhang et al., 2016), and H3K9me3, whose invasion into H3K4me3 regions during oogenesis is prevented by KDM4A (Sankar et al., 2020). However, assay for transposase-accessible chromatin with sequencing (ATAC-seq) data suggest that late zygotes and early 2C embryos have globally permissive chromatin (Wu et al., 2016). Furthermore, H3K27ac enrichment and poised RNAPII at EGA genes is found in late zygotes and early 2C embryos, with similar profiles apparent in early and late 2C stage, thus fulfilling a number of requirements for transcriptional activation (Liu et al., 2020; Wang et al., 2022b). Nevertheless, higher order genome structures may be insufficiently developed at these stages, or affected by H3K4me3 depletion. Indeed, topologically associating domains (TADs) and lamina-associated domains (LADs) form after fertilisation (Du et al., 2017; Borsos et al., 2019), and LAD establishment in the paternal PN appears to be dependent on newly acquired H3K4me3 (Borsos et al., 2019). Taken together, the mechanisms that maintain genome silencing in late zygotes and early 2C embryos in the absence of broad H3K4me3 remain elusive.

Regarding study limitations, we have not identified the specific transcripts produced as a result of transcriptional reactivation in SN oocytes. Of note, our quantification of H3K4me3 signal intensity did not account for differences in chromatin configuration between NSN and SN oocytes. Furthermore, while our experimental system provided high temporal resolution of H3K4me3 and global transcription during early embryo development, it did not allow us to determine which genomic regions were depleted of H3K4me3 following KDM5B overexpression. Although Zhang et al. (2025)

defined broad H3K4me3, referred to as non-canonical, differently when compared to our previous study (21,619 peaks ≥5 kb vs 63,542 domains >10 kb in Dahl et al., 2016), their Cleavage Under Targets and Tagmentation (CUT&Tag) analysis of early 2C embryos confirmed that KDM5B overexpression from the zygote stage resulted in a marked reduction in non-canonical H3K4me3, without inducing premature establishment of a narrow, canonical H3K4me3 pattern. While KDM5B activity has been shown to be important for embryo development, it has remained unclear whether removal of broad domains or reprogramming of promoters to a narrow H3K4me3 pattern is the critical function (Liu et al., 2016; Xu and Xie, 2018). Considered alongside the Zhang et al. (2025) study, our findings point toward a more essential role for the establishment of narrow peaks in the initiation of EGA.

Additionally, some of our expression data, particularly in immature and activated oocytes, were limited to bulk transcription monitoring rather than transcript-level analysis. Due to technical constraints, we were unable to integrate transcriptional assessment with developmental outcome in the same oocyte. Further work is needed to clarify whether genome reactivation at SN stage contributes to the impaired maturation timing. Finally, we cannot rule out the possibility that overexpression of KDM5B may have led to some non-specific loss of promoter-associated H3K4me3, usually associated with active transcription. However, the normal EGA and development observed after KDM5B overexpression suggests that this was not the case.

Interestingly, broad H3K4me3 establishment in oocytes was recently found to be interdependent with incorporation of the histone variant H2A.Z (Mei et al., 2025). Such interdependence of histone marks and variants complicates mechanistic interpretation, as perturbation of one modification can indirectly influence others. Nonetheless, by using transient KDM5B overexpression, our study is designed to primarily target H3K4me3, supporting our conclusions.

In summary, maintenance of broad H3K4me3 domains in oocytes is required for genome silencing and timely oocyte maturation, and may contribute to transmission of a silenced genome to the early zygote. However, maintenance of broad H3K4me3 domains in the late zygote and early 2C embryo is not required for continued transcriptional repression or preimplantation embryo development. The establishment of H3K4me3 broad domains in the oocyte therefore instructs genome silencing during early development, but other mechanisms contribute to the maintenance of silencing prior to EGA.

## MATERIALS AND METHODS
### Subcloning and *in vitro* transcription
A complete coding sequence (cds) of wild-type *Kdm5b* (4945 bp) was obtained from a sequence-verified cDNA clone (MGC: 61207/IMAGE: 6826125, Source BioScience). The *Kdm5b* cds was subcloned in frame into a pcDNA3-YFP expression vector (Addgene plasmid #13033) for *in vitro* transcription with a YFP tag at the C-terminus. The following primers were used for this purpose: EcoRV forward, 5′-AACGATATCGCCCAG-GTCGCGGTGATGGAG-3′, XhoI and stop codon removed reverse, 5′-AACCTCGAGGTGTGTGTTTTGTTTTTCCTTTCGGC-3′. The high-fidelity PrimeSTAR GXL polymerase (Takara Bio) was used due to the long amplicon.

For generation of Del *Kdm5b-yfp*, the *Kdm5b*-YFP-pcDNA3 vector was used and the Jumonji C (JmjC) domain, which is essential for histone demethylase activity (Tsukada et al., 2006; Xiang et al., 2007), was deleted by using a Q5 Site-Directed Mutagenesis Kit (New England Biolabs) as per the manufacturer's instructions. This included using the following standard, non-mutagenic primers that flanked the JmjC domain: forward, 5′-TGCA-CTGTTGATTGGCTG-3′; reverse ,5′-GGGGAGTTTCATGCCACA-3′.

Both constructs were validated by Sanger sequencing (Eurofins Genomics) before further use. Sequencing primers for Kdm5b-YFP-pcDNA3 were: 1K_forward, ACCACACCTTCTGCCTGGTTCC; 2K_forward, AACTGTTCG-GAAATTGGGAGTA; 3K_forward, CCCGCCTACCTGCCCAATGGTA; 4K_forward, AGAGTTGTCTCCCCCTCCATGG. Sequencing primers for Kdm5b-Del-YFP-pcDNA3 were: forward, ACAGAGCTGGTGGAG-AAGGA; reverse, CATCTGGCAACAGCTCAAAA.

As both constructs were prepared in the pcDNA3-YFP expression vector and cloned in frame with the T7 promoter of the vector, we could perform *in vitro* transcription of mRNA for microinjection. First, the promoter and coding region was amplified by PCR using the PrimeSTAR GXL polymerase, as per the manufacturer's instructions, and standard pcDNA3 forward and reverse primers (Eurofins Genomics). The DNA was then purified by phenol-chloroform extraction and ethanol precipitation, and resuspended in nuclease-free water (Sigma-Aldrich, Merck). To generate capped and tailed mRNA, a mMESSAGE mMACHINE T7 Transcription Kit (Invitrogen, Thermo Fisher Scientific) and a Poly(A) Tailing Kit (Invitrogen, Thermo Fisher Scientific) were used according to their manuals. Finally, the mRNA was recovered with RNeasy MinElute Cleanup Kit (Qiagen) as per the manufacturer's instructions, eluted in nuclease-free water, aliquoted and stored at −80°C until further use.

### Animals and husbandry
All experiments involving use of mice were approved by the Norwegian Food Safety Authority (Mattilsynet) and registered in the national FOTS system (Forsøksdyrforvaltningens tilsyns- og søknadssystem) under application IDs 8743 and 23659. Experiments were subsequently conducted in accordance with the Norwegian regulation FOR-2015-06-18-761, which closely aligns with the EU directive 2010/63/EU on the protection of animals used for scientific purposes.

C57BL/6NRj (JANVIER LABS) mice were housed in individually ventilated cages (GM500 Mouse IVC Green Line), with a maximum of five females or one stud male per cage, and maintained under specific pathogen-free (SPF) conditions. Wood shavings (Nordic aspen, TAPVEI) were used as bedding, and all cages were provided with a house, paper and chewing sticks for environmental enrichment. The animal room was maintained on a 12 h light/12 h dark cycle at 21±1°C. Mice had ad libitum access to food (Rat and Mouse No.3 Breeding pellets, SDS Diets) and water. Animal care was provided by staff at the animal facility at Department of Comparative Medicine, Rikshospitalet, Oslo University Hospital.

### Microinjection and live-cell imaging of YFP
Injection of *in vitro*-transcribed mRNA was carried out in an ICSI dish (Vitrolife) with drops of 5 µl M2 medium (Sigma-Aldrich, Merck) covered with paraffin oil (OVOIL-100, Vitrolife). Around 20 oocytes/zygotes/embryos, hereafter referred to as cells, were added to each medium drop. For GV oocytes, 0.2 mM IBMX (Sigma-Aldrich, Merck) was added to the M2 medium to prevent meiotic resumption during the injection procedure. An Axio Observer inverted microscope with connected micromanipulators (Narishige) was used. The microinjection pipettes were made from glass capillaries (borosilicate glass, thin wall without filament, 1.2 mm OD, 0.94 mm ID, MultiChannel Systems) with a micropipette puller (Sutter Instrument). For injection set up, a microinjection pipette was first backfilled with 10 µl Fluorinert FC-770 (Sigma-Aldrich, Merck). The pipette was then filled from the front with mRNA solution at a concentration of ∼1150 ng µl$^{-1}$, or with nuclease-free water in given experiments. Water was favoured over Del *Kdm5b-yfp* mRNA as a control due a slightly higher but non-significant H3K4me3 levels observed in Del *Kdm5b-yfp*-injected zygotes compared to uninjected controls (Fig. 3B, left), and a possible inhibitory effect on development (Fig. 5B). A detailed explanation of the control strategy is provided in the following section.

Cytoplasmic injections were carried out with a FemtoJet microinjector (Eppendorf) and a Piezo PMM4G (Prime Tech) for easier penetration of the membrane. Injections were performed in a consistent manner such that the disturbance seen in the cytoplasm upon injection was about the size of a (pro)nucleus. Approximately 2 h post-injection (or towards the end of the arrest period for SN oocytes in Fig. 1 and Fig. S2, or after the 6 h activation protocol for oocytes in Fig. 1E and Fig. S2I), the cells were imaged for YFP

expression to confirm successful mRNA injection and efficient translation. Cells were placed in pre-warmed 5 µl M2 medium drops covered with paraffin oil in a glass-bottom dish (MatTek). An Axio Observer and Axiocam 503 mono camera were used with a heated stage at 37°C. The cells with no or very weak YFP fluorescence were excluded from further experimentation. After both microinjection and live-cell imaging, the cells were transferred back to their respective culture dish and incubated at 37°C and 6% CO$_2$ until the next experimental step.

### Control strategy for microinjection experiments
Our primary experimental strategy involved injecting mRNA encoding a catalytically inactive version of KDM5B, Del *Kdm5b-yfp*, in which the Jumonji C domain (essential for histone demethylase activity) was deleted. Data using this strategy on zygotes and embryos are presented in Figs 3 and 5, alongside uninjected controls and wild-type *Kdm5b-yfp* mRNA-injected groups. These results indicated a tendency toward higher H3K4me3 levels in zygotes injected with Del *Kdm5b-yfp* mRNA, as well as a possible inhibitory effect on development. This is why we noted in the final paragraph of the Results section: 'We suspect that Del KDM5B proteins may persist after binding to broad H3K4me3 domains due to the absence of catalytic activity. This may interfere with timely H3K4me3 reprogramming by endogenous KDM5B and affect developmental progression, as previously shown (Dahl et al., 2016; Liu et al., 2016).' Based on these observations, we later opted to use water-injected controls (Figs 1 and 4), matching the injection buffer used for mRNA dilution, instead of Del *Kdm5b-yfp* injection, to avoid introducing additional effects potentially caused by overexpression of the catalytically inactive variant.

Importantly, we observed no evidence of an overexpression side effect: RNA-seq analysis (Fig. 4) revealed no significant differences in expression between water- and mRNA-injected zygotes or 2-cell embryos, and *Kdm5b-yfp*-injected embryos developed normally (Fig. 5). Had we observed significant differences between the groups at these stages, we acknowledge that a control mRNA encoding YFP alone would have been warranted to rule out general overexpression effects. That said, such a control would not be equivalent in size (*Yfp*, 726 nt; *Kdm5b-yfp*, 5471 nt), and injection of fluorescently tagged proteins is widely used for overexpression in oocytes and embryos, with minimal reported side effects.

It is also worth noting that different strategies have been employed to study *Kdm5b* overexpression. For example, Zhang et al. (2025) used a point mutation (H499A) to inactivate the catalytic function of KDM5B while retaining the full-length protein, enabling distinction between enzymatic and non-enzymatic roles of the protein in early 2-cell embryos. Our approach, deletion of the entire Jumonji C domain, also renders KDM5B catalytically inactive but may affect additional structural aspects of the protein, thus providing a complementary perspective. Furthermore, Zhang et al. (2016) performed *Kdm5a* and *Kdm5b* overexpression in SN oocytes but did not specify the nature of their control, highlighting the broader challenge of control selection in this system.

While we recognise that a control mRNA encoding YFP alone may have added an additional layer of clarity, we believe our choice of water-injected controls was justified based on the observed effects of the Del *Kdm5b-yfp* construct and the limitations of alternative strategies. Further, we acknowledge that also including a catalytically inactive *Kdm5b* construct for the oocyte experiments (Fig. 1) would help determine whether the observed phenotype is due to the demethylase activity of KDM5B or potential non-enzymatic functions. However, we would like to note that in our zygote and 2-cell embryo experiments (Fig. 3), embryos injected with Del *Kdm5b-yfp* mRNA showed H3K4me3 levels comparable to uninjected controls, whereas embryos injected with wild-type *Kdm5b-yfp* exhibited a marked reduction in H3K4me3. Although these results are from later developmental stages, they suggest that the phenotype observed is primarily due to the enzymatic activity of KDM5B, rather than to non-enzymatic functions.

In conclusion, we opted for water-injected controls in given experiments because results from using the catalytically inactive *Kdm5b* construct, Del *Kdm5b-yfp*, showed that it had specific biological effects, making it unsuitable as an inert control. Water injection allowed us to control for any effects of the injection procedure itself, while also providing a consistent baseline using the same buffer used for mRNA dilution. Had we observed

a difference in RNA-seq between the water- and *Kdm5b-yfp* mRNA-injected embryos, inclusion of an alternative control mRNA would have been warranted.

## GV oocyte isolation and *in vitro* culture

Three- to 5-week-old female mice, which had received an intraperitoneal (IP) injection with 5 IU pregnant mare serum gonadotropin (PMSG, Prospec) 45-48 h earlier, were euthanised by cervical dislocation. The ovaries were dissected out, placed in a microtube with 1 ml M2 medium and quickly brought back to the embryology lab for further processing. Around four ovaries were processed at a time, while the remaining ovaries were kept in the tube with M2 medium at 37°C. First, any fat and other tissue surrounding the ovaries were removed under a stereomicroscope at 37°C. The ovaries were then put in a clean Petri dish with pre-warmed M2 medium containing 0.2 mM IBMX to prevent meiosis resumption. GV oocytes were isolated by holding the ovary in place with fine-tip tweezers and puncturing follicles with a 30 G syringe (BD Micro-Fine+, Becton Dickinson). Oocytes were quickly transferred to a pre-warmed 5-well dish (Vitrolife) with 0.5 ml M2 medium containing 0.2 mM IBMX in all the wells, and covered with paraffin oil. Oocytes were sorted into groups according to the absence or presence of a PVS, i.e. a gap between the oolemma and the zona pellucida. It has been demonstrated that the ability to form a PVS within 1 h of *in vitro* culture with IBMX correlates highly with a SN chromatin configuration, and with successful maturation, fertilisation and blastocyst development (Inoue et al., 2007).

Injection of the SN oocytes and live-cell imaging of YFP were performed as described in the section 'Microinjection and live-cell imaging of YFP' (screened for YFP fluorescence towards the end of their meiotic arrest period, i.e. 6 h or 12-14 h post-injection). After the injections, oocytes were transferred to pre-equilibrated micro-droplet dishes with 25 µl drops of G-1 PLUS culture medium (Vitrolife) containing 0.1 mM IBMX and covered with paraffin oil. The oocytes were incubated at 37°C and 6% $CO_2$ for the duration of their arrest period. For one replicate in the 6 h meiotic arrest experiment (Fig. S2G), MEM Alpha medium (Gibco, Thermo Fisher Scientific) with 10% fetal bovine serum (FBS, Sigma-Aldrich, Merck) and 0.1 mM IBMX was used as culture medium instead. After the arrest period, the SN oocytes were either kept arrested while treated with EU and subsequently fixed, as described in the section 'Click-iT RNA imaging assay coupled with immunofluorescence staining' or prepared for IVM as explained in the following section.

## In vitro maturation

After live-cell imaging, the arrested SN oocytes were transferred to a pre-equilibrated 5-well 'washout' dish with 0.5 ml of the relevant culture medium (G-1 PLUS or MEM Alpha with 10% FBS) in each well covered with paraffin oil. The oocytes were briefly washed in the fresh medium in one well and then left in another well for ~45 min at 37°C and 6% $CO_2$ to fully wash out the IBMX and allow for meiotic resumption. After IBMX washout, the oocytes were moved either to a normal 12-well dish for incubation in a standard incubator (Miri multi-room incubator, ESCO) or to an EmbryoSlide (Vitrolife) culture dish for monitoring the GVBD and PB extrusion timing in a EmbryoScope (Vitrolife) time-lapse incubator. In addition to 37°C and 6% $CO_2$, both the standard incubator and the EmbryoScope used for *in vitro* maturation (IVM) had a reduced $O_2$ level of 5%. Both types of culture dishes had 25 µl culture medium per well and were covered with paraffin oil. G-1 PLUS medium was used for all experiments involving the EmbryoScope and for most other IVM experiments. However, as previously mentioned, MEM Alpha medium with 10% FBS was used for one replicate in the 6 h meiotic arrest experiment (Fig. S2G). For all experiments, IVM was performed for 15-16 h counting from when the oocytes were put in the 5-well 'washout' dish.

## Chemical oocyte activation

Calcium-free activation medium (in house produced CZB; Biopsy medium, Origio, CooperSurgical; or G-PGD, Vitrolife) with 5-10 mM hexahydrate strontium chloride ($SrCl_2$, Sigma-Aldrich, Merck) and 5 µg ml$^{-1}$ cytochalasin B (CB, Sigma-Aldrich, Merck) was used for the activation protocol, as previously described (Kishigami et al., 2006). The commercial media were prepared according to the manufacturer's instructions. Oocytes were either cultured in activation medium for 6 h, or transferred after 3 h to a G-1 PLUS culture medium supplemented with 5 µg ml$^{-1}$ CB and cultured for an additional 3 h at 37°C and 5% $CO_2$ to avoid PB2 extrusion.

Following activation of IVM oocytes, they were inspected for presence of two maternal pronuclei (2PNs). The parthenogenetic 1C embryos were subsequently transferred to a pre-equilibrated 5-well washout dish with 0.5 ml of G-1 PLUS culture medium in each well covered with paraffin oil. The 1C embryos were briefly rinsed in fresh medium in one well and then left in another well for ~45 min at 37°C and 6% $CO_2$ to wash out the CB and allow the restoration of actin polymerisation. Next, the parthenogenetic 1C embryos were taken for 4 h EU treatment, fixed and stained as described in the section 'Click-iT RNA imaging assay coupled with immunofluorescence staining' (Fig. S2G).

For *in vivo* matured MII oocytes (Fig. 1E, Fig. S2H,I), YFP expression was checked after the 6 h activation protocol following the steps in the section 'Microinjection and live-cell imaging of YFP'. The parthenogenetic 1C embryos were then washed with fresh medium to remove the CB, labelled with EU for 2 h, fixed and stained as described in the next section.

## Click-iT RNA imaging assay coupled with immunofluorescence staining

Newly synthesised RNA was detected using Click-iT RNA Alexa Fluor 594 Imaging Kit (Invitrogen, Thermo Fisher Scientific) and coupled with antibody labelling, as per the kit manufacturer's instructions. Adjustments to the protocol were made for adaptation to oocytes/zygotes/embryos, hereafter referred to as cells. First, the cells were labelled with 2 mM 5-ethynyl uridine (EU) added to G-1 PLUS culture medium for 2 h; specific activated oocytes were labelled with EU for 4 h (Fig. S2G). The cells were then transferred to a drop of M2 medium before they were briefly (~10-30 s) held in a drop of Tyrode's Solution, Acidic (Sigma-Aldrich, Merck) to remove the zona pellucida for improved staining. Next, the cells were washed in M2 medium and fixed in 4% paraformaldehyde (PFA, Thermo Fisher Scientific) for 15 min at room temperature. The cells were transferred to 100 µl of Dulbecco's phosphate-buffered saline (DPBS, without calcium chloride and magnesium chloride, Sigma-Aldrich, Merck) with 0.3% bovine serum albumin (BSA, 35% in DPBS, Sigma-Aldrich, Merck) in a well of a non-treated 96-well plate with a V bottom (Nunc, Thermo Fisher Scientific) for 10 min to wash out the PFA. The cells were then transferred to a new well with 200 µl DPBS+0.3% BSA, the lid was put back on, and the plate was sealed with parafilm and kept at 4°C until the protocol was continued (never more than 5 days).

Upon continuation, all subsequent steps were performed at room temperature, unless otherwise stated, and in a volume of 50 µl in wells of the 96-well plate. The cells were first permeabilised in 0.5% Triton X-100 (Sigma-Aldrich, Merck) in DPBS with 0.3% BSA for 15 min, followed by 15 min wash in DPBS with 0.3% BSA. While waiting for the wash, the Click-iT reaction cocktail was prepared, as per the kit manufacturer's instructions, but scaled down to appropriate volumes. The cells were shortly rinsed in a new well of DPBS with 0.3% BSA, before they were treated with the Click-iT reaction cocktail for 30 min. The plate was covered to keep it dark for this and all subsequent steps. After the Click-iT reaction, the cells were washed twice for 15 min in Click-iT reaction rinse buffer with 0.3% BSA added to reduce stickiness. Next followed three additional steps of 15 min blocking in DPBS with 0.3% BSA and 0.01% TWEEN 20 (Sigma-Aldrich, Merck). The cells were then incubated with the primary antibody, rabbit monoclonal anti-H3K4me3 (04-745, Merck Millipore), diluted 1:300 in blocking solution, before the plate was sealed again and left at 4°C overnight (16-20 h).

The following day, the cells were washed three times for 15 min in blocking solution and then left in the secondary antibody diluted in blocking solution for 1 h. The secondary antibody depended on the type of experiment performed. For uninjected zygotes and 2C embryos (Fig. 2A,B) and for some parthenogenetic 1C embryos (Fig. S2G), a goat anti-rabbit IgG cross-adsorbed secondary antibody, Alexa Fluor 488 (A-11008, Invitrogen, Thermo Fisher Scientific) diluted 1:300, was used. For all other experiments where Click-iT RNA imaging assay was coupled with anti-H3K4me3 immunofluorescence staining, a goat anti-rabbit IgG cross-adsorbed secondary antibody, Alexa Fluor 350 (A-11046, Invitrogen, Thermo Fisher Scientific)

diluted 1:500, was used. In all cases, the cells were next washed three times for 15 min in blocking solution again. In the two aforementioned experiment types with the Alexa Fluor 488 secondary antibody, the DNA in the cells was also stained. They were treated with Hoechst 33342 (from the kit) diluted 1:1000 in blocking solution for 15 min, followed by a blocking solution wash for 15 min. A black 8-well diagnostic microscope slide (Thermo Fisher Scientific) was marked with relevant info and 1 μl of SlowFade Gold Antifade Mountant (Invitrogen, Thermo Fisher Scientific) was added to each well. The cells were transferred in relevant groups to the respective wells on the slide, and then a 24×60 mm coverslip was carefully placed on top. The coverslip was sealed with nail polish and left to dry at 4°C.

All the cells on the slide were imaged the same day using an Axio Observer epifluorescence inverted microscope (ZEISS) with an Axiocam 503 mono camera (ZEISS). Images were taken with a 63× oil immersion objective in the central plane of each (pro)nucleus. Fluorescence intensities were measured in ZEN Blue v.2.6 software (ZEISS). For quantifying nascent RNA and H3K4me3 in the cells, the mean intensity of a similar sized region in the cytoplasm was subtracted from the mean intensity of the nucleus. Representative fluorescence microscopy images were processed in ImageJ v1.54f.

### Immunofluorescence and DNA staining of MII oocytes

Oocytes that had been injected at SN stage, kept meiotically arrested for 6 or 12-14 h and then released for IVM were stained using a published protocol (Dahl et al., 2016) with minor adjustments. In these particular experiments, the zona pellucida was not removed in order to keep the PBs joined with the oocytes. After the full IVM period, MII oocytes were fixed in 4% PFA for 15 min at room temperature. The oocytes were transferred to 100 μl of DPBS (without calcium chloride and magnesium chloride) with 0.3% BSA in a well of a non-treated 96-well plate for 15 min to wash out the PFA. The cells were then transferred to a new well of 200 μl DPBS with 0.3% BSA, the lid was put back on, and the plate was sealed with parafilm and kept at 4°C until the protocol was continued (never more than 5 days).

All the following steps were performed at room temperature, unless otherwise stated, and in a volume of 50 μl in the 96-well plate. Oocytes were first permeabilised in 0.1% Triton X-100 in DPBS with 0.3% BSA for 15 min, followed by three rounds of 15 min in blocking solution, which again was DPBS with 0.3% BSA and 0.01% TWEEN 20. The oocytes were then transferred to the primary antibody, rabbit monoclonal anti-H3K4me3 (04-745, Merck Millipore) diluted 1:300 in blocking solution, before the plate was sealed again and left at 4°C overnight (16-18 h).

The following morning, oocytes were washed three times for 15 min in blocking solution and then put in the secondary antibody for 1 h. A goat anti-rabbit IgG cross-adsorbed secondary antibody, Alexa Fluor 647 (A-21244, Invitrogen, Thermo Fisher Scientific) diluted 1:300 in blocking solution, was used. The oocytes were then washed three times for 15 min in blocking solution again, before their DNA was stained with Hoechst 33342 diluted 1:1000 in blocking solution for 15 min. Finally, a 15 min blocking solution wash was performed before the oocytes were prepared for imaging on a microscope slide as described in the section 'Click-iT RNA imaging assay coupled with immunofluorescence staining'.

Oocytes on the slide were imaged the same day using a LSM 880 Axio Observer confocal microscope (Zeiss) located at the Advanced Light Microscopy Core Facility, The Norwegian Radium Hospital. With a 63× oil immersion objective, z-stack imaging was performed to capture all the chromosomes in the oocytes and their PBs. Imaris image analysis software (Oxford Instruments) was used to quantify H3K4me3 by measuring mean Alexa Fluor 647 intensity within a 3D chromosome region defined by the DNA staining.

### Superovulation, zygote/MII oocyte isolation and in vitro culture

Follicle growth was stimulated in 3- to 4 week-old females (Kolbe et al., 2014) by IP injection of 5 IU PMSG between 1:30 p.m. and 2:00 p.m. Approximately 45 h later, 5 IU human chorionic gonadotropin (hCG, Chorulon, MSD Animal Health) was injected intraperitoneally to induce ovulation. Directly after hCG injection, females were housed 1:1 with stud males. Males used as studs ranged from 8 weeks to 8 months. The following morning before 9:00 a.m., the females were removed from the males' cages

and euthanised by cervical dislocation. The ovaries with oviducts and the upper parts of the uterine horns were dissected out in one piece, placed in a microtube with 1 ml M2 medium and quickly transferred to the embryology lab for further processing. Under a stereomicroscope (Nikon), the zygotes surrounded by cumulus cells were isolated from the ampulla in a Petri dish with M2 medium at 37°C. To remove the cumulus cells, zygotes were briefly put in hyaluronidase solution (HYASE-10X, Vitrolife) diluted 1:10 in M2 medium. The zygotes were then washed in three consecutive wells of 0.5 ml M2 medium covered with paraffin oil. The normally fertilised oocytes (2 PNs) were transferred to a pre-equilibrated micro-droplet culture dish (Vitrolife) with 25 μl of G-1 PLUS culture medium in each well and covered with paraffin oil, ~10 zygotes per drop. The zygotes were incubated at 37°C and 6% $CO_2$ until further use.

The same approach was used for obtaining in vivo-matured MII oocytes, but with different hormone injection timings and no copulation. 5 IU PMSG was administered at 5:00 p.m., while 5 IU hCG was given 47 h later. The oocytes were isolated early the following morning, ~16 h post-hCG injection.

### Cryopreservation and thawing of zygotes

Most experiments were performed without cryopreservation and thawing. However, for the second replicate of early 2C blastomere injection (Fig. 3D) and for injected 2C embryos collected for RNA-seq (Fig. 4 and Fig. S4), we made use of thawed zygotes to manage time intervals. The zygotes were isolated and cultured as described in the section 'Superovulation, zygote/MII oocyte isolation and in vitro culture'. Subsequently, they were cryopreserved directly after isolation using the cryopreservation media kit Embryo Freezing Pack (Origio, CooperSurgical) following the protocol included. The zygotes were loaded in CBS High Security embryo straws (Cryo Bio System, NordicCell), with 20-30 zygotes per straw. The straws were put into an Asymptote EF600M controlled rate freezer (Grant Instruments) and the following program was run: from room temp to −7°C at 2°C min⁻¹ followed by manual seeding at this temperature, then further down to −30°C at 0.3°C min⁻¹ and finally down to −100°C at 10°C min⁻¹. The straws were then plunged into liquid nitrogen ($LN_2$) and transferred for storage at −196°C until further use.

On the morning of the experiment, the zygotes were thawed using the Embryo Thawing Pack (Origio, CooperSurgical), as per the manufacturer's instructions. The zygotes were put in culture and monitored every hour from when we could expect the earliest ones to divide to 2C. All 2C embryos were separated out and tracked. Next, they were injected 1-2 h after division following the steps in the section 'Microinjection and live-cell imaging of YFP'.

### Sample collection for total RNA-seq, library preparation and sequencing

The zona pellucidas of injected zygotes (~5-6 h post-injection) and 2C embryos (~5 h post-injection, which was ~5.5-8 h post-division to 2C) were removed as described in the section 'Click-iT RNA imaging assay coupled with immunofluorescence staining' and briefly washed through three drops of DPBS (without calcium chloride and magnesium chloride). In an RNase-free environment, each zygote/embryo was carefully pipetted together with 1 μl DPBS into 8-tube 0.2 ml PCR strips (Axygen) containing 6 μl DPBS with 1:200 RiboLock RNase inhibitor (Thermo Fisher Scientific) in each tube. Samples were flash frozen by submerging the bottom part of the tubes in $LN_2$, followed by direct transfer for storage at −80°C until library preparation was performed. We placed 10 single zygotes in separate tubes for both the Kdm5b and Ctrl water group. To utilise all surviving cells, we pooled some of the 2C embryos where only one cell was intact after injection. The dead cell was excluded during zona pellucida removal. In the Kdm5b group of 2C embryos, there were 6 samples (5 single 2C embryos with both cells intact and one pool of three 2C embryos with one cell intact). In the Ctrl water group, there were nine samples (six single 2C embryos with both cells intact, two pools of three 2C embryos with one cell intact, and one pool of two 2C embryos with one cell intact).

Library preparation was performed with SMART-Seq Stranded Kit (634443, Takara Bio) for full transcriptome analysis of single cells. Libraries of zygotes and 2C embryos were prepared on two separate days

following the kit's user manual starting from intact cells (option 1, with fragmentation). For PCR1, we used indexes from a SMARTer RNA Unique Dual Index Kit – 96U Set A (634452, Takara Bio) to avoid index hopping, and we used 10 cycles (ultra-low input workflow). We did not pool the samples after PCR1. For PCR2, we used 13 cycles, based on a prior pilot experiment. Final RNA-seq libraries were purified using AMPure beads (Agencourt AMPure XP, Beckman Coulter) with a single round of clean-up to minimise sample loss. Successful removal of adapter dimers and appropriate library size distribution were confirmed using a 2100 Bioanalyzer (Agilent). All the zygote and 2C libraries were quantified with Qubit dsDNA HS kit (Thermo Fisher Scientific) and subsequently pooled in equimolar ratios. The final pool was purified once with 1.2× AMPure beads, eluted in EB buffer (10 mM Tris-HCl, pH 8.5) and delivered for sequencing at the Norwegian Sequencing Centre, Department of Medical Genetics Ullevål, Oslo University Hospital. The library pool was sequenced on a NovaSeq 6000 instrument (Illumina) using an SP flow cell with 200 cycles, generating 100 bp paired-end reads.

### RNA-seq data processing and differential expression analyses

First, the FASTQ formatted sequencing data underwent initial quality control using the tool FastQC (Andrews, 2010), and absence of contaminating sequences was verified with FastQ Screen (Wingett and Andrews, 2018). Next, the reads were trimmed by means of the FASTQ pre-processor fastp (Chen et al., 2018) v0.21.0 with default settings, except for the following: –detect_adapter_for_pe –correction –trim_poly_g –trim_poly_x –cut_tail –trim_front1 8 –trim_front2 8 –trim_tail1 1 –trim_tail2 1.

The trimmed reads were then mapped to the *Mus musculus* GRCm38 genome assembly (Genome Reference Consortium, 2012) using the alignment tool STAR (Dobin et al., 2013) v2.7.3a. The alignment was performed in two-pass mode and was guided by a RefSeq gene annotation (NCBI Mus musculus Updated Annotation Release 108.20200622) using the following settings: –sjdbGTFfile <refseq.gtf> –sjdbOverhang 88 –twopassMode Basic –outSAMtype BAM SortedByCoordinate –outSAMattributes All –outSAMunmapped Within –outFilterMismatchNoverLmax 0.1 –outFilterMatchNmin 25 –outFilterMismatchNmax 5 –peOverlapNbasesMin 20.

After mapping, additional quality assessment was performed by various functions in SAMtools (Danecek et al., 2021) v1.10 (stats, flagstats, idxstats) and in RSeQC (Wang et al., 2012) v2.6.4 (geneBody_coverage, tin, read_distribution, inner_distance, junction_annotation, junction_saturation, read_duplication, read_hexamer, infer_experiment). The reads were then assigned to RefSeq genes, using the same gene annotation (NCBI Mus musculus Updated Annotation Release 108.20200622) as previously, with featureCounts (Liao et al., 2014) v1.5.1 and the following settings: -p -B -C -s 2. A gene-by-sample read count table was generated, and the total number of fragments per sample that was assigned to genes was in the range 8.7-17.8 M.

In order to compare *Kdm5b* mRNA-injected versus water-injected control samples and look for differentially expressed genes, statistical analysis of the count data was run using the package DESeq2 (Love et al., 2014) v1.30.1 in the software environment R (R Core Team, 2021) v4.0.3. The injected *Kdm5b* mRNA artificially affected the counts and could potentially skew the data normalisation (*Kdm5b*-mapped counts constituted 2.3-15.8% of all counts in *Kdm5b* samples; 0-1% in control samples). We therefore removed the *Kdm5b*-assigned reads prior to normalisation and statistical analysis. The zygotes and the 2C embryos were first analysed separately. The control zygotes and the control 2C embryos were then compared.

To quantify TEs in our sequence reads, we used SalmonTE (Jeong et al., 2018). The default behaviour of SalmonTE is to assign reads to consensus sequences of repetitive elements from the Repbase database. To achieve finer resolution, however, we decided to map against all repeat instances in the RepeatMasker v4.0.5 database (Smit et al., 2014). For each repeat ($n$=18 538), all chromosomal instance sequences were extracted, including 25 bp flanks, and concatenated using a string of 200 Ns as spacer sequence (as in the RepEnrich method). A salmon index was created on the concatenated instance sequences ('salmon index') and the reads were aligned to the repeat index ('salmon quant –exprtype=count'). Finally, we tested for differential expression ('SalmonTE.py test –tabletype=csv –figtype=png –analysis_type=DE –conditions=<Group1>,<Group2>'). As *Kdm5b* mRNA was injected in the test group, any repeats overlapping *Kdm5b* would produce artificially high counts and skew the results. We therefore removed two repeats ['(AGCTGCC)n' and '(T)n'] that mapped to *Kdm5b* exons after the quantification, but before the statistical tests.

### Immunofluorescence and DNA staining of blastocysts

Blastocysts (day 4.5) were stained using a published protocol (Niakan and Eggan, 2013) that was further adapted. All steps were performed at room temperature unless otherwise stated, and in a volume of 50 µl in wells of a non-treated 96-well plate with V bottom. First, the blastocysts were fixed in 4% PFA for 30 min and washed twice for 15 min in blocking solution, which was DPBS with 3% BSA and 0.1% TWEEN 20. Next, they were transferred to a new well of 200 µl blocking solution, the lid was put back on, and the plate was sealed with parafilm and kept at 4°C until the protocol was continued (never more than 5 days).

Upon continuation, the blastocysts were put in permeabilisation solution, DPBS with 1% TWEEN 20, and the plate was sealed again and left at 4°C overnight (~16 h). The following morning, the blastocysts were washed twice for 15 min in blocking solution, and subsequently put in the primary antibody, rabbit anti-NANOG (61420, Active Motif), diluted 1:200 in blocking solution for 1 h. After this, they were washed three times for 15 min in blocking solution and left for 1 h in the secondary antibody, goat anti-rabbit IgG cross-adsorbed secondary antibody, Alexa Fluor 568 (A-11011, Invitrogen, Thermo Fisher Scientific). The blastocysts were once again washed three times for 15 min in blocking solution. Finally, they were DNA stained with Hoechst 33342, put on slides, and imaged as described in the section 'Click-iT RNA imaging assay coupled with immunofluorescence staining'. This time, however, the images were taken as *z*-stacks to capture all the cells of each blastocyst. Epiblast cells in these late-stage blastocysts were counted by identifying the NANOG-positive cells (Plusa et al., 2008). The total number of cells were counted from the DNA-stained nuclei, and cell counting was manually performed in ImageJ v1.52v using the cell counter plugin.

### Statistical analyses and data presentation

The individual oocyte, zygote or embryo served as the unit of analysis. Data points were pooled from two or three independent biological replicates, as specified in the figure legends. Each biological replicate consisted of freshly isolated oocytes or zygotes collected from approximately five female mice and pooled prior to experimental manipulation. All statistical methods, definitions of $n$ and exact $n$ values are provided in the figure legends. *P*-values or adjusted *P*<0.05 were considered statistically significant. Where applicable, two-tailed (two-sided) tests were used. Graph creation and statistical methods (excluding RNA-seq analyses) were performed using GraphPad Prism (GraphPad Software) version 9.3.1 (Windows) and 10.4.2 (MacOS).

### Acknowledgements

We are very grateful to our colleagues at the Department of Reproductive Medicine, Oslo University Hospital: Inna Johnson for excellent training in mouse work and Dr Maria Vera-Rodriguez for insightful contribution during planning of RNA-seq. We also express our sincere gratitude to the staff at the Norwegian Transgenic Center (NTS), especially head engineer Ingunn Mørk Jermstad, for valuable tips related to mouse work, providing spare mouse samples for testing purposes, assisting with medium preparation and generously allowing use of equipment. We are grateful for the microscopy services provided by the HSØ Advanced Light Microscopy Core Facility at Oslo University Hospital, and we extend special thanks to Dr Vigdis Sørensen and Dr Ellen Skarpen for expert assistance with confocal microscopy and the Imaris software. Finally, we acknowledge the high-quality services provided by the animal facility at the Department of Comparative Medicine, Rikshospitalet, and by the Norwegian Sequencing Centre, Department of Medical Genetics Ullevål, both at Oslo University Hospital. A broader discussion of this work is also included in the publicly available doctoral thesis of T.S. (Skuland, 2024).

### Competing interests

The authors declare no competing or financial interests.

### Author contributions

Conceptualization: J.A.D., G.D.G.; Data curation: T.S., J.V.J., M.L.; Formal analysis: T.S., J.V.J., M.L., P.Z.F., G.D.G.; Funding acquisition: J.A.D., P.Z.F., G.D.G.; Investigation: T.S., M.F., S.K., E.A.A., M.I., B.C.C., G.D.G.; Methodology: T.S., M.F., S.K., E.A.A., M.I., B.C.C., Y.L., J.A.D., G.D.G.; Project administration: T.S., G.D.G.;

Resources: P.Z.F.; Software: J.V.J.; Supervision: M.L., J.A.D., P.Z.F., G.D.G.; Visualization: T.S.; Writing – original draft: T.S., G.D.G.; Writing – review & editing: T.S., M.F., S.K., E.A.A., J.V.J., M.I., B.C.C., Y.L., M.L., J.A.D., P.Z.F., G.D.G.

**Funding**
This work was supported by a grant from the Livsvitenskap, Universitetet i Oslo to the convergence environment 'Epigenetics and bioethics of human embryonic development' (to J.A.D., P.Z.F. and G.D.G.), by the legacy of the Supreme Court lawyer Per Rygh (to T.S.) and by Danmarks Grundforskningsfond (DNRF115 to M.L.). Equipment and supplies were provided by Oslo University Hospital, the University of Oslo, and through purchases made with allocated grant funds. Open Access funding provided by the University of Oslo. Deposited in PMC for immediate release.

**Data and resource availability**
RNA-seq data have been deposited in GEO under the accession number GSE309712. The rest of the experiments did not generate datasets that require deposition in a public repository. Raw data are available upon request. All other relevant data and details of resources can be found within the article and its supplementary information.

**Peer review history**
The peer review history is available online at https://journals.biologists.com/dev/lookup/doi/10.1242/dev.204638.reviewer-comments.pdf

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
