## [Peer Review File · Development (Cambridge, England)]

Broad H3K4me3 domains support oocyte genome silencing and maturation but are dispensable for repression in early embryos

Trine Skuland, Madeleine Fosslie, Sherif Khodeer, Endalkachew Ashenafi Alemu, Jens Vilstrup Johansen, Marie Indahl, Blanca Corral Castroviejo, Yanjiao Li, Mads Lerdrup, John Arne Dahl, Peter Zoltan Fedorcsak and Gareth D. Greggains

DOI: 10.1242/dev.204638

Editor: Peter Rugg-Gunn

Review timeline

Original submission:	10 January 2025
Editorial decision:	6 February 2025
First revision received:	30 May 2025
Editorial decision:	30 June 2025
Second revision received:	30 September 2025
Accepted:	2 October 2025

Original submission

First decision letter

MS ID#: dev.204638

MS TITLE: Broad H3K4me3 domains maintain genome silencing in oocytes and transfer repression to zygotes

AUTHORS: Trine Skuland, Madeleine Fosslie, Sherif Khodeer, Endalkachew Ashenafi Alemu, Jens Vilstrup Johansen, Marie Indahl, Blanca Corral Castroviejo, Yanjiao Li, Mads Lerdrup, John Arne Dahl, Peter Zoltan Fedorcsak and Gareth D. Greggains

Dear Dr Greggains,

I have now received all the referees' reports on the above manuscript, and have reached a decision. The referees' comments are appended below, or you can access them online: please go to:

As you will see, the referees express interest in your work, but have some significant criticisms and recommend a substantial revision of your manuscript before we can consider publication. If you are able to revise the manuscript along the lines suggested, I will be happy receive a revised version of the manuscript. Your revised paper will be re-reviewed by one or more of the original referees, and acceptance of your manuscript will depend on your addressing satisfactorily the reviewers' major concerns. Please also note that Development will normally permit only one round of major revision. If it would be helpful, you are welcome to contact us to discuss your revision in greater detail.

Please attend to all of the reviewers' comments and ensure that you clearly highlight all changes made in the revised manuscript. Please avoid using 'Tracked changes' in Word files as these are lost in PDF conversion. I should be grateful if you would also provide a point-by-point response detailing how you have dealt with the points raised by the reviewers in the 'Response to Reviewers' box. If you do not agree with any of their criticisms or suggestions please explain clearly why this is so.

Reviewer 1

Advance summary and potential significance to field

Skuland et al investigated the role of histone H3K4me3 in the oocytes and early embryos by decreasing the overall level of H3K4me3 using an over-expression of Kdm5b, an H3K4 demethylase. The authors reported that the outcomes of decrease H3K4me3 cause a mild effect on gene de-repression, oocyte maturation but not on later genome activation and later embryonic development. I have a few concerns on the experimental approaches and control experiments and think these issues should be addressed.

Major comments:

-Over expression of Kdm5b

The main findings of this manuscript were to observe the outcomes after over-expression of Kdm5b-yfp. The data will be more convincing if the author can provide the validation to support this approach. For example, I can only see the yfp expression as an indication of the expression of Kdm5b-yfp protein after injection in the schematic experimental procedure in Fig 1A, 3A, and 4A. I can't find the direction evidence of the validation of Kdm5b expression for example qRT-PCR on the RNA level as well as not showing in the RNA-seq dataset (Fig 4) and/or IF data on the protein level to show the efficacy of over-expression. Also, the authors injected >1ug/ul of IVF RNA into oocytes/embryos, I wonder whether any dosage titration experiments have been carried out. The concentration was much higher than the usual operation protocols (usually tens to hundreds ug/ul). I was worried that the data was misinterpreted (no effects on maturation, embryo development and gene expression etc.) if there was an issue of this over-expression approach. Have the authors thought about to perform a parallel expression by overexpression of Kdm5a?

-It will be more informative if the authors can provide the expression of other histone markers to show the consequence after the decrease of H3K4me3 (e.g. H3K9me3).

-Most of the data were showed in the global level after IF or EU labelling. Although the authors showed the decrease of H3K4me3 and increase of nascent RNA synthesis, it will be more informative to show the loss of H3K4me3 mark in the "broad" domain region as well as key reactivation of genes after Kdm5b over-expression.

-control experiments

The author used injection of water as an injection control, it will be more sensible to inject similar concentration of RNA (e.g. yfp) as control to avoid the side-effect of the over-expression.

DNA staining data should be provided in Fig S1B to show the stage of SN oocytes.

Comments for the author

I will suggest the authors adding the control/validation experiments.

Reviewer 2*Advance summary and potential significance to field*

A broad non-canonical pattern of H3K4me3 was established during the late stage of mouse oogenesis, which has been reported to be involved in oocyte genome silencing with unclear mechanisms. Removal of such broad H3K4me3 after ZGA was required for normal mouse preimplantation development. Skuland et al. continued to pursue this issue, by investigating the requirement for broad H3K4me3 in oocyte maturation and preimplantation development at different developmental stages. They showed that broad H3K4me3 was required for genome silencing in mouse oocytes but not pre-EGA gene repression. Premature removal of broad H3K4me3 was compatible with EGA and preimplantation development.

Overall, the manuscript was well designed and written, with most of the results solid and logical. One limitation is that several findings in this paper were reported previously (Andreu-Vieyra, Plos Biology, 2010; Zhang et al., Nature, 2016; Zhang et al., EMBO J, 2025 (see below)). However, the authors are appreciated for their in-depth analysis of this question which provided mechanistic insights on this intriguing question. There are a number of questions that need to be addressed before the paper is considered for publication.

Major comments:

1. It is not clear to me why zygotes from Kdm5b OE MII oocytes displayed genome reactivation but not those from Kdm5b OE SN oocytes. What is the potential explanation? What is the proportion of genome-reactivated zygotes? Could this be because the cell numbers are still limited and only a fraction of them exhibited possible reactivation? I would suggest a larger number of cells for this experiment, given the discrepancy in the results.
2. Related to this question, I suggest that "transfer repression to zygotes" in the title of the paper and "... relieve genome silencing in both oocytes and resulting zygotes" should be re-considered or removed, as these statements depend on the eventual results and their interpretation.
3. Fig. 1F. Following the comments above, what about zygotes from control and Kdm5b OE MII oocytes (in addition to SN oocytes)?
4. Fig. 1C and 1G. It would be necessary to add quantitative analyses and p-value for these results.
5. 4. It would be important to add Kdm5b catalytic dead mutant in the experiments, to confirm that the phenotype is related to H3K4me3 but not non-enzymatic functions.
6. Fig. S1B. The nascent RNA signals between Kdm5b OE and NSN int. were inconsistent with the result in Fig. S1A (right panel).
7. I assume it is probably not feasible to examine the developmental outcome of SN oocytes injected with Kdm5b that exhibit increased EU signals, considering that only half of the oocytes displayed such an increase. It would be helpful at least to discuss this possibility in the paper. What is the blastocyst rate of the parthenogenetic zygotes derived from the MII oocytes injected with Kdm5b overall?
8. What are the products of nascent transcription in SN oocytes upon Kdm5b overexpression? Could the authors perform RNA-seq to investigate this?
9. Interestingly, Kdm5b OE in the SN oocytes would lead to a defective oocyte maturation rate. However, the underlying mechanism still remains elusive. Would it be possible for the authors to do a "rescue" experiment by treating these oocytes with transcriptional inhibitors to see whether the related phenotype could be alleviated upon inhibition of these ectopic transcription?
10. The limited function of broad H3K4me3 to ZGA through Kdm5b OE in the zygotes was recently reported 2. This needs to be discussed and cited.

Minor comments:

1. Fig. S2 was missing.
2. The authors are suggested to present representative figures besides the statistical results where necessary, e.g., Fig. 3 and Fig. S1.

Ref:

- 1 Wang, J. et al. Reconstitution of chromatin reorganization during mammalian oocyte development. *bioRxiv* (2024).
- 2 Zhang, J. et al. Histone methyltransferases MLL2 and SETD1A/B play distinct roles in H3K4me3 deposition during the transition from totipotency to pluripotency. *EMBO J* 44, 437-456, doi:10.1038/s44318-024-00329-5 (2025).

Reviewer 3

Advance summary and potential significance to field

The epigenome of mouse oocytes is unusual in having a 'non-canonical' pattern of the H3K4me3 histone modification: whereas in most cell types, H3K4me3 is localised at promoters and other cis-regulatory sequences, in mouse oocytes it is far more pervasive and also occupies large genomic domains. These broad H3K4me3 domains are predominantly in untranscribed genomic regions that also have low levels of DNA methylation, which is mutually exclusive with H3K4me3. The functional impact of broad H3K4me3 domains is not fully understood. Past work has indicated a potential and paradoxical role of H3K4me3 deposition in transcriptional repression at the end of oocyte growth (ref. 6, 7). Conversely, an impact on zygotic genome activation (ZGA) has also been inferred, from the findings that ZGA genes are relatively enriched in H3K4me3-marked genomic intervals, and that

inhibition of H3K4me3 removal in zygotes by knocking down the H3K4me3 demethylases KDM5A & B impairs ZGA (previous work from this group; ref. 5).

The current study represents a re-evaluation of the functional impact and a refinement in understanding of oocyte H3K4me3 broad domains. It comprises an elegant set of experiments that largely focus around the use of forced expression of KDM5B in oocytes, zygotes or 2-cell embryos, coupled with immunofluorescence analysis, to interrogate when broad H3K4me3 domains constitute a block to transcription of the genome. In essence, the authors find that ablation of bulk H3K4me3 in transcriptionally quiescent GV oocytes derepresses transcription (consistent with earlier findings), while precocious removal of H3K4me3 in zygotes or early 2C embryos does not induce premature embryonic genome activation (EGA) or qualitatively affect EGA. Consistent with this finding, precocious removal of H3K4me3 is compatible with normal progression of preimplantation development. Figure 3E summarizes the key outcomes. The absence of an effect of precocious H3K4me3 removal is unexpected and novel. On the other hand, the study does not really advance the mechanistic understanding of the role of broad H3K4me3 in transcriptional repression.

Comments for the author

The authors provide an informed discussion that recognises some of the limitations of their study. Amongst these is the fact that detection of genomic H3K4me3 in oocytes and embryos in the current study is entirely by immunofluorescence, such that global effects are measured. The authors provide the important caveat (lines 355-357) that "it is not clear whether removal of broad domains or reprogramming of promoters to a narrow H3K4me3 pattern is the critical function." Further in this regard, although the transcriptional reactivation observed upon expression of KDM5B in SN-stage GV oocytes is compelling, it remains an inference that broad H3K4me3 domains themselves contribute to transcriptional repression in oocytes. The authors should perhaps note that while broad H3K4me3 domains have been described in mouse, rat, bovine and porcine oocytes, they are not present in human oocytes (PMID: 31273069; 34818044). Therefore, whatever functional impacts H3K4me3 broad domains have on transcriptional repression in the oocyte and zygote may not apply universally in mammalian oocytes.

In view of the ambiguity of whether the effects of H3K4me3 on transcriptional repression in oocytes are mediated by broad domains or other genomic sites of H3K4me3, there is an argument that the title should be changed, as the authors cannot conclusively show involvement of broad domains in the molecular phenotypes.

In relation to the inability of forced removal of bulk H3K4me3 to induce precocious EGA in zygotes or early 2C embryos, the authors offer some possibilities (lines 336-350), but ultimately conclude that "the mechanisms that maintain genome silencing in late zygotes and early 2C embryos in the absence of broad H3K4me3 remain elusive." The authors should note that broad H3K4me3 domains are confined to the maternal genome, so the paternal genome is maintained in a quiescent state (excepting minor ZGA) in any case in the absence of broad H3K4me3 domains. Again, this fact should be recognised in the discussion.

Specific comments to address:

Line 120: "Kdm5b-yfp mRNA injection and 6 hours arrest was sufficient to reduce H3K4me3 to NSN levels". Given the very different chromatin distribution of NSN compared to SN oocytes, how are the authors able to make a comparison in H3K4me3 IF intensity between NSN and SN? It is notable that there is a huge variation in H3K4me3 signal intensity particularly in NSN oocytes (Fig. S1E). The Methods section does not give much detail about how IF intensity was measured and how differences in parameters such as chromatin compaction were taken into account.

Fig. 1C/1D: test the effect of Kdm5b-mRNA injection on IVM rates. In these experiments, the controls presented are uninjected oocytes or oocytes injected with water. These are not ideal controls, as they do not assess the effect of injection of a reagent, which could have off-target effects in addition to the desired target. Better controls would be injection of an irrelevant mRNA, scrambled mRNA, mRNA unable to express the protein, or mRNA encoding a catalytically inactive KDM5B protein. Elsewhere in the study, the authors have used an mRNA for a catalytically-dead KDM5B (e.g., Fig. 3B, 5). The authors do explain the preference for use of water as control over the

use of catalytically-dead Kdm5b-mRNA (lines 635-637), which might be justified, but an alternative mRNA as indicated above would be preferable. A better experimental strategy would have been to perform the first set of experiments (as in Figs. 1C, D) with a control injected mRNA and once having established that the control mRNA had an outcome indistinguishable from water injection, subsequent experiments would be justified in using water as control.

Fig. 1C is a bar chart showing the IVM rates in control, control injected or Kdm5b-mRNA injected GV oocytes. The figure legend gives the number of manipulated oocytes per group, and states that three replicates were carried out. If this is so, does this mean 3 replicates each of 36, 33 and 33 in the three groups, or that there were 3 replicates totalling 36, 33 and 33 in the three groups? Also, given that 3 replicates were carried out, the bar chart should indicate the variation between replicates. This comment would apply to other cases in which replicate experiments were performed, so the meaning is fully transparent.

Lines 146-148: "To further assess developmental competence, we cultured parthenogenetic zygotes to day 5 and found that a reduction of H3K4me3 at SN stage was also compatible with development to blastocyst (Fig. 1G)." This statement needs to be supported by an indication of the numbers of oocytes manipulated and activated, and the percentages reaching the blastocyst stage. In comparison in Fig. 5 the authors do properly quantify developmental progression of embryos injected at the 2C stage.

Fig. 2C: the data for Kdm5b mRNA from the RNA-seq of zygotes and early 2C embryos are consistent with Kdm5b being transcribed as part of the minor phase of ZGA; they could also indicate preservation of maternal Kdm5b mRNA during the phase of global maternal RNA degradation. Although not directly addressing this difference, it would be useful to complement these data with IF from KDM5B in zygotes and 2-cell embryos to demonstrate timing of appearance of KDM5B protein.

In general, evidence of 'exogenous' KDM5B expression in relation to timing of appearance or level of the native protein would be useful, although this may not be feasible depending upon availability of suitable antibodies. The authors may be justified in responding that abundance of the target modification H3K4me3, which they have quantified extensively, is what really matters.

Discussion lines 332-333: it is not clear how the authors are able to conclude "Finally, SN oocytes undergoing H3K4me3 depletion display a lag of several hours before increased transcription is detected", because there did not appear to be any figure that showed a time-course of transcriptional derepression following H3K4me3 depletion.

In the Methods (lines 837-840) the authors explain that the RNA-seq libraries from 2C embryos comprised both single 2C embryos and various small pools of 2C embryos with damaged blastomeres. It might be useful to highlight on the PCA (Suppl Fig. S4A) the single 2C embryos and the 2C embryo pools, in case the nature of the 2C embryos contributes to variation amongst the libraries.

First revision

Author response to reviewers' comments

Opening remarks from the authors

First and foremost, we would like to express our gratitude to you, the reviewers, for taking the time to evaluate our manuscript. We greatly appreciate your feedback and the level of detail in your assessments. Your insightful comments and constructive suggestions have been invaluable in improving our work.

Here, we have included your original comments (presented in black text), which we have numbered for clarity and ease of reference. We have provided a point-by-point response outlining

how we have addressed each of the concerns raised. Our responses are highlighted in red text.

Reviewer 1

Summary of the advance made in this paper and its potential significance to the field

Skuland et al investigated the role of histone H3K4me3 in the oocytes and early embryos by decreasing the overall level of H3K4me3 using an over-expression of Kdm5b, an H3K4 demethylase. The authors reported that the outcomes of decreased H3K4me3 cause a mild effect on gene de-repression, oocyte maturation but not on later genome activation and later embryonic development. I have a few concerns on the experimental approaches and control experiments and think these issues should be addressed.

Response: We thank reviewer 1 for the comments and suggestions, which we have addressed in detail below.

Major comments:

- Overexpression of Kdm5b: The main findings of this manuscript were to observe the outcomes after over-expression of Kdm5b-yfp.
 - **Comment #1:**
The data will be more convincing if the author can provide the validation to support this approach. For example, I can only see the yfp expression as an indication of the expression of Kdm5b-yfp protein after injection in the schematic experimental procedure in Fig 1A, 3A, and 4A. I can't find the direction evidence of the validation of Kdm5b expression for example qRT-PCR on the RNA level as well as not showing in the RNA-seq dataset (Fig 4) and/or IF data on the protein level to show the efficacy of over-expression.

Response: We appreciate the reviewer's suggestion and agree that additional evidence of *Kdm5b-yfp* expression at the RNA and protein levels would further support our approach.

As described in the Materials and Methods section, we conducted live-cell YFP imaging of all oocytes and embryos in relevant experiments to confirm successful mRNA injection and efficient translation into protein. Oocytes/embryos with no or weak YFP fluorescence were excluded from further experimentation. To further validate KDM5B-YFP expression at the protein level, we now provide additional quantification of YFP fluorescence following live-cell imaging (New Fig. S1A, B). Quantification of YFP fluorescence intensity in the nuclei of SN oocytes demonstrates that exogenous *Kdm5b-yfp* mRNA is robustly translated into protein already at 3h post injection, with levels remaining high at 6h post injection (Fig. S1A). We would like to emphasise that the *Kdm5b* coding sequence was subcloned in-frame into the pcDNA3-YFP expression vector, placing the *Yfp* tag at the C-terminus of *Kdm5b*. This ensures that any detected YFP fluorescence in live-cell imaging directly corresponds to the full-length KDM5B-YFP fusion protein, confirming its successful translation.

As further validation, we provide *Yfp*- and *Kdm5b*-aligned read counts from the RNA-sequencing dataset (New Fig. S1C, D). The *Yfp* tag is detected at high levels in both *Kdm5b-yfp* mRNA-injected zygotes and 2C embryos (New Fig. S1C). *Kdm5b* is also endogenously expressed - at low levels in zygotes and at higher levels in early 2C embryos (Fig. 2C). However, *Kdm5b*-aligned read counts are significantly elevated in both zygotes and 2C embryos injected with *Kdm5b-yfp* mRNA (New Fig. S1D).

We have included this new figure, now designated as Fig. S1, in the updated Supplementary Information. Consequently, the figure previously labeled Fig. S1 in the original submission has been renumbered as Fig. S2. This adjustment fits well, as there was no Fig. S2 in the original version - a point also addressed under reviewer 2's comment #20. Additionally, we have inserted a sentence referring to the validation data in the new Fig. S1 at the end of the first paragraph in the Results

section (originally lines 102-108). The revised text is as follows (original submission text in black, new text in blue):

“To further explore the role of broad H3K4me3 domains in maintaining transcriptional repression in the oocyte and zygote, as well as their functional importance for oocyte maturation and embryo development, we overexpressed KDM5B in SN and MII stage oocytes. We cultured SN oocytes in 3-isobutyl-1-methylxanthine (IBMX), a phosphodiesterase inhibitor that induces reversible meiotic prophase arrest, to ensure sufficient time for KDM5B expression. For the *Kdm5b* mRNA used for overexpression, we employed a yellow fluorescent protein (YFP) tag to ensure successful protein expression prior to downstream analysis and/or in vitro maturation (IVM) (Fig. 1A). *The KDM5B overexpression approach was validated by live-cell imaging of YFP fluorescence and RNA-sequencing (RNA-seq), confirming robust expression at both protein and RNA level (Fig. S1).*”

New Fig. S1. Validation of KDM5B overexpression by YFP fluorescence and RNA-seq. (A) YFP fluorescence intensity in nuclei of SN oocytes following live-cell imaging at 3 h and 6 h post-injection, comparing uninjected controls (Ctrl) and *Kdm5b-yfp* mRNA-injected oocytes (*Kdm5b*). Total number of oocytes (one biological replicate): Ctrl $n = 8$ and *Kdm5b* $n = 8$. Lines and error bars: median and IQR. Pre-selected pairs were compared using the Friedman test (repeated measures design) with Dunn’s multiple comparisons test; adjusted significance indicated by P value notation. (B) Representative live-cell microscopy images of SN oocytes corresponding to (A). Images show brightfield and YFP fluorescence, with brightness and contrast adjusted consistently across samples (scale bar: 50 μm). (C) *Yfp*-aligned read counts from the RNA-seq dataset described in Fig. 4A for all samples: Ctrl water zygotes ($n = 10$), *Kdm5b* zygotes ($n = 10$), Ctrl water 2C ($n = 9$), and *Kdm5b* 2C ($n = 6$). Bars and error bars: median and IQR. Pre-selected pairs were compared using Kruskal-Wallis with Dunn’s multiple comparisons test; adjusted

significance indicated by P value notation. Read counts were obtained by Bowtie2 paired-end alignment to the YFP coding sequence (from pcDNA3-YFP; see Materials and Methods), followed by quantification with featureCounts.

(D) *Kdm5b*-aligned read counts from the RNA-seq dataset as in (C). Bars and error bars: median and IQR. Pre-selected pairs were compared using Kruskal-Wallis with Dunn's multiple comparisons test; adjusted significance indicated by P value notation. Read counts were extracted from the existing gene count table (see Materials and Methods).

P value notation: ns $P > 0.05$, * $P \leq 0.05$, ** $P \leq 0.01$, *** $P \leq 0.001$.

Finally, throughout the manuscript we show robust depletion of H3K4me3 in oocytes, zygotes, and embryos injected with *Kdm5b-yfp mRNA*. This demonstrates that injected mRNA is indeed translated into functional KDM5B-YFP protein. We hope this additional data and information sufficiently address the reviewer's concerns.

○ **Comment #2:**

Also, the authors injected $>1\mu\text{g}/\mu\text{l}$ of IVF RNA into oocytes/embryos, I wonder whether any dosage titration experiments have been carried out. The concentration was much higher than the usual operation protocols (usually tens to hundreds $\mu\text{g}/\mu\text{l}$). I was worried that the data was misinterpreted (no effects on maturation, embryo development and gene expression etc.) if there was an issue of this over-expression approach.

Response: We understand the reviewer's concern and would like to reassure you that there are strong reasons for the injection concentration used in our study.

The *Kdm5b-yfp* construct has a coding sequence of 5471 nucleotides (nt), excluding the 5' cap and poly-A tail, which was added following *in vitro* transcription for increased stability. This length is substantially longer than mRNAs encoding commonly injected fusion proteins such as H2B-mCherry (~1098 nt) and α -Tubulin-EGFP (~2064 nt) - approximately 5-fold and 2.7-fold longer, respectively. These shorter mRNAs are typically injected at concentrations ranging from 25 to 50 $\text{ng}/\mu\text{l}$, although higher concentrations up to 500 $\text{ng}/\mu\text{l}$ have also been reported. A larger transcript size likely contributes to reduced translation efficiency and/or increased susceptibility to degradation, thereby necessitating a higher injection concentration to achieve robust and consistent protein expression.

We initially tested lower concentrations of *Kdm5b-yfp mRNA* (data not shown), but these resulted in weak or undetectable YFP fluorescence during live-cell imaging. The final concentration of ~1150 $\text{ng}/\mu\text{l}$ was selected to ensure reliable expression in all injected oocytes and embryos used in the experiments. Importantly, based on our embryonic development data (see Fig. 5 in the manuscript), this concentration did not appear to have any detrimental effects on embryo viability or progression.

A similar approach was taken in the recent study highlighted by reviewer 2 (Zhang et al., 2025), where *Kdm5b* and *Setd1a* mRNAs, both encoding large proteins with coding sequences of ~5100 nucleotides, were injected at 800 $\text{ng}/\mu\text{l}$, while the shorter *Setd1b* mRNA (~4200 nucleotides) was injected at a lower concentration of 200 $\text{ng}/\mu\text{l}$. These differences further support the notion that different mRNAs require different injection concentrations depending on their properties, such as coding sequence length.

Reference cited in this response:

Zhang, J. et al. (2025) 'Histone methyltransferases MLL2 and SETD1A/B play distinct roles in H3K4me3 deposition during the transition from totipotency to pluripotency', *The EMBO Journal*, 44(2), pp. 437-456. Available at: <https://doi.org/10.1038/s44318-024-00329-5>.

○ **Comment #3:**

Have the authors thought about to perform a parallel expression by

overexpression of *Kdm5a*?

Response: Yes, we did consider including *Kdm5a* in parallel. When planning the study and initiating the subcloning procedures, we also prepared a *Kdm5a-mCerulean* construct. This was motivated by our earlier work (Dahl et al., 2016), in which simultaneous morpholino knockdown of *Kdm5a* and *Kdm5b* in 2-cell embryos prevented the removal of H3K4me3 that would normally occur at this stage.

However, based on findings from our 2016 study, we suspected that KDM5B may be the primary contributor due to its higher RNA levels in 2-cell embryos. This interpretation was further supported by a study published back-to-back with ours (Zhang et al., 2016), which showed that *Kdm5b* overexpression led to a more pronounced removal of H3K4me3 compared to *Kdm5a* overexpression in SN oocytes. For these reasons, we chose to focus on *Kdm5b-yfp* overexpression and did not proceed with *Kdm5a-mCerulean* mRNA injections.

The RNA-sequencing we perform in the present study further supports this decision: *Kdm5b* expression appears to be higher than *Kdm5a* in early 2-cell embryos (see Fig. 2C in the manuscript). Additionally, the recent study highlighted by reviewer 2 (Zhang et al., 2025) provides further evidence that KDM5B is the primary contributor: late 2-cell embryos with additional *Kdm5a* knockdown did not exhibit higher H3K4me3 levels than embryos with *Kdm5b* knockdown alone.

However, we do think it would be interesting to tease out the differences in function, if any, between KDM5B and KDM5A in a future study.

References cited in this response:

Dahl, J.A. *et al.* (2016) 'Broad histone H3K4me3 domains in mouse oocytes modulate maternal-to-zygotic transition', *Nature*, 537, pp. 548-552. Available at: <https://doi.org/10.1038/nature19360>.

Zhang, B. *et al.* (2016) 'Allelic reprogramming of the histone modification H3K4me3 in early mammalian development', *Nature*, 537, pp. 553-557. Available at: <https://doi.org/10.1038/nature19361>.

Zhang, J. *et al.* (2025) 'Histone methyltransferases MLL2 and SETD1A/B play distinct roles in H3K4me3 deposition during the transition from totipotency to pluripotency', *The EMBO Journal*, 44(2), pp. 437-456. Available at: <https://doi.org/10.1038/s44318-024-00329-5>.

- **Comment #4:**

It will be more informative if the authors can provide the expression of other histone markers to show the consequence after the decrease of H3K4me3 (e.g. H3K9me3).

Response: We certainly agree that assessment of additional histone marks, particularly those that may compensate for the loss of broad H3K4me3 domains, could provide insight into the mechanisms of genome silencing in the early embryo. We believe that publication of our findings would motivate other researchers to explore the relationship between H3K4me3 depletion and the spread of other repressive marks at this important developmental stage. Such studies would likely require use of ChIP-seq, or an orthogonal method, for sequence-specific evaluation of both broad and narrow marks with low cell numbers. The focus of our study was on broad H3K4me3 as a repressive mark during oogenesis and early embryogenesis and its role in early development.

- **Comment #5:**

Most of the data were shown in the global level after IF or EU labelling. Although the authors showed the decrease of H3K4me3 and increase of nascent RNA synthesis, it will be more informative to show the loss of H3K4me3 mark in the "broad" domain region as well as key reactivation of genes after *Kdm5b* over-expression.

Response: We share the reviewer's view that demonstrating loss of H3K4me3 specifically within broad domains, as well as identifying reactivated genes, would have strengthened our study. We acknowledged this limitation in our discussion (originally lines 351-362), but have now expanded this section in our updated discussion in response to reviewer 2's comment #17. Nonetheless, we believe our study offers valuable insights and provides a solid foundation for future research.

Importantly, the recent study highlighted by reviewer 2 (Zhang et al., 2025) partly addresses this point using Cleavage Under Targets and Tagmentation (CUT&Tag) for genome-wide H3K4me3 profiling of early 2-cell embryos in which *Kdm5b* was overexpressed from the zygote stage. Zhang et al. show extensive loss of H3K4me3 in broad genomic regions, referred to as non-canonical H3K4me3, following *Kdm5b* overexpression. Interestingly, this loss did not result in premature formation of canonical H3K4me3 peaks, defined as the narrower marks typically found at promoters of active genes. This point has also been incorporated into our updated Discussion section, in response to reviewer 2's comment #19. While Zhang et al. did not assess H3K4me3 profiles or transcriptomic changes in GV oocytes, we agree that further investigation of both epigenomic and gene-specific transcriptional responses to *Kdm5b* overexpression at the SN stage will be an important direction for future work.

Reference cited in this response:

Zhang, J. *et al.* (2025) 'Histone methyltransferases MLL2 and SETD1A/B play distinct roles in H3K4me3 deposition during the transition from totipotency to pluripotency', *The EMBO Journal*, 44(2), pp. 437-456. Available at: <https://doi.org/10.1038/s44318-024-00329-5>.

- **Comment #6:**

Control experiments: The author used injection of water as an injection control, it will be more sensible to inject similar concentration of RNA (e.g. *yfp*) as control to avoid the side-effect of the over-expression.

Response: Our primary experimental strategy involved the injection of mRNA encoding a catalytically inactive version, *Del Kdm5b-yfp* mRNA, in which the Jumonji C domain, essential for histone demethylase activity, was deleted. Data using this strategy on zygotes and embryos was presented in figure 3 and 5, alongside wild type *Kdm5b-yfp* mRNA injected groups and uninjected controls.

From preliminary data collected for zygotes and embryos, we saw a tendency towards higher levels of H3K4me3 in zygotes and embryos injected with *Del Kdm5b-yfp* mRNA. Indeed, we wrote in the final paragraph of the Results section: "*We suspect that Del KDM5B proteins may persist after binding to broad H3K4me3 domains due to the absence of catalytic activity. This may interfere with timely endogenous H3K4me3 reprogramming and developmental progression, as previously shown.*" The phrasing of the latter sentence may have been unclear, and we have adjusted this to (originally lines 296-297, original submission text in black and new text in blue): "*This may interfere with timely H3K4me3 reprogramming by endogenous KDM5B and affect developmental progression, as previously shown (Dahl et al., 2016; Liu et al., 2016).*"

Based on the mentioned observations, we opted to use water-injected controls, matching the injection buffer used for mRNA dilution, instead of *Del Kdm5b-yfp* injection, to avoid introducing additional effects potentially caused by overexpression of the catalytically inactive variant. We saw no evidence of an overexpression side-effect in zygotes or embryos, as RNA-seq data showed no significant differences in expression between water and mRNA injected zygotes or 2-cell embryos. Furthermore, we observed normal embryo development in *Kdm5b-yfp* injected zygotes and embryos. If we had observed significant differences between *Kdm5b-yfp* and water-injected controls at these stages, we acknowledge that there would have been a strong argument for injection of a *Yfp*-coding mRNA as a control, to rule out an overexpression effect. That said, the coding sequence length of *Yfp* alone at 726 nt and *Kdm5b-yfp* at 5471 nt would nevertheless have been substantially different. Additionally, injection of fluorescent tagged proteins is a widely used strategy for overexpression in oocytes

and embryos, and is considered to have limited side-effects, in itself.

It is worth noting that different strategies have been employed to study *Kdm5b* overexpression. For example, the Zhang et al., 2025 study used a point mutation (H499A) to specifically inactivate the catalytic function of KDM5B while retaining the full-length protein, allowing them to distinguish between enzymatic and non-enzymatic roles of the protein in early 2-cell embryos. Our approach, deletion of the entire Jumonji C domain, also renders KDM5B catalytically inactive, but may affect additional structural aspects of the protein, thus providing a complementary perspective.

Furthermore, the Zhang et al., 2016 study performed *Kdm5a* and *Kdm5b* overexpression in SN oocytes but did not specify the nature of their control injections, referring only to “control”. This highlights a general challenge in selecting and reporting appropriate controls in this system.

While we recognise that a *Yfp*-only control may have added an additional layer of clarity, we believe our choice of water-injected controls was justified based on the observed effects of the *Del Kdm5b-yfp* construct and the limitations of alternative strategies. We appreciate the reviewer’s suggestion and will take it into consideration in future studies.

References cited in this response:

Zhang, B. *et al.* (2016) ‘Allelic reprogramming of the histone modification H3K4me3 in early mammalian development’, *Nature*, 537, pp. 553-557. Available at: <https://doi.org/10.1038/nature19361>.

Zhang, J. *et al.* (2025) ‘Histone methyltransferases MLL2 and SETD1A/B play distinct roles in H3K4me3 deposition during the transition from totipotency to pluripotency’, *The EMBO Journal*, 44(2), pp. 437-456. Available at: <https://doi.org/10.1038/s44318-024-00329-5>.

- Comment #7:
DNA staining data should be provided in Fig S1B to show the stage of SN oocytes.

Response: We agree that including Hoechst DNA staining in what was originally Fig. S1B (now Fig. S2B, ref. response to comment #1) would have provided a helpful visual confirmation of the SN stage. However, DNA staining was omitted in order to extract the maximum amount of information from the samples, whilst avoiding potential crosstalk or bleedthrough from imaging four fluorophores in a single cell. We used an Alexa Fluor 350-conjugated secondary antibody (blue, emission peak ~442 nm) to detect H3K4me3, avoiding spectral overlap with both KDM5B-YFP (yellow, ~527 nm) and EU-Alexa Fluor 594 (red, ~617nm). This fluorophore combination allowed us to simultaneously confirm KDM5B-YFP expression while quantifying H3K4me3 and nascent RNA (EU) levels in the same oocyte.

We considered omitting the YFP tag in favour of a DNA stain but ultimately chose the current setup, as the large nucleus is readily identifiable from H3K4me3, EU, or brightfield images. We believe this approach provides greater overall utility and avoids compromising the multiplexed imaging. Importantly, we are confident in the SN-stage identity of the oocytes based on our well-established sorting method, described in the Materials and Methods section. Specifically, oocytes were grouped by the presence or absence of a perivitelline space (PVS). As demonstrated by Inoue et al. (2007), and confirmed in our hands, the ability to form a PVS within 1 hour of *in vitro* culture with IBMX correlates strongly with SN chromatin configuration and meiotic competence.

Importantly, we note that none of the oocytes categorised as SN in the control group (Ctrl water SN in Fig. 1B) display elevated levels of nascent RNA, consistent with their transcriptionally silent state and thus providing robust functional support for accurate stage classification.

In conclusion, although DNA staining was omitted, all oocytes were carefully staged using a well-established morphological and physiological marker.

Reference cited in this response:

Inoue, A. *et al.* (2007) 'The perivitelline space-forming capacity of mouse oocytes is associated with meiotic competence', *Journal of Reproduction and Development*, 53(5), pp. 1043-1052. Available at: <https://doi.org/10.1262/jrd.19064>.

Suggestions to authors:

I will suggest the authors adding the control/validation experiments.

Response: Thank you for the suggestion. We have addressed this point in our response to comment #1, where we provide additional validation data and further information.

Reviewer 2**Summary of the advance made in this paper and its potential significance to the field**

A broad non-canonical pattern of H3K4me3 was established during the late stage of mouse oogenesis, which has been reported to be involved in oocyte genome silencing with unclear mechanisms. Removal of such broad H3K4me3 after ZGA was required for normal mouse preimplantation development. Skuland *et al.* continued to pursue this issue, by investigating the requirement for broad H3K4me3 in oocyte maturation and preimplantation development at different developmental stages. They showed that broad H3K4me3 was required for genome silencing in mouse oocytes but not pre-EGA gene repression. Premature removal of broad H3K4me3 was compatible with EGA and preimplantation development.

Overall, the manuscript was well designed and written, with most of the results solid and logical. One limitation is that several findings in this paper were reported previously (Andreu-Vieyra, *Plos Biology*, 2010; Zhang *et al.*, *Nature*, 2016; Zhang *et al.*, *EMBO J*, 2025 (see below)). However, the authors are appreciated for their in-depth analysis of this question which provided mechanistic insights on this intriguing question. There are a number of questions that need to be addressed before the paper is considered for publication.

Response: We appreciate reviewer 2's thoughtful summary and constructive feedback. We are pleased that the strengths of our study, particularly its design, clarity, and depth, were recognised.

We acknowledge that some findings overlap with previous studies, including the recent publication by Zhang *et al.* (2025). While it is unfortunate that we were unable to publish earlier, we believe that independent confirmation strengthens the field and that our study offers complementary and extended analyses. Importantly, our work rigorously tests the hypothesis that broad H3K4me3 removal initiates embryonic genome activation (EGA) and reveals that, contrary to expectations, this is not the case. This is particularly surprising given that gene reactivation was observed in a subset of oocytes following H3K4me3 removal.

Major comments:

- It is not clear to me why zygotes from Kdm5b OE MII oocytes displayed genome reactivation but not those from Kdm5b OE SN oocytes.
 - **Comment #8:**
What is the potential explanation?

Response:

We thank the reviewer for this excellent question, and we agree that it warrants further elaboration. One likely explanation involves the maternal RNA degradation process (M-decay) that occurs as SN-stage GV oocytes resume meiosis and progress to the MII stage. Previous studies have demonstrated a major reduction in total RNA during mouse oocyte maturation, including degradation of transcripts related to prophase I arrest and protein translation

(Bachvarova et al., 1985; Paynton et al., 1988; Su et al., 2007). M-decay has been found to be necessary for developmental competence in mice (Yu et al., 2016), and appears equally important in human oocytes (Sha et al., 2020).

Thus, when oocytes were injected with *Kdm5b-yfp* mRNA at the SN stage and subsequently subjected to 6 h meiotic arrest followed by 15-16 h IVM prior to activation, the exogenous mRNAs may have been subject to active degradation during M-decay. This could have reduced *Kdm5b-yfp* transcript abundance below the level required for effective H3K4 demethylation in the resulting parthenogenetic zygotes and tipped the balance toward remethylation. This interpretation is also consistent with our statement in the Results section (originally lines 132-135) referencing what was Fig. S1G (right), now Fig. S2G: *“Interestingly, PN chromatin in the KDM5B overexpression group showed higher H3K4me3 levels than PB chromatin (Fig. S1G, right), in contrast to similar levels seen between MII oocyte chromatin and first PBs (Fig. S1F), suggesting that some H3K4 remethylation may occur in the early zygote.”* If remethylation predominated and led to slightly elevated H3K4me3, this may have been enough to prevent genome reactivation in the parthenogenetic zygotes derived from SN-stage injected oocytes.

We have now clarified this reasoning in the Results section. The revised text reads as follows (original submission text in black, new text in blue):

“Interestingly, PN chromatin in the KDM5B overexpression group showed higher H3K4me3 levels than PB chromatin (Fig. S2G, right), in contrast to similar levels seen between MII oocyte chromatin and first PBs (Fig. S2F), suggesting that some H3K4 remethylation may occur in the early zygote. This could reflect degradation of Kdm5b-yfp mRNA as part of maternal RNA decay during meiotic maturation (Paynton et al., 1988), with the resulting elevated H3K4me3 levels potentially preventing genome reactivation in zygotes derived from SN-stage injected oocytes.”

References cited in this response:

Bachvarova, R. et al. (1985) ‘Changes in total RNA, polyadenylated RNA, and actin mRNA during meiotic maturation of mouse oocytes’, *Developmental Biology*, 108(2), pp. 325-331. Available at: [https://doi.org/10.1016/0012-1606\(85\)90036-3](https://doi.org/10.1016/0012-1606(85)90036-3).

Paynton, B.V. et al. (1988) ‘Changes in state of adenylation and time course of degradation of maternal mRNAs during oocyte maturation and early embryonic development in the mouse’, *Developmental Biology*, 129(2), pp. 304-314. Available at: [https://doi.org/10.1016/0012-1606\(88\)90377-6](https://doi.org/10.1016/0012-1606(88)90377-6).

Su, Y.-Q. et al. (2007) ‘Selective degradation of transcripts during meiotic maturation of mouse oocytes’, *Developmental biology*, 302(1), pp. 104-117. Available at: <https://doi.org/10.1016/j.ydbio.2006.09.008>.

Yu, C. et al. (2016) ‘BTG4 is a meiotic cell cycle-coupled maternal-zygotic-transition licensing factor in oocytes’, *Nature Structural & Molecular Biology*, 23, pp. 387-394. Available at: <https://doi.org/10.1038/nsmb.3204>.

Sha, Q.Q. et al. (2020) ‘Dynamics and clinical relevance of maternal mRNA clearance during the oocyte-to-embryo transition in humans’, *Nature Communications*, 11. Available at: <https://doi.org/10.1038/s41467-020-18680-6>.

○ Comment #9:

What is the proportion of genome-reactivated zygotes? Could this be because the cell numbers are still limited and only a fraction of them exhibited possible reactivation? I would suggest a larger number of cells for this experiment, given the discrepancy in the results.

Response:

The proportion of parthenogenetic zygotes derived from *Kdm5b-yfp* mRNA-injected MII oocytes that exhibited nascent RNA levels above those of the control group is 44.4%. To support this, we include an additional figure here (Response fig. 1, not included in the manuscript) that helps visualise the proportion of reactivated zygotes. We have also added the percentage indication to Fig. 1E in the revised manuscript and added it to the relevant Results section (originally lines 137-141). The revised section is as follows (original submission text in black, new text in blue):

“In the parthenogenetic zygotes derived from MII oocytes injected with *Kdm5b-yfp* mRNA, we found that H3K4me3 was maintained at a similarly low level to uninjected 2C embryos at the time of mid-EGA (Fig. S2I). However, in contrast to zygotes from SN-stage injected oocytes, 44% of the zygotes originating from MII oocytes injected with *Kdm5b-yfp* mRNA displayed elevated levels of nascent RNA (Fig. 1E, right).”

Nascent RNA in 2PN zygotes following oocyte activation after MII injection

Response fig. 1

Nascent RNA signal intensity in 2PN zygotes (bimaternal) following chemical activation after injection of *in vivo*-matured MII oocytes. Total number of samples (summed across two biological replicates): Ctrl water 2PN $n = 9$ and *Kdm5b* 2PN $n = 9$. Each point represents a single pronucleus, and connecting lines link the two pronuclei from the same zygote (PN1 and PN2). The dashed grey line indicates the upper control value to aid visualisation of the proportion of zygotes in the *Kdm5b* group that exhibit genome reactivation.

We fully agree that the number of samples in this experiment is limited, and we acknowledge that larger numbers would have strengthened the conclusions. However, due to resource constraints, we are unable to expand this experiment at this time. Regarding the discrepancy noted, we interpret this to refer to the observation that parthenogenetic zygotes derived from SN-stage injected oocytes did not exhibit genome reactivation, while those from MII-injected oocytes did. As explained in our response to comment #8, we believe this difference is likely due to degradation of *Kdm5b-yfp* transcripts during oocyte maturation, limiting the effectiveness of demethylation at zygote stage in the SN-injected group. Nevertheless, we agree with the reviewer that the sample size should be considered when interpreting the MII-injection findings and have addressed this in our response to comment #10 and 11 below.

○ Comment #10 and 11:

Related to this question, I suggest that "transfer repression to zygotes" in the title (#10) of the paper and "... relieve genome silencing in both oocytes and resulting zygotes" (#11) should be re-considered or removed, as these statements depend on the eventual results and their interpretation.

Response:

We thank the reviewer for this valuable comment. We agree that the original title and

related phrasing in the abstract may have implied a more definitive interpretation than our data can support, particularly given the limited sample size in the relevant experiment. To address this, we have revised the title to better reflect the main conclusions supported by the data, and we have also modified related phrasings. We include below the revised elements with the new additions highlighted (original submission text in black, new text in blue):

- Title of the manuscript (addressing comment #10):
“Broad H3K4me3 domains support oocyte genome silencing and maturation but are dispensable for repression in early embryos”
- Related phrasing in Abstract (original lines 32-33, addressing comment #11):
“Here, we show that inducing broad H3K4me3 removal in oocytes can relieve genome silencing and may result in transcriptional reactivation in resulting zygotes.”
- Related phrasing in Introduction (original lines 90-93):
“Our results indicate that genome silencing promoted by broad H3K4me3 domains is important for oocyte maturation and possibly for transmitting transcriptional repression of the maternal genome to the early zygote. Nevertheless, the repressive state is then maintained independently of the original epigenetic mark until EGA.”
- Related phrasing in Results (original lines 149-150):
“In summary, our results confirm a repressive role for broad H3K4me3 domains in SN-stage oocytes and suggest a potential role for these domains in transferring repression to zygotes.”
- First related phrasing in Discussion (original lines 311-313):
“We found that broad H3K4me3 maintains silencing in oocytes and is necessary for normal oocyte maturation, and may also play a role in transferring transcriptional repression to the early zygote.”
- Second related phrasing in Discussion (original lines 363-364):
“In summary, maintenance of broad H3K4me3 domains in oocytes is required for genome silencing and normal oocyte maturation, and may contribute to transmission of a silenced genome to the early zygote.”
- Comment #12:
 Fig. 1F. Following the comments above, what about zygotes from control and Kdm5b OE MII oocytes (in addition to SN oocytes)?

Response: We agree with the reviewer that the MII oocyte activation data should be included alongside the SN-stage results. Although this experiment was limited in terms of sample size, as the reviewer noted, we acknowledge that the absence of these data was a disadvantage. We have now included the 2PN formation results following activation of *in vivo* matured MII oocytes in the revised Fig. 1F, alongside the GV injection data. The updated panel is shown here for your convenience:

F 2PN zygotes following oocyte activation after GV (6h arrest and IVM) and MII (*in vivo* matured) injection

New Fig. 1F

(F) Results of chemical activation following injection at GV SN-stage (6 h meiotic arrest and 15 -16 h IVM) or at MII stage (*in vivo* matured). Each triangle represents one biological replicate. Bars and error bars: mean \pm s.d., showing mean percentage of 2PN zygotes (bimaternal), based on surviving oocytes. Total number of oocytes (summed across all replicates): Ctrl n = 16 (GV); n = 0 (MII), Ctrl water n = 18 (GV); n = 27 (MII), and *Kdm5b* n = 27 (GV); n = 28 (MII). Pairwise comparisons by Fisher's exact test on pooled counts; significance indicated by P value notation.

We also note that we have updated the figure and figure legend to incorporate additional details requested in your next comment (#13), as well as in one of reviewer 3's comment (#26). Specifically, we now display P value notation from the Fisher's exact test (previously included in the legend but not shown on the plot due to non-significance), individual data points for each replicate (triangles), and the variation between replicates.

• **Comment #13:**

Fig. 1C and 1G. It would be necessary to add quantitative analyses and p-value for these results.

Response: Thank you for pointing this out. We provide separate responses for each of the figure panels below.

Related to Fig. 1C: We now display the p-values from Fisher's exact test, which were previously included in the legend but not shown on the plots due to non-significance. We have also added individual data points for each replicate (triangles) and show variation between replicates, in line with the updated Fig. 1F above and reviewer 3's comment (#26). We also note that we have updated the figure legend accordingly. The updated panel is shown here for your convenience:

New Fig. 1C

(C) Maturation after 15.4 -16.5 h IVM following 6 h meiotic arrest. Each triangle represents one biological replicate. Bars and error bars: mean \pm s.d., showing mean percentage reaching MI and MII stages. Total number of SN oocytes (summed across three replicates): Ctrl n = 36, Ctrl water n = 33, and *Kdm5b* n = 33. Pairwise comparisons by Fisher's exact

test on pooled counts; significance indicated by P value notation.

Related to Fig. 1G: As shown in the updated Fig. 1F (referenced in our response to comment #12), the activation rates varied between replicates. Due to the multiple experimental steps, the number of oocytes available for subsequent analysis was sometimes low. The main experimental steps were:

- GV oocyte SN-stage injection (where lysis can occur)
- 6 h meiotic arrest
- *In vitro* maturation (IVM; with variable success, see new Fig. 1C)
- Chemical activation of MII oocytes (can also result in lysis, especially in IVM oocytes)

To make the best use of the material, we prioritised staining-based analyses (shown in Fig. 1E, and what was originally S1G and S1I - now S2G and S2I). From one replicate with particularly few oocytes surviving activation treatment, however, we chose to culture those that had undergone successful activation (Ctrl n = 1, Ctrl water n = 1, *Kdm5b* n = 2) to the blastocyst stage, resulting in 0, 1, and 2 blastocysts, respectively. Due to the small sample sizes, no statistical analysis was performed. Our intention was simply to demonstrate, using the panel of images, that blastocyst development was possible following GV-stage injection. However, we understand that this may not have been sufficiently clear to readers. To clarify, we have now revised the figure legend, to make our intention more transparent (originally lines 458-461, original submission text in black and new text in blue):

“(G) Development after chemical activation following injection at GV-SN stage, 6 h arrest, and 15 -16 h IVM. Brightfield microscopy images show the 2PN zygotes that developed to blastocysts and were taken at 8.3 h (PNs marked by red circles), 101 h, and 120 h post-activation (scale bar: 100 μm). One replicate with few 2PN zygotes: Ctrl n = 1, Ctrl water n = 1, and Kdm5b n = 2. No statistical analysis due to small sample size.”

Furthermore, we have also updated the relevant paragraph of the Results section (originally lines 146-148) to better reflect the limited sample size and our intent. Please see our response to reviewer 3, comment #27, for further details.

- **Comment #6** (as for reviewer 1 and 3):

It would be important to add *Kdm5b* catalytic dead mutant in the experiments, to confirm that the phenotype is related to H3K4me3 but not non-enzymatic functions.

Response: We agree that including a catalytically inactive *Kdm5b* construct would help determine whether the observed phenotype is due to KDM5B's demethylase activity or potential non-enzymatic functions. In the oocyte experiments, we did not include the catalytically inactive *Del Kdm5b-yfp* construct, but instead used water-injected oocytes as the control. For a more detailed explanation, please refer to our response to Reviewer 1, comment #6.

We would like to note that in our zygote and 2-cell embryo experiments (Fig. 3), embryos injected with *Del Kdm5b-yfp* mRNA showed H3K4me3 levels comparable to uninjected controls, whereas embryos injected with wild-type *Kdm5b-yfp* exhibited a marked reduction in H3K4me3. Although these results are from later developmental stages, they suggest that the phenotype observed is primarily due to the enzymatic activity of KDM5B, rather than non-enzymatic functions.

We appreciate the reviewer's suggestion and agree that future studies would benefit from incorporating a catalytically inactive *Kdm5b* variant in oocyte-stage experiments to further dissect this.

- **Comment #14:**

Fig. S1B. The nascent RNA signals between *Kdm5b* OE and NSN int. were inconsistent with the result in Fig. S1A (right panel).

Response: We agree with the reviewer that the nascent RNA signal shown for the *Kdm5b*-injected SN oocyte and the NSN int. in the original Fig. S1B was not fully consistent with the quantification shown in the original Fig. S1A. We have updated the figure, which has

now been designated Fig. S2B (ref. response under comment #1), to better reflect the representative signal intensities. Specifically:

- In column 3, NSN (+EU), and column 4, NSN int. (+EU), we replaced the images with oocytes whose signal levels are closer to the median nascent RNA values in Fig. S1A (now Fig. S2A).
- Additionally, we updated the cropping to include the entire oocyte, allowing us to add corresponding brightfield images alongside the fluorescence panels.
- We also adjusted the scale bar to 20 μm (previously 10 μm) for consistency and better spatial reference.

We have updated the figure legend accordingly. The updated panel is shown here for your convenience:

New Fig. S2

(B) Representative microscopy images of GV oocytes corresponding to (A). Panel shows H3K4me3 and nascent RNA fluorescence together with brightfield images. Brightness and contrast were adjusted consistently across all samples (scale bar: 20 μm).

- **Comment #15:**

I assume it is probably not feasible to examine the developmental outcome of SN oocytes injected with *Kdm5b* that exhibit increased EU signals, considering that only half of the oocytes displayed such an increase. It would be helpful at least to discuss this possibility in the paper.

Response:

Thank you for this thoughtful point! As noted, only a subset of SN oocytes injected with *Kdm5b-yfp* mRNA showed increased nascent RNA signal, and we agree that assessing the developmental outcome specifically in those reactivated oocytes would be highly informative. However, it was not possible to follow the same individual oocytes from transcriptional assessment after injection through to developmental outcome, due to technical limitations that preclude live tracking of nascent transcription. We did briefly acknowledge this in the submitted discussion (original lines 356-359): “*Additionally, some of our expression data, particularly in immature and activated oocytes, were limited to bulk transcription monitoring rather than transcript level analysis, and we were unable to integrate transcription analysis with development in the same oocyte or embryo.*”

To clarify this point, we have now revised the relevant section in the discussion, and also made an additional related revision under comment #18. The updated text reads (original submission text in black, new text in blue):

“*Additionally, some of our expression data, particularly in immature and activated oocytes, were limited to bulk transcription monitoring rather than transcript-level analysis. Due to technical constraints, we were unable to integrate transcriptional assessment with developmental outcome in the same oocyte.*”

- **Comment #16:**

What is the blastocyst rate of the parthenogenetic zygotes derived from the MII oocytes

injected with Kdm5b overall?

Response: We thank the reviewer for this question and would like to clarify that we did not culture the parthenogenetic zygotes derived from *in vivo* matured, *Kdm5b*-injected MII oocytes to the blastocyst stage. Blastocyst development, as shown in Fig. 1G, was only assessed for one replicate of parthenogenetic zygotes derived from GV oocytes (injected at the SN stage, followed by 6 h meiotic arrest, IVM, and chemical activation).

As noted in our response to comment #13, we recognise that this may not have been entirely clear in the original submission, and we have updated the figure legend for Fig. 1G for clarification.

- **Comment #17:**

What are the products of nascent transcription in SN oocytes upon *Kdm5b* overexpression? Could the authors perform RNA-seq to investigate this?

Response:

We appreciate the reviewer's suggestion and agree that identifying the specific transcripts produced upon *Kdm5b* overexpression in SN oocytes could have further clarified the role of broad H3K4me3 domains at this stage. However, our focus for RNA-seq experiments was directed towards zygotes and 2-cell embryos, as we hypothesised that removal of broad H3K4me3 domains would bring about transcriptional derepression. Due to resource constraints, we are not able to return to the question of transcription in SN oocytes after *Kdm5b* overexpression. Nonetheless, we recognise this as a limitation of our study and have now added a note to the discussion (prior to revisions done to address other comments, and prior to what was originally lines 351-353). The revised text reads (original in black, new addition in blue):

“Regarding study limitations, we have not identified the specific transcripts produced as a result of transcriptional reactivation in SN oocytes. Of note, our quantification of H3K4me3 signal intensity did not account for differences in chromatin configuration between NSN and SN oocytes. Furthermore, while our experimental system provided high temporal resolution of H3K4me3 and global transcription during early development, it did not allow us to determine which genomic regions were depleted of H3K4me3 following KDM5B overexpression.”

- **Comment #18:**

Interestingly, *Kdm5b* OE in the SN oocytes would lead to a defective oocyte maturation rate. However, the underlying mechanism still remains elusive. Would it be possible for the authors to do a "rescue" experiment by treating these oocytes with transcriptional inhibitors to see whether the related phenotype could be alleviated upon inhibition of these ectopic transcription?

Response:

We thank the reviewer for this suggestion. Performing a "rescue" experiment using transcriptional inhibitor(s) to test whether defective maturation can be alleviated is indeed a compelling approach. That said, this may be technically challenging to interpret in our specific context. Even if a transcriptional inhibitor were to rescue the impaired maturation phenotype, it would be difficult to determine whether this effect results specifically from inhibition of ectopic transcription, the absence of transcripts that would otherwise have been produced, or broader effects of the inhibitor.

Nevertheless, we agree that investigating whether there is a relationship between transcriptional reactivation and impaired maturation would be informative, and we have added a brief note about this to the revised Discussion section, which follows our revisions done in the response to comment #15:

“Additionally, some of our expression data, particularly in immature and activated oocytes, were limited to bulk transcription monitoring rather than transcript-level analysis. Due to technical constraints, we were unable to integrate transcriptional assessment with developmental outcome in the same oocyte. Further work is needed to clarify whether genome reactivation at SN stage contributes to the impaired maturation phenotype. Finally, we cannot rule out that overexpression of KDM5B may have led to some non-specific loss of

promoter-associated H3K4me3, usually associated with active transcription. However, the normal EGA and development observed after KDM5B overexpression suggests that this was not the case.”

- **Comment #19:**

The limited function of broad H3K4me3 to ZGA through Kdm5b OE in the zygotes was recently reported (ref. 2). This needs to be discussed and cited.

Response:

Thank you for pointing out this recently published study. It is indeed highly relevant to our work, and we have now incorporated key findings from Zhang et al. (2025) into our updated Discussion section. We include below the revised discussion text with the new additions highlighted (original submission text in black, new text in blue):

- Originally lines 315-319 (without the new, blue text):
“However, we were surprised to find that genome silencing in the late zygote and early preimplantation embryo is not dependent on broad H3K4me3, and premature removal does not impair EGA or early preimplantation development. Similar findings were independently reported in a recent study, which showed that experimental depletion of H3K4me3, by either KDM5B overexpression or KMT2B knockdown, did not markedly alter transcriptional activity or the expression of EGA genes in 2C embryos (Zhang et al., 2025). Taken together, our findings suggest an enigmatic role for broad H3K4me3 domains, where their contribution to genome silencing changes during the course of oogenesis and early preimplantation embryo development.”
- Originally lines 351-356 (without the new, blue text):
“Furthermore, while our experimental system provided high temporal resolution of H3K4me3 and global transcription during early development, it did not allow us to determine which genomic regions were depleted of H3K4me3 following KDM5B overexpression. Although Zhang et al. (2025) defined broad H3K4me3, referred to as non-canonical, differently from our previous study (21,619 peaks ≥ 5 kb vs. 63,542 domains >10 kb in Dahl et al., 2016), their Cleavage Under Targets and Tagmentation (CUT&Tag) analysis of early 2C embryos confirmed that KDM5B overexpression from the zygote stage led to a marked reduction in non-canonical H3K4me3, without inducing premature establishment of a narrow, canonical H3K4me3 pattern. While KDM5B activity has been shown to be important for embryo development, it has remained unclear whether removal of broad domains or reprogramming of promoters to a narrow H3K4me3 pattern is the critical function (Liu et al., 2016; Xu and Xie, 2018). Considered alongside the Zhang et al. (2025) study, our findings point toward a more essential role for the establishment of narrow peaks in the initiation of EGA.”

Minor comments:

- **Comment #20:**

Fig. S2 was missing.

Response: We apologise for the confusion. We had originally chosen to omit a Fig. S2 because there were no supplementary panels corresponding to main Fig. 2, and we thought it would be more intuitive for the supplementary figure numbers to align with the main figures. However, to avoid any further confusion and ensure consistency, we have now revised the supplementary figure numbering so that the figures are numbered consecutively regardless of their relation to the main figures.

- **Comment #21:**

The authors are suggested to present representative figures besides the statistical results where necessary, e.g., Fig. 3 and Fig. S1.

Response: We appreciate the suggestion and agree that representative images enhance the presentation of the results. We have now included representative

immunofluorescence (IF) images corresponding to Fig. 3 in two new supplementary panels, Fig. S3A and S3B. Due to space limitations, these IF images could not be placed directly alongside the graphs in the main figure. The updated Fig. S3 is provided here:

New Fig. S3. H3K4me3 and nascent RNA signal intensities from Click-iT RNA imaging assay (CiRia) with anti-H3K4me3 IF staining, associated with Fig.3.

(A) Representative fluorescence microscopy images of zygotes (mat. PN = maternal pronucleus) and 2C embryos corresponding to Fig. 3B and 3C. Panel shows H3K4me3, YFP, and nascent RNA after background subtraction, with brightness and contrast adjusted consistently across all images (scale bar: 10 μ m).

(B) Representative fluorescence microscopy images of 2C embryos corresponding to Fig. 3D. Panel shows H3K4me3, YFP, and nascent RNA after background subtraction, with brightness and contrast adjusted consistently across all images (scale bar: 10 μ m).

(C) H3K4me3 signal intensity (x-axis) plotted against nascent RNA signal intensity (y-axis), corresponding to Fig. 3B.

(D) H3K4me3 signal intensity (x-axis) plotted against nascent RNA signal intensity (y-axis), corresponding to in Fig. 3C.

(E) H3K4me3 signal intensity (x-axis) plotted against nascent RNA signal intensity (y-axis), corresponding to Fig. 3D.

Regarding what was originally Fig. S1, but have now been designated Fig. S2, we have already maximised the space available on a single page, and given that representative IF images are already included in Fig. S2B, we opted not to add more images. We hope the existing example sufficiently illustrates the relevant findings.

Ref:

1. Wang, J. et al. Reconstitution of chromatin reorganization during mammalian oocyte development. *bioRxiv* (2024).

→ Not related to any comment, so we numbered this comment #22.

Response: We thank the reviewer for including the preprint by Wang et al. (2024). It is indeed interesting that the authors were able to induce the transition to the SN stage through various methods of RNA polymerase II (RNAPII) degradation, but not through simple transcriptional inhibition. Their findings suggest that degradation of chromatin-bound, phosphorylated RNAPII occurs during the NSN to SN transition and that this may act as a driving transitional force *in vivo*. The lower RNAPII levels reported in SN oocytes could potentially explain why we observed only partial transcriptional reactivation, i.e., nascent RNA levels did not reach NSN levels, following *Kdm5b-yfp* mRNA injection at the SN stage in our study.

While the Wang et al. (2024) preprint offers valuable insight into this specific stage of oogenesis, it has not yet undergone peer review. As a general policy, we prefer to cite peer-reviewed literature unless findings are directly relevant or critical for interpreting our own data. Given that this manuscript has not been published, we are reluctant to include a citation, at this stage.

2. Zhang, J. et al. Histone methyltransferases MLL2 and SETD1A/B play distinct roles in H3K4me3 deposition during the transition from totipotency to pluripotency. *EMBO J* 44, 437-456, doi:10.1038/s44318-024-00329-5 (2025).

→ Related to comment #19, where we have addressed this reference.

Reviewer 3

Summary of the advance made in this paper and its potential significance to the field

The epigenome of mouse oocytes is unusual in having a 'non-canonical' pattern of the H3K4me3 histone modification: whereas in most cell types, H3K4me3 is localised at promoters and other cis-regulatory sequences, in mouse oocytes it is far more pervasive and also occupies large genomic domains. These broad H3K4me3 domains are predominantly in untranscribed genomic regions that also have low levels of DNA methylation, which is mutually exclusive with H3K4me3. The functional impact of broad H3K4me3 domains is not fully understood. Past work has indicated a potential and paradoxical role of H3K4me3 deposition in transcriptional repression at the end of oocyte growth (ref. 6, 7). Conversely, an impact on zygotic genome activation (ZGA) has also been inferred, from the findings that ZGA genes are relatively enriched in H3K4me3-marked genomic intervals, and that inhibition of H3K4me3 removal in zygotes by knocking down the H3K4me3 demethylases KDM5A & B impairs ZGA (previous work from this group; ref. 5).

The current study represents a re-evaluation of the functional impact and a refinement in understanding of oocyte H3K4me3 broad domains. It comprises an elegant set of experiments that largely focus around the use of forced expression of KDM5B in oocytes, zygotes or 2-cell embryos, coupled with immunofluorescence analysis, to interrogate when broad H3K4me3 domains constitute a block to transcription of the genome. In essence, the authors find that ablation of bulk H3K4me3 in transcriptionally quiescent GV oocytes derepresses transcription (consistent with earlier findings), while precocious removal of H3K4me3 in zygotes or early 2C embryos does not induce premature embryonic genome activation (EGA) or qualitatively affect EGA. Consistent with this finding, precocious removal of H3K4me3 is compatible with normal progression of preimplantation development. Figure 3E summaries the key outcomes. The absence of an effect of precocious H3K4me3 removal is unexpected and novel. On the other hand, the study does not really advance the mechanistic understanding of the role of broad H3K4me3 in transcriptional repression.

Response: We thank the reviewer for the clear and constructive summary, and for recognising the novelty of our findings. We appreciate the positive comments on our experimental approach

and agree that while the mechanistic basis remains to be explored, the study offers important clarification of the functional timing of broad H3K4me3 domains.

Suggestions to authors:

The authors provide an informed discussion that recognises some of the limitations of their study. Amongst these is the fact that detection of genomic H3K4me3 in oocytes and embryos in the current study is entirely by immunofluorescence, such that global effects are measured. The authors provide the important caveat (lines 355-357) that "it is not clear whether removal of broad domains or reprogramming of promoters to a narrow H3K4me3 pattern is the critical function." Further in this regard, although the transcriptional reactivation observed upon expression of KDM5B in SN-stage GV oocytes is compelling, it remains an inference that broad H3K4me3 domains themselves contribute to transcriptional repression in oocytes.

Response: We thank the reviewer for these insightful observations. We agree and have addressed this further by incorporating relevant findings from the recent Zhang et al. (2025) study, as highlighted by reviewer 2, into our updated Discussion. Please see our response to comment #19 for further details.

- **Comment #23:**

The authors should perhaps note that while broad H3K4me3 domains have been described in mouse, rat, bovine and porcine oocytes, they are not present in human oocytes (PMID: 31273069; 34818044). Therefore, whatever functional impacts H3K4me3 broad domains have on transcriptional repression in the oocyte and zygote may not apply universally in mammalian oocytes.

Response:

We thank the reviewer for raising this important point. It is true that a prior study has found that human H3K4me3 dynamics differ from those in mouse, starting with canonical, promoter-associated H3K4me3 and an absence of broad domains in GV oocytes (it is unclear whether these were NSN or SN stage, but most likely they were SN oocytes since they were obtained during fertility treatment). It was further shown that in human 4-cell embryos, i.e., pre-EGA, the H3K4me3 signal broadens and increases, including at distal sites. During EGA at the 8-cell stage, *Kdm5b* transcript levels were found to increase and broad domains were largely removed, with H3K4me3 retained at a subset of active promoters (Xia et al., 2019).

However, mature MII oocytes, zygotes, and 2-cell embryos were not assessed for H3K4me3 in that study, leaving an incomplete picture. As noted by Xia et al., certain aspects of the human dynamics may still resemble the reprogramming trajectory seen in mouse embryos. Thus, we believe that studying broad H3K4me3 in the mouse remains valuable for uncovering general principles of silencing, reprogramming, and genome activation to help us understand what may be relevant, or different, in a human context. We chose not to directly address human oocytes in the original submission, given the lack of human data and word limitations. However, we are currently preparing a separate manuscript comparing H3K4me3 in oocytes of several species, including human. These results indicate that broad H3K4me3 is already established in the human zygote and appears to be pre-defined at certain regions in mature MII oocytes, albeit with narrower characteristics than those observed in mouse.

That said, we acknowledge that the point raised by the reviewer is worth including in this manuscript and have added a couple of sentences to the revised Introduction (originally lines 47-53; original submission text in black, new text in blue):

“Remarkably, mouse metaphase II (MII) oocytes exhibit a unique broad H3K4me3 pattern, which covers 22% of the oocyte genome (Dahl et al., 2016; Zhang et al., 2016). In contrast to the frequent association of H3K4me3 with active genes, these broad H3K4me3 domains have been proposed to promote global genome silencing in oocytes, and contribute to acquisition of meiotic and developmental competence (Andreu-Vieyra et al., 2010; Zhang et al., 2016). Similar broad domains have been identified in bovine, porcine, and rat oocytes, but notably, they have not been observed in immature human germinal vesicle (GV)

oocytes (Lu et al., 2021; Xia et al., 2019). Nonetheless, human post-fertilisation dynamics show shared features of H3K4me3 reprogramming, and mouse studies remain useful for understanding transcriptional silencing and genome activation. Global genome silencing is observed during late oocyte growth in mouse, and coincides with, but is not dependent on, the transition of the GV nucleus from a loosely packed non-surrounded nucleolus (NSN) to a more condensed surrounded nucleolus (SN) chromatin configuration (Bouniol-Baly et al., 1999; De La Fuente et al., 2004; Debey et al., 1993)."

References cited in this response:

Lu, X. et al. (2021) 'Evolutionary epigenomic analyses in mammalian early embryos reveal species-specific innovations and conserved principles of imprinting', *Science Advances*, 7(48), p. eabi6178. Available at: <https://doi.org/10.1126/sciadv.abi6178>.

Xia, W. et al. (2019) 'Resetting histone modifications during human parental-to-zygotic transition', *Science*, 365(6451), pp. 353-360. Available at: <https://doi.org/10.1126/science.aaw5118>.

- **Comment #10 (as for reviewer 2):**

In view of the ambiguity of whether the effects of H3K4me3 on transcriptional repression in oocytes are mediated by broad domains or other genomic sites of H3K4me3, there is an argument that the title should be changed, as the authors cannot conclusively show involvement of broad domains in the molecular phenotypes.

Response: We appreciate the reviewer's point and agree that caution in wording is warranted. We have addressed this concern under reviewer 2's comments #10 and #11, where we revised the title. Additionally, in light of the recent publication by Zhang et al. (2025), which used H3K4me3 CUT&Tag profiling to show that *Kdm5b* overexpression from the zygote stage leads to extensive loss of H3K4me3 in broad domains (non-canonical), there is now stronger support for a direct connection between KDM5B-induced demethylation and broad domain loss. As noted in earlier responses, this study has also been discussed and cited in our revised Discussion section, in response to reviewer 2's comment #19.

Reference cited in this response:

Zhang, J. et al. (2025) 'Histone methyltransferases MLL2 and SETD1A/B play distinct roles in H3K4me3 deposition during the transition from totipotency to pluripotency', *The EMBO Journal*, 44(2), pp. 437-456. Available at: <https://doi.org/10.1038/s44318-024-00329-5>.

- **Comment #24:**

In relation to the inability of forced removal of bulk H3K4me3 to induce precocious EGA in zygotes or early 2C embryos, the authors offer some possibilities (lines 336-350), but ultimately conclude that "the mechanisms that maintain genome silencing in late zygotes and early 2C embryos in the absence of broad H3K4me3 remain elusive." The authors should note that broad H3K4me3 domains are confined to the maternal genome, so the paternal genome is maintained in a quiescent state (excepting minor ZGA) in any case in the absence of broad H3K4me3 domains. Again, this fact should be recognised in the discussion.

Response:

We appreciate the reviewer's comment and agree that broad H3K4me3 domains are established during oogenesis and are consequently most pronounced in the maternal genome following fertilisation. However, we would like to clarify that a previous study has also reported the emergence of broad H3K4me3 domains, referred to as non-canonical H3K4me3 (ncH3K4me3), on the paternal genome in zygotes, although at lower levels than in the maternal (Zhang et al., 2016). This is consistent with the idea that maternal stores of H3K4 methyltransferases can act on the paternal genome.

Specifically, Zhang et al. (2016) write:

"Using a highly sensitive ChIP-seq method, we investigated genome-wide H3K4me3 in

mouse early development at the nucleosome resolution. Our data unveiled surprisingly dynamic chromatin landscapes in gametes and early embryos characterized by the presence of nH3K4me3 (Fig. 4d). The non-canonical H3K4me3 pattern appears to be inherited from oocytes to early embryos before ZGA. Extensive reprogramming also occurs on the paternal genome in zygotes where H3K4me3 peaks are largely depleted. Instead, very broad H3K4me3 domains at weak levels were observed which could be considered as a type of paternal non-canonical H3K4me3 (Fig. 4d)."

The above ChIP-seq findings are consistent with an earlier immunofluorescence study showing that paternal H3K4me3 becomes detectable around 8-10 hours after fertilisation and continues to rise, reaching levels comparable to the maternal genome by approximately 12 hours post-fertilisation (Lepikhov and Walter, 2004).

Further support for the presence of broad H3K4me3 in the paternal pronucleus at the zygote stage comes from our own data, where we observed similarly low H3K4me3 levels in both pronuclei following *Kdm5b* overexpression (Fig. 3B). However, as no transcriptional reactivation was observed in either pronucleus under these conditions, we opted not to discuss paternal H3K4me3 dynamics separately in the manuscript. We continue to believe this was appropriate, but we thank the reviewer for prompting this clarification.

References cited in this response:

Lepikhov, K. and Walter, J. (2004) 'Differential dynamics of histone H3 methylation at positions K4 and K9 in the mouse zygote', *BMC Developmental Biology*, 4(12), pp. 1-5. Available at: <https://doi.org/10.1186/1471-213X-4-12>.

Zhang, B. *et al.* (2016) 'Allelic reprogramming of the histone modification H3K4me3 in early mammalian development', *Nature*, 537, pp. 553-557. Available at: <https://doi.org/10.1038/nature19361>.

Specific comments to address:

- **Comment #25:**
Line 120: "*Kdm5b-yfp* mRNA injection and 6 hours arrest was sufficient to reduce H3K4me3 to NSN levels". Given the very different chromatin distribution of NSN compared to SN oocytes, how are the authors able to make a comparison in H3K4me3 IF intensity between NSN and SN? It is notable that there is a huge variation in H3K4me3 signal intensity particularly in NSN oocytes (Fig. S1E). The Methods section does not give much detail about how IF intensity was measured and how differences in parameters such as chromatin compaction were taken into account.

Response:

We thank the reviewer for this important point. In this manuscript, we are not specifically seeking to make claims regarding the relative levels of H3K4me3 after *Kdm5b* overexpression in one stage, relative to controls of *other* stages. Rather, we use uninjected, Del *Kdm5b* mRNA-injected, and *Kdm5b* mRNA-injected oocytes/zygotes/embryos of specific stages for direct comparison. The effect of overexpression of *Kdm5b* on H3K4me3 levels in different stages is compared as a ratio of reduction, rather than as absolute values of H3K4me3 in Fig. 3E. It is nevertheless useful to consider if *Kdm5b* overexpression produces a reduction in H3K4me3 to levels that are compatible with transcription in similar stages. For example, we compare H3K4me3 in normally transcriptionally repressed early 2C after *Kdm5b* overexpression to transcriptionally active mid 2C EGA embryos in Fig. 3C, and find that *Kdm5b* overexpression leads to lower levels of H3K4me3 than that permissive of EGA.

We agree that the chromatin configuration differs substantially between NSN and SN oocytes, and ideally, this would be accounted for during imaging and quantification. As noted in our response to reviewer 1's comment #7, DNA staining was intentionally omitted in these GV-stage experiments to maximise spectral separation and extract the

maximum amount of information from the samples. Specifically, an Alexa Fluor 350-conjugated secondary antibody was used to detect H3K4me3, allowing simultaneous detection of both EU-Alexa Fluor 594 (nascent RNA) and YFP (the KDM5B fluorescent tag) without spectral overlap. As a result, we applied a widely used approach for quantifying IF signal intensity, as detailed in our Materials and Methods section (originally lines 757-761): “Images were taken with a 63X oil immersion objective in the central plane of each (pro)nucleus. Fluorescence intensities were measured in ZEN Blue v.2.6 software (ZEISS). For quantifying nascent RNA and H3K4me3 in the cells, the mean intensity of a similar sized region in the cytoplasm was subtracted from the mean intensity of the nucleus.”

If an accurate comparison of H3K4me3 in *Kdm5b* overexpressed oocytes/embryos with control oocytes/embryos of other stages had been required, we could have performed a separate round of experiments that included DNA staining and omitted nascent RNA quantification, as described for MII oocytes. That said, we acknowledge that our approach does not account for chromatin configuration differences between NSN and SN oocytes and we recognise this as a limitation. We have therefore included this in the Discussion section of the revised manuscript (following previously revised text, and with original lines 351-353 after). The revised text reads (original in black, new addition in blue):

“Regarding study limitations, we have not identified the specific transcripts produced as a result of transcriptional reactivation in SN oocytes. Of note, our quantification of H3K4me3 signal intensity did not account for differences in chromatin configuration between NSN and SN oocytes. Furthermore, while our experimental system provided high temporal resolution of H3K4me3 and global transcription during early development, it did not allow us to determine which genomic regions were depleted of H3K4me3 following KDM5B overexpression.”

- **Comment #6 (as for reviewer 1 and 2):**

Fig. 1C/1D: test the effect of *Kdm5b*-mRNA injection on IVM rates. In these experiments, the controls presented are uninjected oocytes or oocytes injected with water. These are not ideal controls, as they do not assess the effect of injection of a reagent, which could have off-target effects in addition to the desired target. Better controls would be injection of an irrelevant mRNA, scrambled mRNA, mRNA unable to express the protein, or mRNA encoding a catalytically inactive KDM5B protein. Elsewhere in the study, the authors have used an mRNA for a catalytically-dead KDM5B (e.g., Fig. 3B, 5). The authors do explain the preference for use of water as control over the use of catalytically-dead *Kdm5b*-mRNA (lines 635-637), which might be justified, but an alternative mRNA as indicated above would be preferable. A better experimental strategy would have been to perform the first set of experiments (as in Figs. 1C, D) with a control injected mRNA and once having established that the control mRNA had an outcome indistinguishable from water injection, subsequent experiments would be justified in using water as control.

Response: We thank the reviewer for this valuable suggestion. As noted in our responses to reviewer 1 and reviewer 2 (both labelled comment #6), we opted for water injected controls because our preliminary data showed that the catalytically inactive *Kdm5b* construct, *Del Kdm5b-yfp*, had specific biological effects, making it unsuitable as an inert control. Water injection allowed us to control for any effects of the injection procedure itself, while also providing a consistent baseline using the same buffer used for mRNA dilution. Had we observed a difference in RNA-seq between the *Kdm5b-yfp* mRNA and water injected embryos, inclusion of an alternative control mRNA would have been beneficial, and we appreciate the reviewer’s recommendation for future studies.

- **Comment #26:**

Fig. 1C is a bar chart showing the IVM rates in control, control injected or *Kdm5b*-mRNA injected GV oocytes. The figure legend gives the number of manipulated oocytes per group, and states that three replicates were carried out. If this is so, does this mean 3 replicates each of 36, 33 and 33 in the three groups, or that there were 3 replicates totalling 36, 33 and 33 in the three groups? Also, given that 3 replicates were carried out, the bar chart should indicate the variation between replicates. This comment would apply to other cases in which replicate experiments were performed, so the meaning is fully transparent.

Response: The reviewer raises an important point regarding clarity and transparency. To address this, we have updated Fig. 1C, as well as other relevant figure panels (Fig. 1F, and originally Fig. S1C that is now Fig. S2C), to include individual data points for each replicate (represented as triangles) and to show variation between replicates with error bars. The updated figure legends now also clarify that the total number of oocytes reported reflects the sum across all replicates. We would also like to mention that there is variation between replicates, both in figure 1C and 1F, due to the different sizes of replicates and the number of experimental steps required, as discussed in the response to reviewer 2's comment #13.

In addition, as noted in our response to comment #13, we now display P value notation from Fisher's exact test directly on the plots. These values were previously reported in the legends but not shown graphically due to non-significance. For reference, the updated Fig. 1C is shown below as an example; the same adjustments have been applied to Fig. 1F and Fig. S1C (now Fig. S2C).

New Fig. 1C

(C) Maturation after 15.4 -16.5 h IVM following 6 h meiotic arrest. Each triangle represents one biological replicate. Bars and error bars: mean \pm s.d., showing mean percentage reaching MI and MII stages. Total number of oocytes (summed across three replicates): Ctrl SNs $n = 36$, Ctrl water SNs $n = 33$, and *Kdm5b* SNs $n = 33$. Pairwise comparisons by Fisher's exact test on pooled counts; significance indicated by P value notation.

- **Comment #27:**

Lines 146-148: "To further assess developmental competence, we cultured parthenogenetic zygotes to day 5 and found that a reduction of H3K4me3 at SN stage was also compatible with development to blastocyst (Fig. 1G)." This statement needs to be supported by an indication of the numbers of oocytes manipulated and activated, and the percentages reaching the blastocyst stage. In comparison in Fig. 5 the authors do properly quantify developmental progression of embryos injected at the 2C stage.

Response: We appreciate this feedback and have taken the following steps to address it. As noted in our response to reviewer 2, comment #13, the number of oocytes that had undergone successful activation and were cultured to day 5 in this experiment was low (Ctrl $n = 1$, Ctrl water $n = 1$, *Kdm5b* $n = 2$), all from a single replicate with limited maturation and activation survival. We have clarified this in the revised figure legend - please see our response to comment #13, under the section called "RelatedtoFig.1G", for further details.

Given the limited sample size, we do not consider it appropriate to report percentages for this experiment. However, in light of this comment #27, we have updated the Results section (lines 146-148) to more accurately reflect the scope and intent of the data. We include the revised main text below, with the new addition highlighted (original submission text in black, new text in blue):

"To further assess developmental competence, we cultured parthenogenetic zygotes to day 5 and found that a reduction of H3K4me3 at SN stage was also compatible with development to blastocyst (Fig. 1G). *Based on a small number of zygotes from one replicate, this observation illustrates developmental potential rather than providing*

quantitative results.”

- **Comment #28:**
Fig. 2C: the data for *Kdm5b* mRNA from the RNA-seq of zygotes and early 2C embryos are consistent with *Kdm5b* being transcribed as part of the minor phase of ZGA; they could also indicate preservation of maternal *Kdm5b* mRNA during the phase of global maternal RNA degradation. Although not directly addressing this difference, it would be useful to complement these data with IF from KDM5B in zygotes and 2-cell embryos to demonstrate timing of appearance of KDM5B protein.

Response: We are grateful for this helpful input. We agree that our RNA-seq data showing increased *Kdm5b* mRNA levels in early 2C embryos could theoretically reflect either transcriptional activation during minor ZGA or stabilisation of maternal transcripts. However, previous RNA-seq studies have reported minimal maternal *Kdm5b* expression in MII oocytes - undetectable in Dahl et al. (2016, Extended Data Fig. 9a) and very low in Dang et al. (2022, Fig. S7E). This supports our interpretation that *Kdm5b* is transcriptionally upregulated after fertilisation. Further support for ZGA-driven *Kdm5b* expression comes from Liu et al. (2020, Fig. 4b), who show RNA polymerase II binding at the *Kdm5b* locus already in zygotes, with increased occupancy in early and late 2-cell embryos.

At the protein level, KDM5B has been detected in 2-cell embryos by both IF (Shao et al., 2014, Fig. 4) and western blot (Dahl et al., 2016, Extended Data Fig. 9d). We agree that future investigations could usefully determine whether the protein becomes detectable already at the zygote stage or during the early 2-cell stage, as suggested by both the early onset of endogenous H3K4me3 removal (Fig. 2A) and our RNA-seq data (Fig. 2C). However, we believe this question lies somewhat outside the primary scope of our current study, and therefore we have not included KDM5B IF data.

That said, to reflect this comment and the studies discussed above, we have added a new sentence in the Results section (originally lines 176-183) to clarify the interpretation. We include the revised main text below, with the new addition highlighted (original submission text in black, new text in blue):

“To determine the exact timing of transcription and whether the H3K4 demethylase mRNAs originated from the minor genome activation, we quantified transcript levels in late zygotes and early 2C embryos by RNA-seq. We found that low levels of *Kdm5a* and *Kdm5b* mRNA were present in late zygotes (17 hours after ovulation), followed by a substantial increase in *Kdm5b*, but not *Kdm5a*, transcripts at the early 2C stage (5.5-8 hours after division) prior to EGA (Fig. 2C). This suggests that *Kdm5b* transcription in the early 2C embryo, after minor genome activation, initiates H3K4me3 removal before EGA onset. *Consistent with this, previous studies have shown minimal *Kdm5b* expression in MII oocytes, as well as RNA polymerase II (RNAPII) binding at the *Kdm5b* locus and protein detection in 2C embryos, supporting transcriptional activation, rather than maternal mRNA stabilisation, as the source of KDM5B after fertilisation (Dahl et al., 2016; Dang et al., 2022; Liu et al., 2020; Shao et al., 2014).*”

References cited in this response:

Dahl, J.A. et al. (2016) ‘Broad histone H3K4me3 domains in mouse oocytes modulate maternal-to-zygotic transition’, *Nature*, 537, pp. 548-552. Available at: <https://doi.org/10.1038/nature19360>.

Dang, Y. et al. (2022) ‘The lysine deacetylase activity of histone deacetylases 1 and 2 is required to safeguard zygotic genome activation in mice and cattle’, *Development*, 149(11), pp. 1-14. Available at: <https://doi.org/10.1242/dev.200854>.

Liu, B. et al. (2020) ‘The landscape of RNA Pol II binding reveals a step-wise transition during ZGA’, *Nature*, 587, pp. 139-144. Available at: <https://doi.org/10.1038/s41586-020-2847-y>.

Shao, G.-B. et al. (2014) ‘Dynamic patterns of histone H3 lysine 4 methyltransferases and demethylases during mouse preimplantation development’, *In Vitro Cellular &*

Developmental Biology - Animal, 50(7), pp. 603-613. Available at: <https://doi.org/10.1007/s11626-014-9741-6>.

- **Comment #1 (as for reviewer 1):**

In general, evidence of 'exogenous' KDM5B expression in relation to timing of appearance or level of the native protein would be useful, although this may not be feasible depending upon availability of suitable antibodies. The authors may be justified in responding that abundance of the target modification H3K4me3, which they have quantified extensively, is what really matters.

Response: We acknowledge the relevance of this suggestion and have addressed it in our response to reviewer 1 under comment #1. We would like to direct you there for further details.

- **Comment #29:**

Discussion lines 332-333: it is not clear how the authors are able to conclude "Finally, SN oocytes undergoing H3K4me3 depletion display a lag of several hours before increased transcription is detected", because there did not appear to be any figure that showed a time-course of transcriptional derepression following H3K4me3 depletion.

Response:

We thank the reviewer for this observation and agree that the original phrasing may be misleading. The statement was not meant to imply that a time-course experiment was performed. Rather, it refers to the comparison between two different experimental time points: 6 hours and 12-14 hours of meiotic arrest following *Kdm5b* mRNA-injection at SN stage. As described in our Results section, we observed no transcriptional reactivation after 6 hours arrest (original lines 120-122, Fig. S1E; now Fig. S2E), despite effective H3K4me3 reduction, whereas 44% of oocytes showed elevated RNA levels following 12-14 hours arrest (original lines 112-113, Fig. 1B). This suggests that a delay of several hours may occur between the reduction in H3K4me3 and the onset of transcriptional reactivation in oocytes.

To avoid misinterpretation, we have revised the sentence in the discussion accordingly (originally lines 332-333). The updated sentence now reads (original submission text in black, new text in blue):

"Finally, SN oocytes undergoing H3K4me3 depletion display a lag of several hours before increased transcription is detected, evident from our observation of reactivation after 12-14 hours of arrest but not after 6 hours."

- **Comment #30:**

In the Methods (lines 837-840) the authors explain that the RNA-seq libraries from 2C embryos comprised both single 2C embryos and various small pools of 2C embryos with damaged blastomeres. It might be useful to highlight on the PCA (Suppl Fig. S4A) the single 2C embryos and the 2C embryo pools, in case the nature of the 2C embryos contributes to variation amongst the libraries.

Response: We thank the reviewer for this helpful suggestion. First, we would like to clarify that the sample pools did not contain "damaged blastomeres." As explained in the Materials and Methods section (originally lines 835-837), only surviving blastomeres were collected for RNA-seq. Damaged or lysed blastomeres, typically resulting from the injection procedure, were excluded during the zona pellucida removal step prior to sample collection.

As part of our initial quality control checks, we also assessed whether pooled samples introduced variation in the RNA-seq data and found that they clustered closely with single-embryo samples within each group. Nevertheless, we agree that explicitly showing this distinction in the PCA plot improves transparency.

We have now updated Fig. S4A to distinguish between single 2C embryos and pooled 2C samples comprising single surviving blastomeres, where pooled samples are encircled or marked by a square. As detailed in the Methods (originally lines 837-840), the *Kdm5b* 2C group included five single embryos and one pool of three blastomeres from different embryos, while the Ctrl water

2C group included six single embryos, two pools of three blastomeres, and one pool of two blastomeres from different embryos. The updated PCA demonstrates that sample type does not account for the observed variation. The figure legend has been updated accordingly. The revised Fig. S4A is provided here:

New Fig. S4. Further analyses of RNA-seq data obtained as described in Fig. 4A.

(A) Principal component analysis (PCA) of RNA-seq data from all samples: Ctrl water zygotes ($n = 10$), *Kdm5b* zygotes ($n = 10$), Ctrl water 2C ($n = 9$), and *Kdm5b* 2C ($n = 6$). Each dot represents one sample. Pooled 2C samples consisting of three blastomeres from different embryos are encircled; the pool of two blastomeres is marked by a square. Samples without markings represent single zygotes or single 2C embryos with both blastomeres intact (see Materials and Methods for details).

Additional changes made to the manuscript

We also wish to note that, in preparing the revised version, we made improvements to the manuscript's clarity and presentation, and have adjusted the formatting to comply with *Development's* guidelines. In the manuscript, all new text is highlighted in blue, while original submission text is black. The following changes were made:

- Updated the title page to required format.
- Added a summary statement following the abstract.
- Updated the citation style to the Harvard referencing system. As a result, line numbering has shifted significantly compared to the original submission, which used the numerical Vancouver referencing style.
- Optimised the layout of figure panels and overall figure sizes.
- Revised figure legends to improve clarity and conciseness, and to ensure inclusion of all required information in line with reviewers' comments and journal guidelines. This goes for both the main figure legends and the supplementary ones.
- Updated parts of the Materials and Methods section, especially the "Statistical analyses and data presentation" subsection.

Second decision letter

MS ID#: dev.204638R1

MS TITLE: Broad H3K4me3 domains support oocyte genome silencing and maturation but are dispensable for repression in early embryos

AUTHORS: Trine Skuland, Madeleine Fosslie, Sherif Khodeer, Endalkachew Ashenafi Alemu, Jens Vilstrup Johansen, Marie Indahl, Blanca Corral Castroviejo, Yanjiao Li, Mads Lerdrup, John Arne Dahl, Peter Zoltan Fedorcsak and Gareth D. Greggains

Dear Dr Greggains,

I have now received all the referees reports on the above manuscript, and have reached a decision. The referees' comments are appended below, or you can access them online: please go to .

There is support for the study, but also concerns about controls and small sample sizes. As you mentioned in your cover letter that you are unable to carry out additional experiments, the referees and I have discussed whether there is a productive way forward. If you are able and willing to revise the manuscript along the following lines, then I would be happy to evaluate it:

Required:

In relation to the question of appropriate controls for the injections. Please add the detailed justification/explanation that you provided in your response into the Methods section of the manuscript so that it is transparent to readers.

In relation to the data and statements derived from Figures 1F and 1G. The sample size is insufficient and too preliminary to include. Please remove these data, and any statements, including in the abstract, specifically relating to these findings should also be removed from the manuscript.

Reviewer 1's comments about Figure 1C and the description of the parthenogenetic embryos can be addressed with changes to the manuscript text.

Optional:

If comments 1-3 of Reviewer 2 can be addressed, either fully or in part, then please do so.

Reviewer 3 mentioned a recent relevant publication and the implications for the current work.

If you do not agree with these suggestions, please contact me with an explanation.

Reviewer 1*Advance summary and potential significance to field*

The authors have addressed most of my comments. However the authors haven't addressed comment 6 (injection control) properly since all three reviewers have raised the same concern. As well as Comment 13 (Fig 1G), I think sample size between 1-2 is sufficient to report. The authors should either increase the number to sufficient statistical power or delete it. For abstract, since there is no statistical difference on oocyte maturation between groups (Fig 1C), it is not suitable for reporting "impaired oocyte maturation".

Minor comemnts:

To avoid confusion for describing parthenogenetic 1-cell embryos, it will be suitable to use parthenotes or parthenogenetic 1-cell embryos, not zygotes. For PA embryos, they don't have 2PN but pPN (pseudopronuclei).

In FigS2B, the signal of H3K4me3 in NSN int (+EU) looks strange.

Reviewer 2

Advance summary and potential significance to field

I thank the authors for their response and revision which well addressed part of my previous questions. On the other hand, several key questions remain unaddressed in the revision (see below), even though many of the related experiments seem to be straightforward and require only the existing experiment conditions in the paper. In addition, several figures lack sufficient samples which make the data inclusive or unconvincing (detailed below). As a result, I believe that it would be premature to publish the current version of manuscript. Please see below for the specific comments.

1. Comment #8, the authors proposed that maternal RNA decay could help explain the difference between Kdm5b OE in SN or MII oocytes. This idea in principle can be easily tested. Could the authors show the H3K4me3 and YFP IF for the zygotes from Kdm5b OE SN oocytes? This may help validate their hypothesis.
2. Comment #9, the authors only calculated the nascent RNA intensity in 9 zygotes which are too few to reach a conclusive conclusion.
3. Comment #16. It was reported that KDM5B OE in mouse MII oocytes followed by IVF resulted in preimplantation arrest (PMID: 40449591). Since the authors have also overexpressed KDM5B in MII and found that parthenogenetic zygotes originating from these oocytes exhibited elevated levels of nascent RNA, it would greatly strengthen the paper if the authors can examine the blastocyst rate of these derived embryos.
4. Revised Fig. 1F. The results showed huge variations among replicates, with one replicate show nearly 0% and the other replicate showing 80%, making the results nearly impossible to evaluate. It would be premature to conclude based on these data. The authors are advised to include more replicates or remove related statements/data based on these results.
5. Fig. 1G. One replicate with only 1-2 embryos is far from being sufficient for evaluation of the developmental competency of the resultant embryos. Such data should not be used to draw conclusions and removed. More replicates with substantially more embryos are needed to make the conclusions of the paper convincing.

Reviewer 3

Advance summary and potential significance to field

I welcome the extensive efforts the authors have made to provide greater clarity of their experimental design, choices and analyses, and in providing greater transparency in their results, e.g., in relation to sample numbers and replicates, etc. Their responses to reviewers' comments are universally well argued; the edits and additional information provided appropriate and adequate.

Comments for the author

It has very recently been reported that non-canonical H3K4me3 and H2A.Z deposition in broad domains in mouse oocytes is mutually reinforcing, and that depletion of nH2A.Z by knocking out the two H2A.Z-encoding genes (H2afv, H2afz) in oocytes phenocopies many of the phenotypes of the MLL2-KO (PMID: 40514539 - although an effect on transcriptional silencing was not explicitly tested in that study). That study helps emphasize the exquisite dependencies of histone marks, and

variants, in oocytes (both positive and negative), where altering one mark will undoubtedly lead to changes in others. Against this background, mechanistic inferences are challenging, if the mechanistically relevant mark is not what is measured in any particular study. On the other hand, in the current manuscript, the authors may be justified in identifying broad H3K4me3 as being responsible for enforcing transcriptional silencing in GV oocytes, rather than a mark/variant dependent on H3K4me3, because the intervention deployed (KDM5B over-expression for a limited period rather than a genetic knock-out over the oocyte growth phase) may limit the effect to H3K4me3 without incurring further changes in chromatin states. The authors might like to reflect this point.

Second revision

Author response to reviewers' comments

Opening remarks from the authors

Thank you very much for taking the time to evaluate our revised manuscript and for providing us with further constructive comments to improve the work. We greatly appreciate your thoughtful feedback and hope that this final version addresses your remaining concerns and meets the standards for publication.

As before, we have included your original comments (black text) and provided our point-by-point responses (red text). For reference, we have numbered the comments as a continuation from our previous response letter.

Reviewer 1

The authors have addressed most of my comments.

- **Comment #31:**
However the authors haven't addressed previous comment #6 (injection control) properly since all three reviewers have raised the same concern.

Response: We appreciate the reviewer's continued attention to this point and hope that our rationale remains understandable. As suggested by the Handling Editor, we have now incorporated the detailed justification previously provided in our first response letter into the Materials and Methods section to ensure full transparency for readers. The justification has been lightly revised to create a streamlined and consolidated version of our previous responses to all three reviewers regarding this concern. The justification we have added is as follows (new incorporation in green text - also in green in the revised manuscript v2):

“Control strategy for microinjection experiments

Our primary experimental strategy involved injecting mRNA encoding a catalytically inactive version of KDM5B, Del Kdm5b-yfp, in which the Jumonji C domain (essential for histone demethylase activity) was deleted. Data using this strategy on zygotes and embryos are presented in Fig. 3 and Fig. 5, alongside uninjected controls and wild-type Kdm5b-yfp mRNA-injected groups. These results indicated a tendency toward higher H3K4me3 levels in zygotes injected with Del Kdm5b-yfp mRNA, as well as a possible inhibitory effect on development. This is why we noted in the final paragraph of the Results section: “We suspect that Del KDM5B proteins may persist after binding to broad H3K4me3 domains due to the absence of catalytic activity. This may interfere with timely H3K4me3 reprogramming by endogenous KDM5B and affect developmental progression, as previously shown (Dahl et al., 2016; Liu et al., 2016).” Based on these observations, we later opted to use water-injected controls (Fig. 1 and Fig. 4), matching the injection

buffer used for mRNA dilution, instead of *Del Kdm5b-yfp* injection, to avoid introducing additional effects potentially caused by overexpression of the catalytically inactive variant.

Importantly, we observed no evidence of an overexpression side effect: RNA-seq analysis (Fig. 4) revealed no significant differences in expression between water- and mRNA-injected zygotes or 2-cell embryos, and *Kdm5b-yfp*-injected embryos developed normally (Fig. 5). Had we observed significant differences between the groups at these stages, we acknowledge that a control mRNA encoding YFP alone would have been warranted to rule out general overexpression effects. That said, such a control would not be equivalent in size (Yfp: 726 nt; *Kdm5b-yfp*: 5471 nt), and injection of fluorescently tagged proteins is widely used for overexpression in oocytes and embryos, with minimal reported side effects.

It is also worth noting that different strategies have been employed to study *Kdm5b* overexpression. For example, Zhang et al. (2025) used a point mutation (H499A) to inactivate the catalytic function of KDM5B while retaining the full-length protein, enabling distinction between enzymatic and non-enzymatic roles of the protein in early 2-cell embryos. Our approach, deletion of the entire Jumonji C domain, also renders KDM5B catalytically inactive but may affect additional structural aspects of the protein, thus providing a complementary perspective. Furthermore, Zhang et al. (2016) performed *Kdm5a* and *Kdm5b* overexpression in SN oocytes but did not specify the nature of their control, highlighting the broader challenge of control selection in this system.

While we recognise that a control mRNA encoding YFP alone may have added an additional layer of clarity, we believe our choice of water-injected controls was justified based on the observed effects of the *Del Kdm5b-yfp* construct and the limitations of alternative strategies. Further, we acknowledge that including a catalytically inactive *Kdm5b* construct also for the oocyte experiments (Fig. 1) would help determine whether the observed phenotype is due to KDM5B's demethylase activity or potential non-enzymatic functions. However, we would like to note that in our zygote and 2-cell embryo experiments (Fig. 3), embryos injected with *Del Kdm5b-yfp* mRNA showed H3K4me3 levels comparable to uninjected controls, whereas embryos injected with wild-type *Kdm5b-yfp* exhibited a marked reduction in H3K4me3. Although these results are from later developmental stages, they suggest that the phenotype observed is primarily due to the enzymatic activity of KDM5B, rather than non-enzymatic functions.

In conclusion, we opted for water-injected controls in given experiments because results from using the catalytically inactive *Kdm5b* construct, *Del Kdm5b-yfp*, showed that it had specific biological effects, making it unsuitable as an inert control. Water injection allowed us to control for any effects of the injection procedure itself, while also providing a consistent baseline using the same buffer used for mRNA dilution. Had we observed a difference in RNA-seq between the water- and *Kdm5b-yfp* mRNA-injected embryos, inclusion of an alternative control mRNA would have been warranted."

We would also like to note that, in relation to this matter, we have added a brief clarification under the 'Microinjection and live-cell imaging of YFP' section, which is the section preceding the new one (original text in black and the new in green):

"Water was favoured over *Del Kdm5b-yfp* mRNA as a control due a slightly higher but non-significant H3K4me3 level observed in *Del Kdm5b-yfp*-injected zygotes compared to uninjected controls (Fig. 3B, left), and a possible inhibitory effect on development (Fig. 5B). A detailed explanation of the control strategy is provided in the following section."

References cited in this response:

Zhang, B. et al. (2016) 'Allelic reprogramming of the histone modification H3K4me3 in early mammalian development', *Nature*, 537, pp. 553-557. Available at: <https://doi.org/10.1038/nature19361>.

Zhang, J. et al. (2025) 'Histone methyltransferases MLL2 and SETD1A/B play distinct roles in H3K4me3 deposition during the transition from totipotency to pluripotency', *The EMBO*

Journal, 44(2), pp. 437-456. Available at: <https://doi.org/10.1038/s44318-024-00329-5>.

- **Comment #32:**

Fig 1G: I think sample size between 1-2 is sufficient to report (previous comment #13). The authors should either increase the number to sufficient statistical power or delete it.

Response: We thank the reviewer for highlighting this point. In line with the reviewer's suggestion and the Handling Editor's instruction, we have now removed the data from Fig. 1G, and Fig. 1F (as suggested by reviewer 2, comment #39), as well as all related statements from the abstract, main text, and figure legends.

The following text with strikethrough has been removed:

- **Abstract (was line 35-36):** "~~Reduced H3K4me3 levels impair oocyte maturation but are compatible with embryonic development.~~" (Related to the removed Fig. 1G.)
NB: "impair oocyte maturation" has been kept and moved to the sentence before, where we in addition added the word "timing" after, to address the next comment (#33).
- **Results, section 1 (line 117-119):** "~~To further explore the role of broad H3K4me3 domains in maintaining transcriptional repression in the oocyte and zygote, as well as their functional importance for oocyte maturation and embryo development,...~~" (Related to the removed Fig. 1G.)
- **Results, section 1 (was line 162-171):** "~~Next, we tested whether removal of broad H3K4me3 in oocytes may have a functional effect on zygote formation and embryo development. We injected SN oocytes with Kdm5b-yfp mRNA, followed by IVM and chemical activation (Fig. 1A). The proportion of parthenogenetic zygotes obtained after Kdm5b-yfp mRNA or control injection, IBMX-induced arrest, IVM, and activation, showed no significant difference between groups (Fig. 1F, GV inj.). Similar results were observed after injection at MII stage (Fig. 1F, MII inj.). To further assess developmental competence, we cultured parthenogenetic zygotes to day 5 and found that a reduction of H3K4me3 at SN stage was also compatible with development to blastocyst (Fig. 1G). Based on a small number of zygotes from one replicate, this observation illustrates developmental potential rather than providing quantitative results.~~" (Related to the removed Fig. 1F and 1G.)
- **Results, section 1 (was line 175-177):** "~~However, maintenance of broad H3K4me3 domains beyond the SN stage appears not to be vital for subsequent oocyte activation and preimplantation development.~~" (Related to the removed Fig. 1F and 1G.)
- **Discussion:** - - - (Nothing related to the removed Fig. 1F and 1G.)
- **Materials and methods:** Removed all references to the removed Fig. 1F and 1G, including this (was line 542-546): "~~Next, the activated oocytes were either taken for 4 h EU treatment, fixed, and stained as described under Click-iT RNA imaging assay coupled with immunofluorescence staining (Fig. S2G), or they were transferred to a pre-equilibrated EmbryoSlide culture dish and put in the EmbryoScope incubator for blastocyst culture (Fig. 1G). The EmbryoSlide was prepared as explained under the "In vitro maturation (IVM)" section.~~"
- **Figure legends:** Removed entire legend for Fig. 1F and 1G (was line 1070-1080).

- **Comment #33:**

Abstract: Since there is no statistical difference on oocyte maturation between groups (Fig. 1C), it is not suitable for reporting "impaired oocyte maturation".

Response: We apologise for not being specific enough, because the reviewer is right that IVM rates (Fig. 1C) were not significantly different between groups. We meant to refer to the

delay in resuming meiosis (GVBD in Fig. 1D). We have therefore changed the phrasing slightly in the abstract, where it now says: "..., impair oocyte maturation timing,..." Further down in the abstract we have also added "timely": "Our findings suggest that broad H3K4me3 domains are required for oocyte genome silencing, timely maturation, and post-fertilisation silencing,..." We have used the same phrasings throughout the Results section and the Discussion when referring to these results.

Minor comments:

- **Comment #34:**
To avoid confusion for describing parthenogenetic 1-cell embryos, it will be suitable to use parthenotes or parthenogenetic 1-cell embryos, not zygotes. For PA embryos, they don't have 2PN but pPN (pseudopronuclei).

Response: We agree that the term "zygotes" may cause confusion, particularly since *in vivo*-fertilised zygotes are also included in this study. As suggested, we have now consistently used the term "parthenogenetic 1-cell (1C) embryos" for all relevant figures and text. However, we are reluctant to refer to the pronuclei in these embryos as "pseudopronuclei", as this term is most commonly associated with somatic cell nuclear transfer (SCNT) embryos (Matoba and Zhang, 2018), and has also been used to describe cytoplasmic vacuoles that resemble pronuclei (Van Blerkom, Bell and Henry, 1987). In this context, we believe that the pronuclei in parthenogenetic 1-cell embryos are indistinguishable from the maternal pronucleus in zygotes. We therefore feel that using "pseudopronuclei" in our study may introduce further confusion. That said, to prevent misunderstanding, we have removed the label "2PN" from the relevant groups in the revised Fig. 1E, Fig. S2G, and Fig. S2I.

References cited in this response:

Matoba, S. and Zhang, Y. (2018) 'Somatic Cell Nuclear Transfer Reprogramming: Mechanisms and Applications', *Cell Stem Cell*, 23(4), pp. 471-485. Available at: <https://doi.org/10.1016/j.stem.2018.06.018>.

Van Blerkom, J., Bell, H. and Henry, G. (1987) 'The occurrence, recognition and developmental fate of pseudo-multipronuclear eggs after in-vitro fertilization of human oocytes', *Human Reproduction*, 2(3), pp. 217-225. Available at: <https://doi.org/10.1093/oxfordjournals.humrep.a136517>.

- **Comment #35:**
Fig. S2B: The signal of H3K4me3 in NSN int (+EU) looks strange.

Response: Yes, we apologise for that. We have now updated the figure.

Reviewer 2

I thank the authors for their response and revision which well addressed part of my previous questions. On the other hand, several key questions remain unaddressed in the revision (see below), even though many of the related experiments seem to be straightforward and require only the existing experiment conditions in the paper. In addition, several figures lack sufficient samples which make the data inclusive or unconvincing (detailed below). As a result, I believe that it would be premature to publish the current version of manuscript. Please see below for the specific comments.

- **Comment #36:**
The authors proposed that maternal RNA decay could help explain the difference between Kdm5b OE in SN or MII oocytes (previous comment #8). This idea in principle can be easily tested. Could the authors show the H3K4me3 and YFP IF for the zygotes from Kdm5b OE SN oocytes? This may help validate their hypothesis.

Response: We agree that this would be a nice way to validate our hypothesis regarding the exogenous *Kdm5b-yfp* mRNA being subject to active degradation during M-decay.

Unfortunately, we did not collect YFP signal intensity data for the fixed parthenogenetic 1C embryos originating from injected SN oocytes (Fig. S2G). Live-cell YFP imaging was carried out 2 hours after injection to ensure expression of KDM5B-YFP, but not after fixation. In hindsight, we acknowledge that this may have been useful data to collect.

- **Comment #37:**

The authors only calculated the nascent RNA intensity in 9 zygotes which are too few to reach a conclusive conclusion (previous comment #9).

Response: We understand the reviewer's concern. As we wrote in our first response letter, we agree that larger numbers would have strengthened the conclusion but due to resource constraints, we are unable to expand this experiment. This is also why we have not made any definitive conclusion from this experiment (Fig. 1E). In response to previous comments on this matter, we did revise the manuscript title, which no longer includes something related to this, and we modified phrasings to better reflect that these are less robust results. Although there are only 9 parthenogenetic 1C embryos included here, we would like to highlight that the two groups are significantly distinct as found using the nonparametric Mann-Whitney test.

- **Comment #38:**

It was reported that KDM5B OE in mouse MII oocytes followed by IVF resulted in preimplantation arrest (PMID: 40449591). Since the authors have also overexpressed KDM5B in MII and found that parthenogenetic zygotes originating from these oocytes exhibited elevated levels of nascent RNA, it would greatly strengthen the paper if the authors can examine the blastocyst rate of these derived embryos (previous comment #16).

Response: Yes, we completely agree that it would be informative if we could assess the blastocyst development following injection and activation of MII oocytes. However, as noted in the previous response, we are unfortunately unable to expand experiments now.

We had a closer look at the publication you highlighted (Takasu et al., 2025). It is not clear how many IVF zygotes were included in the blastocyst development experiment. Furthermore, the relatively low blastocyst rate of 40% in the GFP-injected control group is a sign that caution should be taken in interpreting these results. Details regarding when the MII oocytes were isolated are lacking, which is critical to ensure they are within the optimal fertilisation window. Also, the methods state that "The mRNA-injected oocytes were cultured in M16 medium for 6 hours before in vitro fertilization". This raises the concern that the MII oocytes used for IVF may not have been optimal and therefore yielded poorer blastocyst development than expected. Consequently, we have chosen not to include this reference in our manuscript's discussion.

Reference cited in this response:

Takasu, A. et al. (2025) 'Characterization of H3K4me3 in mouse oocytes at the metaphase II stage', *Journal of Biological Chemistry*, 301(7). Available at: <https://doi.org/10.1016/j.jbc.2025.110308>.

- **Comment #39:**

Revised Fig. 1F: The results showed huge variations among replicates, with one replicate show nearly 0% and the other replicate showing 80%, making the results nearly impossible to evaluate. It would be premature to conclude based on these data. The authors are advised to include more replicates or remove related statements/data based on these results.

Response: In line with the reviewer's suggestion, we have now removed the data from Fig. 1F, and Fig. 1G (as suggested in your next comment and also by reviewer 1, comment #32), as well as all related statements from the abstract, main text, and figure legends. Please see further details under our response to comment #32.

- **Comment #32 (as for reviewer 1):**

Fig. 1G: One replicate with only 1-2 embryos is far from being sufficient for evaluation of

the developmental competency of the resultant embryos. Such data should not be used to draw conclusions and removed. More replicates with substantially more embryos are needed to make the conclusions of the paper convincing.

Response: We have now removed Fig. 1G and all related statements. Please see further details under our response to comment #32.

Reviewer 3

I welcome the extensive efforts the authors have made to provide greater clarity of their experimental design, choices and analyses, and in providing greater transparency in their results, e.g., in relation to sample numbers and replicates, etc. Their responses to reviewers' comments are universally well argued; the edits and additional information provided appropriate and adequate.

- Comment #40:**
 Suggestions to authors: It has very recently been reported that non-canonical H3K4me3 and H2A.Z deposition in broad domains in mouse oocytes is mutually reinforcing, and that depletion of nCH2A.Z by knocking out the two H2A.Z-encoding genes (H2afv, H2afz) in oocytes phenocopies many of the phenotypes of the Mll2-KO (PMID: 40514539 - although an effect on transcriptional silencing was not explicitly tested in that study). That study helps emphasize the exquisite dependencies of histone marks, and variants, in oocytes (both positive and negative), where altering one mark will undoubtedly lead to changes in others. Against this background, mechanistic inferences are challenging, if the mechanistically relevant mark is not what is measured in any particular study. On the other hand, in the current manuscript, the authors may be justified in identifying broad H3K4me3 as being responsible for enforcing transcriptional silencing in GV oocytes, rather than a mark/variant dependent on H3K4me3, because the intervention deployed (KDM5B over-expression for a limited period rather than a genetic knock-out over the oocyte growth phase) may limit the effect to H3K4me3 without incurring further changes in chromatin states. The authors might like to reflect this point.

Response: Thank you for highlighting this interesting study (Mei et al., 2025) that we read with excitement, as we have also been studying the significance of H2A.Z in oocytes (Fossli et al., 2025). The reviewer makes a good point here. We have taken the advice and added the following to the discussion:

“Interestingly, broad H3K4me3 establishment in oocytes was recently found to be interdependent with incorporation of the histone variant H2A.Z (Mei et al., 2025). Such interdependence of histone marks and variants complicates mechanistic interpretation, as perturbation of one modification can indirectly influence others. Nonetheless, by using transient KDM5B overexpression, our study is designed to primarily target H3K4me3, supporting our conclusions.”

Reference cited in this response:

Mei, H. et al. (2025) ‘H2A.Z reinforces maternal H3K4me3 formation and is essential for meiotic progression in mouse oocytes’, *Nature Structural & Molecular Biology* [Preprint]. Available at: <https://doi.org/10.1038/s41594-025-01573-x>.

Fossli, M., Ilasslan, E., Skuland, T. et al. (2025) ‘Major waves of H2A.Z incorporation during mouse oogenesis’. *bioRxiv*, p. 2025.06.14.659461. Available at: <https://doi.org/10.1101/2025.06.14.659461>.

Third decision letter

MS ID#: dev.204638R2

MS TITLE: Broad H3K4me3 domains support oocyte genome silencing and maturation but are dispensable for repression in early embryos

AUTHORS: Trine Skuland, Madeleine Fossli, Sherif Khodeer, Endalkachew Ashenafi Alemu, Jens Vilstrup Johansen, Marie Indahl, Blanca Corral Castroviejo, Yanjiao Li, Mads Lerdrup, John Arne Dahl, Peter Zoltan Fedorcsak and Gareth D. Greggains

Dear Dr Greggains,

I am happy to tell you that your manuscript has been accepted for publication in Development, pending our standard publication integrity checks.